

# Alpha and beta diversity patterns of polychaete assemblages across the nodule province of the Clarion-Clipperton Fracture Zone (Equatorial Pacific)

Paulo Bonifácio[1], Pedro Martinez-Arbizu[2], Lénaïck Menot[1]

[1]Ifremer, Centre Bretagne, REM EEP, Laboratoire Environnement Profond, ZI de la Pointe du Diable, CS 10070, F-29280 Plouzané, France
[2] Senckenberg am Meer, DZMB, F-26382 Wilhelmshaven, Germany

*Correspondence to*: Lénaïck Menot (lenaick.menot@ifremer.fr), Paulo Bonifácio (bonif@me.com)

**Abstract.** In the abyssal Equatorial Pacific Ocean, most of the seafloor of the Clarion-Clipperton Fracture Zone (CCFZ), a 6

million km² polymetallic nodule province, has been preempted for future mining. In light of the large footprint that mining would leave, and given the diversity and the vulnerability of the abyssal fauna, the International Seabed Authority has implemented a regional management plan that includes the creation of nine areas of particular environmental interest (APEIs) located at the periphery of the CCFZ. The APEIs were defined based on the best – albeit very limited – scientific knowledge for the area. The fauna and habitats in the APEIs are unknown, as are species' ranges and the extent of biodiversity across the

CCFZ.

As part of the Joint Programming Initiative Healthy and Productive Seas and Oceans (JPI Oceans) pilot action "Ecological aspects of deep-sea mining", the SO239 cruise aimed at improving species inventories, determining species ranges, identifying the drivers of beta diversity patterns and assessing the representativeness of an APEI. Four exploration contract areas and an APEI (APEI#3) were sampled along a gradient of sea-surface primary productivity that spanned a distance of 1440 km in the

eastern CCFZ. Between 3 and 8 quantitative box cores (0.25 m²; 0–10 cm) were sampled in each study area, resulting in a large collection of polychaetes that were morphologically and molecularly (COI and 16S genes) analyzed.

A total of 275 polychaete morphotypes were identified. Only one morphotype was shared among all five study areas and 49% were singletons. The patterns in community structure and composition were mainly attributed to variations in food fluxes at the regional scale and nodule density at the local scale. The four exploration contract areas belong to a mesotrophic province.

The distance-decay of similarity among the four areas provides an estimated species turnover of 0.04 species km⁻¹ and an average species range of 25 km. The polychaete assemblage in APEI#3 showed the lowest densities, lowest diversity as well as very low, distant-independent similarity with the other four study areas. Given that APEI#3 is located in an oligotrophic province and separated from the CCFZ by the Clarion Fracture Zone, our results call into question the representativeness and the appropriateness of APEI#3 to meet its purpose of preserving the biodiversity of the CCFZ fauna. Two methods for

estimating the total number of polychaete species gave estimates that ranged from 498 to 240 000 species. Both methods are biased by the high frequency of singletons in the dataset, which likely result from under-sampling; our estimates thereby



merely reflect our level of uncertainty. The assessment of potential risks and scales of biodiversity loss due to nodule mining thus requires an appropriate inventory of species richness in the CCFZ.

## 1 Introduction

The abyssal plain is vast, covering 54 % of the Earth's surface and 75 % of the ocean floor, typically located between 3000–
6000 m depth; it generally features low temperature, low current, and well-oxygenated, oligotrophic waters (Gage and Tyler, 1991; Smith and Demopoulos, 2003; Ramirez-Llodra et al., 2010). Only about 1 % of abyssal plains have been explored to date: much remains to be discovered. In particular, polymetallic nodule fields are one of the unique habitats in the abyss (Ramirez-Llodra et al., 2010; Vanreusel et al., 2016). Nodules are potato-shaped, variably sized aggregations of minerals, mainly manganese and iron, that are patchily distributed (Michael et al., 2000; Morgan, 2000). Polymetallic nodules were
discovered during the Challenger expedition in the 1870s at depths below 4000 m in the Pacific, Atlantic and Indian oceans (Boudreau and Scott, 1978). In the Equatorial Pacific Ocean, the Clarion-Clipperton Fracture Zone (CCFZ) harbors the largest polymetallic nodule field with nodule densities as high as 75 kg m$^{-2}$ (average 15 kg m$^{-2}$) and possibly containing 34 billion metric tons of manganese (Michael et al., 2000; Morgan, 2000). The presence of abundant metal resources (e.g. manganese, iron, copper, nickel and cobalt) has attracted the interest of industries. Established by the United Nations Convention on the
Law of the Sea (UNCLOS), the International Seabed Authority (ISA) manages the deep-sea mineral resources in international waters and is in charge of protecting fauna against any pollution or other hazards (Articles 145, 156, UNCLOS) (Lodge et al., 2014). Currently, the ISA has granted 16 nodule exploration contracts and approved nine "areas of particular environmental interest" (APEIs) for preservation (Lodge et al., 2014) in the CCFZ.

Nodule mining will clearly have detrimental effects on the benthic ecosystem, but the severity of the impacts is difficult to
predict. Long-term surveys of small-scale disturbances or mining tests have shown that the direct impacts of seafloor disturbances may last for over 30 years in the CCFZ (Vanreusel et al., 2016; Jones et al., 2017). Such small experiments however hardly mimic the cumulative impacts of any single nodule mining operation that could last for 20 years. A mining operation is anticipated to directly affect over 100 km$^2$ yr$^{-1}$ of the seabed and create sediment plumes that can indirectly increase the footprint of mining by a factor 2 to 5 (Oebius et al., 2001; Glover and Smith, 2003; Volkmann and Lehnen, 2018). Beyond
this scaling issue, the extent of biodiversity and species' ranges in the CCFZ are two major unknowns that prevent the assessment of potential biodiversity loss due to nodule mining. The few biodiversity studies undertaken so far in the CCFZ have revealed the high diversity of communities of megafauna [over 130 morphotypes (Amon et al., 2016; Simon-Lledó et al., 2019)], polychaetes [over 180 morphotypes (Paterson et al., 1998; Glover et al., 2002; Wilson, 2017)], isopods [over 160 morphotypes (Wilson, 2017)], tanaids [over 100 morphotypes (Wilson, 2017; Pabis et al., submitted)] and nematodes [over
300 morphotypes (Miljutina et al., 2010)]. Overall, over 870 morphotypes are already known in the CCFZ, but almost none have been named and 90 % of them are likely new to science (Glover et al., 2002; Miljutina et al., 2010). Therefore, new inventories of CCFZ biodiversity cannot be compared with these previous ones. To overcome this bias, DNA taxonomy is





increasingly used (Glover et al., 2016). In the CCFZ, in two exploration contract areas separated by 1300 km, the first assessment of macrofaunal diversity based on DNA taxonomy already increased the number of known polychaetes to 233 "molecular operational taxonomic units" (MOTUs) (Janssen et al., 2015). This study further highlighted three characteristics of abyssal biodiversity: (a) high rates of species turnover with only 12 % of polychaete MOTUs and 1 % of isopod MOTUs

shared between the two areas; (b) high frequencies of singletons ranging from 60 % to 70 % for polychaetes and isopods, respectively; and (c) cryptic richness within polychaete and isopod morphotypes, suggesting that previous surveys have underestimated alpha and beta diversity of these two taxa.

Considering the large footprint of nodule mining disturbances on the seafloor, as well as the diversity and vulnerability of the abyssal fauna, the need for marine spatial planning to preserve species, habitats and functions in the CCFZ has emerged,

concomitant to a renewed interest for deep-sea mineral resources (Wedding et al., 2013). Due to the paucity of biological data in the CCFZ, the spatial management plan was designed mainly based on nitrogen flux at 100 m depth (a proxy for trophic inputs to the seafloor), modeled nodule densities, the distribution of large seamounts and the dispersal distances of shallow water taxa (Wedding et al., 2013). The nine proposed 400 x 400 km managed (non-mining) areas were included in the regional management plan for the CCFZ and designated as APEIs (Lodge et al., 2014). Most of the CCFZ however has already been

preempted to current exploration contracts and areas reserved for future exploration. The distribution of APEIs at the periphery of the CCFZ thus deviates from an optimal design.

The European project "Managing Impacts of Deep-seA reSource exploitation" (MIDAS) and the "Joint Programming Initiative Healthy and Productive Seas and Oceans" (JPI Oceans) pilot action "Ecological aspects of deep-sea mining" aimed at improving the scientific grounds on which to assess and manage the potential impacts arising from nodule mining. In this

context, four exploration contract areas and one APEI (separated by 240 to 1440 km) were sampled along a sea-surface primary productivity gradient from east to west across the eastern portion of the CCFZ nodule province. Polychaetes were identified using a combination of morphological and molecular criteria. The structure and composition of polychaete assemblages were analyzed to describe and identify alpha and beta diversity patterns, test the hypotheses that support spatial conservation planning in the CCFZ, assess the representativeness of an APEI and potentially improve the assessment of potential risks to

biodiversity due to nodule mining.

## 2 Materials and methods

### 2.1 Clarion-Clipperton Fracture Zone

The CCFZ is located in the Equatorial Pacific Ocean between the Clarion Fracture to the north and the Clipperton Fracture to the south, and between Kiribati to the west and Mexico to the east (Fig. 1). This area covers about 6 million $km^2$ being

composed of a high variety of habitats such as abyssal hills or seamounts, as well as polymetallic nodule fields (Glover et al., 2016). As part of the JPI Oceans project "Ecological aspects of deep-sea mining", the EcoResponse cruise SO239 took place from 9 March to 30 April 2015 aboard the RV *Sonne* (Martinez Arbizu and Haeckel, 2015). Sampling during the cruise focused





on four exploration contract areas as well as APEI number 3 (APEI#3; Fig. 1). All five study areas were located between 4000 and 5000 m depth (Fig. 1). The four exploration areas were licensed by the ISA to the Federal Institute for Geosciences and Natural Resources of Germany (BGR); the InterOceanMetal Joint Organization (IOM); the G-TEC Sea Mineral Resources NV (GSR); and the *Institut Français de Recherche pour l'Exploitation de la Mer* (Ifremer). Furthermore, the ISA administrates

APEI#3 as part of the regional environmental plan for the CCFZ. The distances separating the areas ranged from 243 km (BGR to IOM) to 1440 km (BGR to Ifremer or APEI#3).

## 2.2 Sampling strategy

Within each area, macrofaunal samples were collected using an USNEL spade box corer of 0.25 m$^2$ (Hessler and Jumars, 1974) with five to nine replicates in each area, totaling 34 box core samples (Table 1). Although in the BGR and IOM areas

nine box cores had been sampled from different localities within the study area, they were considered as replicates (as were environmental variables). The overlying water column was siphoned and filtered using a sieve of 300 µm of mesh size. The box core sample surface was photographed, and all nodules picked up from the sediment surface, washed and individually measured and weighed. The upper 10 cm of each core was sliced into three layers (0–3, 3–5 and 5–10 cm), each layer transferred into cold seawater and sieved using the same mesh size. The overlying water residue and the 0–3 cm layer were

immediately sieved in the cold room with cold seawater (4 °C) and then live-sorted. All polychaete specimens were photographed, individualized and preserved in cold (-20 °C) 80 % ethanol and then kept at -20 °C (DNA-friendly). The 0–3 cm residue, 3–5 and 5–10 cm layers were fixed in formalin for 48 to 96 h and preserved in 96 % ethanol and later sorted in the laboratory (not DNA-friendly). The sieve residues from the overlying water and the washed nodules were combined with all layers for the community analysis. In the laboratory, from each DNA-friendly polychaete specimen, a small piece of tissue

was dissected, fixed in cold 96 % ethanol and frozen at -20 °C for molecular studies (see Section 2.3 DNA extraction, amplification, sequencing, and alignment).

## 2.3 DNA extraction, amplification, sequencing, and alignment

The DNA of the subsampled tissues was extracted using a NucleoSpin Tissue kit (Macherey-Nagel), following the manufacturer's protocol. Approximately 450 base pairs (bp) of 16S, 700 bp of COI (cytochrome c oxidase subunit I) and 1600

bp of 18S genes were amplified using the following primers: Ann16SF and 16SbrH for 16S (Palumbi, 1996; Sjölin et al., 2005); polyLCO, polyHCO, LCO1490, and HCO2198 for COI (Folmer et al., 1994; Carr et al., 2011); and 18SA, 18SB, 620F, and 1324R for 18S (Medlin et al., 1988; Cohen et al., 1998; Nygren and Sundberg, 2003) for 18S. The polymerase chain reaction (PCR) mixtures of 25 µL contained 5 µL of Green GoTaq® Flexi Buffer (final concentration of 1X), 2.5 µL of MgCl$_2$ solution (final concentration of 2.5 mM), 0.5 µL of PCR nucleotide mix (final concentration of 0.2 mM each dNTP), 9.875 µL

of nuclease-free water, 2.5 µL of each primer (final concentration of 1 µM), 2 µL template DNA and 0.125 U of GoTaq® G2 Flexi DNA Polymerase (Promega). The temperature profile was as follows: 95 °C/240 s – (94 °C/30 s – 52 °C/60 s – 72 °C/75 s (for 16S and COI) or 180 s (for 18S) *35 cycles (for 16S or 18S) or *40 cycles (for COI)) – 72 °C/480 s. PCR products,



visualized after electrophoresis on 1 % agarose gel, were sent to the MacroGen Europe Laboratory in Amsterdam (the Netherlands) to obtain sequences, using the same set of primers as used for the PCR.

Overlapping sequence (forward and reverse) fragments were aligned into consensus sequences using Geneious Pro 8.1.7 2005–2015 (Biomatters Ltd). For COI, the sequences were translated into amino-acid alignments and checked for stop codons to

avoid pseudogenes. The minimum length coverage was 207 bp for 16S, 327 bp for COI and 1615 bp for 18S.

The sequences were blasted in GenBank to check for the presence of contamination. Each set of genes was aligned separately using the following plugins: MAAFT (Katoh et al., 2002) for 16S and 18S; and MUSCLE (Edgar, 2004) for COI. All sequences obtained in this study have been deposited in BOLD (http://www.boldsystems.org; (Ratnasingham and Hebert, 2007)) or GenBank (http://www.ncbi.nlm.nih.gov/genbank/).

**2.4 Operational taxonomic units (OTUs)**

Preserved specimens were examined under a Leica M125 stereomicroscope and a Nikon Eclipse E400 microscope, and morphologically identified using the deep-sea polychaete fauna bibliography (Fauchald, 1972, 1977; Böggemann, 2009) at the lowest taxonomic level possible (species-morphotype). To separate closely related species, the principle of phylogenetic species was used, whereby the genetic divergence among specimens belonging to the same species (intraspecific) is smaller

than the divergence among specimens from different species (interspecific) (Hebert et al., 2003b). In the distribution of pairwise divergences among all sequences of a typical barcode data set, a gap can be observed between intraspecific and interspecific variations. Molecular operational taxonomic units (MOTUs) were generally recognized using a threshold of 97 % or 99 % similarity between COI and 16S sequences, respectively (Hebert et al., 2003a; Hebert et al., 2003b; Brasier et al., 2016). The similarity of sequences within species was considered when identifying morphologically similar species.

**2.5 Environmental data**

Environmental data were compiled from Hauquier et al. (2018) and Volz et al. (2018). Sediment samples were collected with a multi-corer or a gravity corer during the same cruise and in the same areas (see Martinez Arbizu and Haeckel, 2015 for details). The sediment characteristics studied by Hauquier et al. (2018) included a clay fraction (< 4 µm), a silt fraction (4–63 µm), total nitrogen (TN in weight%), total organic carbon (TOC in weight%) and chloroplastic pigment equivalents (CPE in

µg ml$^{-1}$). Nodules were weighed onboard for each box-core sample to calculate nodule density (kg m$^{-2}$; Table 1). Particulate organic carbon flux (POC, mg C m$^{-2}$ d$^{-1}$) at the seafloor for our study areas were provided by Volz et al. (2018). At the regional scale, POC flux (mg C m$^{-2}$ d$^{-1}$) at the seafloor was approximated using net surface primary production provided by the Ocean Productivity site (Westberry et al., 2008) averaged over years 2002 to 2018 and applying the Suess algorithm (POC at the seafloor as a function of the net primary production scaled by depth; Suess, 1980; Table 2).



## 2.6 Regional-scale data

To put the results of our study in a regional context, we compiled data from previous surveys of polychaete assemblages in the Northeast Pacific, including CLIMAX II sampled in 1969, DOMES A, B and C in 1977 and 1978, ECHO I in 1983, PRA in 1989, EqPac in 1992, Kaplan East in 2003, Kaplan West and Central in 2004, KR5 in 2012, 2013 and 2014 and GSRNOD15A

(B4N01, B4S03 and B6S02) in 2015 (Paterson et al., 1998; Glover et al., 2002; Wilson, 2017). From these studies, we compiled (when available) the mean abundance (ind. 0.25 $m^{-2}$), total number of species, ES163 and bootstrap (Table 2).

## 2.7 Data analysis

### 2.7.1 Univariate analyses

Mean abundance and mean number of species per box-core were used as descriptors of alpha diversity. A few cryptic or

damaged species that could not be classified to a lower taxonomical level were included in total abundance calculations but excluded from subsequent diversity analyses. To compare diversity among the studied areas, rarefaction curves were computed based on the total number of individuals and the total number of box core samples (Hurlbert, 1971; Gotelli and Colwell, 2001). Non-parametric estimators of species richness were used to estimate the total number of species at local and regional scales. Abundance-based estimators included Chao1 and abundance-based coverage estimator (ACE) (O'Hara, 2005; Chiu et al.,

2014). Incidence-based estimators included Chao2 (Chao, 1984), first- and second-order jackknife (Jackknife 1st and Jackknife 2nd, respectively) (Burnham and Overton, 1979) and bootstrap (Smith and van Belle, 1984). A Venn diagram was used to show the distribution of rare, wide and common species across the CCFZ.

Univariate analyses relied on non-parametric tests. The Kruskal-Wallis rank sum test was used to test differences among areas (Hollander and Wolfe, 1973); and the Conover multiple pairs-rank comparisons (adjusted p-value by Holm) was used to

identify the pairs showing differences (Conover and Iman, 1979; Holm, 1979). Spearman correlations were sought between biotic and abiotic variables, using data from the SO239 cruise in the CCFZ and data compiled from the literature. The latter analysis aimed at testing correlations between biotic variables and POC fluxes at the regional scale.

### 2.7.2 Multivariate analyses

Three indices of faunal similarity were used in multivariate analyses, the chord-normalized expected species shared (CNESS),

the new normalized expected species shared (NNESS) (Trueblood et al., 1994; Gallagher, 1996) and the Jaccard-family indices (Baselga, 2010; Legendre, 2014). The CNESS and NNESS are computed from probabilities of species occurrence in random draws of $m$ individuals. Low values of $m$ give a high weight to dominant species, high values of $m$ give a high weight to rare species. The best trade-off value of $m$ is the one providing the highest Kendall correlation between the similarity matrix for $m$ = 1 and the similarity matrix for $m$ = $m$max. The value of $m$max is given by the total abundance of the least abundant sample.

CNESS is a distance metric that can be used to perform a redundancy analysis (RDA) (Legendre and Legendre, 2012). The RDA is a constrained multivariate analysis that tests the influence of multiple environmental covariates on multispecific





assemblages. Species contributing significantly to the ordination were plotted out of the equilibrium circle in RDA (scaling 1). The best set of environmental variables was selected using forward, backward and stepwise selection procedures among the environmental variables available (see Section 2.5 Environmental data): clay fraction, silt fraction, TN, TOC, CPE, nodule density and POC flux at the seafloor. Furthermore, when selected variables had more than 80 % co-correlation, they were

excluded, and the selection procedure started over again. Also, the variance inflation factor (VIF) was used to verify the possible linear dependency among variables in the RDA model.

The NNESS index was used to perform a distance-decay analysis as in Wilson (2017). Distance-decay screens for a negative correlation between faunal similarities and geographic distances among pairs of areas. Wilson (2017) used the slope of linear regression between NNESS and distance to compute the rate of change (species km$^{-1}$) and the species range (km species$^{-1}$).

The rate of change is the slope of linear regression between NNESS and distance multiplied by the mean total estimated species from all areas. The species range is the inverse of the rate of change.

The Jaccard-family indices were used to partition beta-diversity into its three components: similarity; turnover – which is dissimilarity due to species turnover – and nestedness – which is dissimilarity due to differences in the number of species (Baselga, 2010).

All analyses were conducted using the R language (R Core Team, 2018) with RStudio (RStudio Team, 2015) and the following specific packages or functions: adespatial (Dray et al., 2019), BiodiversityR (Kindt and Coe, 2005), fossil (Vavrek, 2011), vegan (Oksanen et al., 2015), VennDiagram (Chen, 2018), beta.div.comp (Legendre, 2014), ness (Menot, 2019).

## 3 Results

### 3.1 Abundance and alpha diversity

During the SO239 cruise, 1233 polychaete specimens were sampled in the five study areas. The mean abundance in each study area tended to decrease from east to west with high spatial variation (Fig. 2a). Mean densities ranged from $58 \pm 18$ ind. 0.25 m$^{-2}$ in the BGR area to $5 \pm 2$ ind. 0.25 m$^{-2}$ in APEI#3. The mean abundance differed significantly between areas (Kruskal-Wallis test, $p < 0.001$). The pairwise comparison test (Conover-Holm) showed that (a) APEI#3 had significantly lower abundance than the other areas ($p \leq 0.01$) except Ifremer; (b) the Ifremer exploration area ($28 \pm 8$ ind. 0.25 m$^{-2}$) had significantly

lower abundance than the BGR and GSR areas ($59 \pm 10$ ind. 0.25 m$^{-2}$) ($p < 0.001$); and (c) the IOM area ($37 \pm 10$ ind. 0.25 m$^{-2}$) had significantly lower ($p < 0.01$) abundance than the BGR and GSR areas. Furthermore, the mean abundance per box core was significantly correlated (Spearman correlations; Fig. 3) with the mean number of species (rho = 0.96, $p < 0.001$), POC at the seafloor (rho = 0.61, $p < 0.001$), CPE (rho = 0.52, $p < 0.01$), clay fraction (rho = -0.43, $p < 0.05$), silt fraction (rho = 0.40, $p < 0.05$), and nodule density (rho = 0.36, $p < 0.05$). At the scale of the NE Pacific, polychaete densities were also significantly

correlated with POC flux at the seafloor (rho = 0.75, $p < 0.001$, n = 19; Fig. 4a).

The polychaetes belonged to 41 families (Fig. 5a) with the most abundant being spionids (20 %), cirratulids (13 %), paranoids (11 %), and lumbrinerids (6 %). Spionids showed the highest relative abundance at the Ifremer (34 %), GSR (27 %) and IOM



(19 %) areas, whereas cirratulids were dominant at APEI#3 (36 %) and the BGR (17 %) areas. The relative contributions of trophic guilds also varied among the areas (Fig. 5b). In particular, carnivores were more common at BGR, IOM and GSR (23–28 %) than at Ifremer and APEI#3 (12–14 %), whereas deposit feeders were overwhelmingly dominant at Ifremer and APEI#3 (78–86 %) and less so at BGR, IOM and GSR (63–65 %). Suspension feeders, omnivores and scavengers contributed to less

than 13 % of abundance in each area and were not found in APEI#3.

Off the 1223 polychaetes, 1118 specimens belonging to 78 possible genera within 40 families were identified down to morphospecies (see Section Data availability). The 115 remaining specimens were too damaged or cryptic/doubtful to be assigned to a morphospecies and were thus not included in diversity and composition analyses. The DNA-friendly samples totaled 430 specimens; 265 of which were successfully barcoded with either or both the COI and 16S genes. The success rates

were 17 % for COI and 60 % for 16S. The COI gene was successfully sequenced for 71 specimens totaling 45 MOTUs; for the 16S gene, 259 specimens were successfully sequenced covering 104 MOTUs; only 65 specimens were successfully sequenced using both genes and yielded 40 MOTUs.

Based on both morphological and molecular identification, a total of 275 morphospecies (i.e. OTUs) were recognized. The mean number of species tended to decrease from east to west with high spatial variation (Fig. 2b). Mean richness varied from

37 ± 10 taxa 0.25 m$^{-2}$ in BGR to 3 ± 2 taxa 0.25 m$^{-2}$ in APEI#3. The number of species differed significantly among areas (Kruskal-Wallis test, p < 0.001). The pairwise comparison test (Conover-Holm) showed that the number of species per box core was (a) significantly lower at APEI#3 than all other areas (p ≤ 0.01) except Ifremer; (b) significantly lower at Ifremer (19 ± 5 taxa 0.25 m$^{-2}$) than at BGR and GSR (35 ± 7 taxa 0.25 m$^{-2}$) (p < 0.001); and (c) significantly lower at IOM (25 ± 6 taxa 0.25 m$^{-2}$) (p < 0.05) than at BGR and GSR. A total of 156 species (observed species richness, Sobs) were sampled at BGR

from eight box core samples, 107 species at IOM from eight box cores, 104 species at GSR from five box cores, 73 species at Ifremer from six box cores and 9 species at APEI#3 from three box cores (Table 3). Species rarefaction curves, based on individuals or samples, did not reach an asymptote at the local scale (Fig. 6a, b). Individual-based rarefaction curves did not show any clear diversity patterns among study areas (Fig. 6a). Sample-based rarefaction curves followed a pattern similar to abundance (Fig. 6b). From a random draw of three box cores, BGR and GSR, with 82 and 77 species, respectively, had higher

expected numbers of species than did IOM and Ifremer with 58 and 45 species, respectively; APEI#3, with only 9 species, had the lowest expected number of species (Fig. 6b, Table 3). The non-parametric estimators of local diversity followed the same patterns with the highest values for BGR and the lowest for APEI#3 (Table 3).

The mean number of species in each study area was significantly correlated (Spearman correlations; Fig. 3) with POC flux (rho = 0.62, p < 0.001), CPE (rho = 0.55, p < 0.01), clay fraction (rho = -0.45, p < 0.05), silt fraction (rho = 0.38, p < 0.05),

and TOC (rho = 0.39, p < 0.05).

At the scale of the NE Pacific, neither ES163 (rho = 0.59, p = 0.09, n = 9) nor bootstrap (rho = 0.10, p = 0.8, n = 8) were correlated with POC flux at the seafloor (Fig. 4b, c).



## 3.2 Beta and gamma diversity

In the RDA, the forward selection procedure kept CPE, clay fraction and nodule density as the best explanatory variables. The model explained 13 % ($R^2_{adj}$) of the total variance in the composition of polychaete assemblages (Fig. 7a). The first axis of the RDA discriminated the eastern areas (BGR, IOM, GSR) from the western areas (Ifremer, APEI#3). The second axis of the

RDA discriminated Ifremer from APEI#3 but also captured local-scale variation, because replicate samples within areas were distributed along this second axis. The CPE concentrations mostly explained variance along the first axis. CPE was also positively and highly correlated with POC flux and TOC (Fig. 3). The first axis of the RDA thus illustrates the influence of food inputs on species composition. The clay fraction contributed to the first and the second axis of the RDA. Grain size distribution indeed differentiated APEI#3 from all other areas in the CCFZ (see Hauquier et al., 2018 for details). In the RDA,

the clay fraction accounted for the large dissimilarity in species composition of the APEI#3. Nodule density was the main contributor to the second axis of the RDA. Variation in nodule density likely accounted for some of the local variation in species composition. The ordination of species (Fig. 7b) showed that *Lumbrinerides* sp. 2107 was the species most characteristic of the eastern areas; a cirratulid (*Aphelochaeta* sp. 2062) and a maldanid (Maldanidae sp. 121) were characteristic of APEI#3; and two spionids (*Aurospio* sp. 249 and *Laonice* sp. 349), a paraonid (*Levinsenia* sp. 498) and an opheliid

(*Ammotrypanella* sp. 2045) were characteristic of the Ifremer area.

The distance-decay of similarity showed two different patterns (Fig. 8a, b). APEI#3 had very low values of NNESS compared with all other areas, irrespective of distance (Fig. 8a). There was no statistically significant correlation between NNESS and distance ($R^2_{adj}$ = 18 %, p = 0.12). However, without APEI#3, the NNESS values among pairs of exploration contract areas (Fig. 8b) within the CCFZ per se were negatively correlated with distance ($R^2_{adj}$ = 0.85, p = 0.006). The slope of the linear

regression (-0.0003) multiplied by the mean of species richness estimators for each area (Table 3) provided a rate of species change that ranged from 0.04 species km$^{-1}$ with the bootstrap estimator (mean species richness of 135 species) to 0.07 species km$^{-1}$ for the ACE estimator (mean species richness of 234 species). The inverse of these rates of species change predicted geographic ranges of 14 to 25 km.

Beta diversity was thus high across the CCFZ and particularly so between the exploration contract areas, south of the Clarion

Fracture Zone, and APEI#3, north of the Clarion Fracture Zone. In addition, the decomposition of beta diversity showed that dissimilarity was mainly due to species turnover (91 %) and not nestedness (9 %). However, species turnover was driven by singletons. The Venn diagram (Fig. 9) showed that, in each area, at least 30 % and up to 67 % of species were unique to one area, so that overall 169 out of 275 species were unique to a given area. Of these, 134 species were singletons. Only one species, *Aurospio* sp. 249, was sampled in all five areas, 16 species (6 %) were sampled in four areas, 33 species (12 %) were

shared among three areas and 56 species (20 %) were shared between two areas.

When all individuals and samples were pooled together, rarefaction curves did not level off (Fig. 10a, b) and the number of singletons steadily increased with increasing sample size (Fig. 10b). At this regional scale, non-parametric estimators of species richness ranged from 334 to 498 species (Table 3).





## 4 Discussion

### 4.1 Major forces driving local- and regional-scale patterns in community structure and composition

Food supply, sediment grain size and the density of nodules are the three main environmental factors that seem to drive the structure and composition of polychaete assemblages in the CCFZ.

Nodules have antagonistic influences on different size groups of benthic communities. Meiofaunal assemblages are less abundant in nodule-rich than in nodule-free sediments (Miljutina et al., 2010; Pape et al., 2018). Nodules however increase habitat heterogeneity, providing hard substrate for sessile organisms and generally enhancing the standing stocks of both sessile and vagile megafauna (Amon et al., 2016; Vanreusel et al., 2016; Simon-Lledó et al., 2019). Similarly, nodules seem to enhance macrofaunal density (De Smet et al., 2017) and diversity (Yu et al., 2018). Our results support the reported positive and

significant relationship between polychaete abundance and nodule density (De Smet et al. (2017). The macrofauna in nodule fields may benefit from increased food supply and the release from competition with meiofauna. Nodules increase seafloor roughness, thereby increasing friction (Sternberg, 1970; Boudreau and Scott, 1978) and potentially sediment deposition rates. The large sessile suspension feeders may similarly enhance biodeposition (Graf and Rosenberg, 1997). Both processes may stabilize sediments and increase organic carbon supply as tube lawns do, for example (Michael et al., 2000). An increase in

food supply may explain the higher densities of polychaetes in nodule-rich areas. The divergent response of meiofauna to the presence of nodules further suggests some sort of competition between meiofauna and macrofauna. The contribution of meiofauna to benthic biomass generally increases along a bathymetric gradient to outweigh that of macrofauna at abyssal depths (Thiel, 1975; Rex et al., 2006; Wei et al., 2010). This pattern is assumed to reflect a selective advantage for small size at very low levels of food input (Thiel, 1975, 1979; Sebens, 1982, 1987; Rex and Etter, 1998). Sibuet et al. (1989) reported

however a linear relationship between meiofaunal and macrofaunal biomass at abyssal sites. Both size classes indeed co-varied with organic carbon burial flux, which suggests the occurrence of a dynamic equilibrium between meiofauna and macrofauna at abyssal depths. Due to its small size, meiofauna is likely more efficient at exploiting the low level of food input, but this interstitial fauna may also be more sensitive to high nodule coverage because its ambit is largely limited to superficial sediments. The opposite effects of nodule coverage on meiofaunal and macrofaunal densities may thus lie in a release from

the advantage of being smaller in the abyss, inducing a shift in size-group equilibrium toward increased macrofaunal densities. These results suggest that nodule coverage have an influence on the functioning of the ecosystem, because it modifies biotic interactions and resource allocation among functional groups.

At regional to global scales, food input is among the main forcing factors of the structure and functions of the abyssal ecosystem, which mainly rely upon 0.5–2 % of the organic carbon derived from sea-surface primary production (Rowe et al.,

1991; Smith et al., 1997; Smith et al., 2008a). Variations in sea-surface primary productivity divide the NE Pacific abyss into three main areas (Sokolova, 1997; Hannides and Smith, 2003; Smith and Demopoulos, 2003): the eutrophic abyss in the equatorial upwelling zone (-5°S–5°N) with POC flux about 1–2 g C m$^{-2}$ year$^{-1}$; the mesotrophic abyss in the Equatorial North Pacific (5–15°N) with a POC flux of about 0.5–1.5 g C m$^{-2}$ year$^{-1}$ and the oligotrophic abyss underlying the North Pacific



Subtropical Gyre (15–35˚N) with a POC flux lower than 0.5 g C m$^{-2}$ year$^{-1}$. Our metadata analysis confirmed that polychaete abundance was significantly and positively correlated with POC flux, distinguishing areas in the oligotrophic abyss (APEI#3, CLIMAX II, DOMES A, EqPac 9 and Kaplan West) with low abundance (4–21 ind. 0.25m$^{-2}$) from areas in the mesotrophic abyss (Kaplan Central, Ifremer, PRA, ECHO 1, GSRNOD15A, GSR, IOM, Kaplan East and BGR) with average to high

abundance (14–85 ind. 0.25m$^{-2}$) and areas in the eutrophic abyss (EqPac 0, 2 and 5) with abundance in the highest range (60–84 ind. 0.25m$^{-2}$) (see Table 2).

The exploration areas sampled in our study all lie within the mesotrophic zone, but APEI#3 lies within the oligotrophic zone. An analysis of biogeochemical processes confirmed the very low POC fluxes at APEI#3 ($< 1$ mg C m$^{-2}$ d$^{-1}$) and found respiration rates that were 2-fold lower than in the exploration areas of the mesotrophic zone (Volz et al., 2018). APEI#3 was

also characterized by higher clay content, which may be caused by lower sedimentation rate and a different sedimentation regime (Hauquier et al., 2018; Volz et al., 2018). Polychaete assemblages in APEI#3 consistently showed lower abundance, lower species richness and lower alpha diversity. Species turnover was also very high, with APEI#3 showing the highest rate of species unique to an area and the lowest NNESS for all pairs of comparisons. The redundancy analysis also suggested that, in addition to food supply, the higher relative proportion of clay contributed to variation in species composition at APEI#3.

The polychaete assemblage was dominated by cirratulids, with one species significantly contributing to ordination (*Aphelochaeta* sp. 2062). Some cirratulids are recognized as surface deposit-feeders (Jumars et al., 2015), and may prefer the smaller particles predominantly present at APEI#3 (D$_{4-3}$ = 15.71 μm). At least two cirratulid species can effectively select particle sizes in the clay-size range using their tentacles (Magalhães and Bailey-Brock (2017). The strong shift in community structure and composition of polychaete assemblages between the APEI#3 and the exploration areas echoes that of megafaunal

(Vanreusel et al., 2016), nematode (Hauquier et al., 2018), and tanaid assemblages (Pabis et al., submitted). The biogeochemical settings as well as the biological patterns of the three size groups of the benthic fauna thus converge to conclude that the structure and functioning of the benthic ecosystem in APEI#3 is not representative of any of the four exploration contract areas included in this study.

Within the mesotrophic zone, the species composition of polychaete assemblages in the Ifremer exploration area differed from

the other exploration areas. Differences were driven by species belonging to common deep-sea deposit feeders such as spionids, paraonids, and opheliids (Jumars et al., 2015), whereas a lumbrinerid species characterised the eastern exploration areas (BGR, IOM and GSR). These results agree with Smith et al. (2008b) who observed higher abundances of lumbrinerids and amphinomids, two families of carnivorous polychaetes (Jumars et al., 2015), in the eastern CCFZ (Kaplan East). The upper trophic levels indeed tended to be less represented in the Ifremer and APEI#3 areas than in the eastern areas. This pattern

matches model predictions that food-chain length is positively correlated with resource availability in very low productivity systems ($< 1$–$10$ g C m$^{-2}$ year$^{-1}$) (Moore and de Ruiter, 2000; Post, 2002). McClain and Schlacher (2015) formulated this food-chain length-productivity relationship as the "one-more-trophic-level" hypothesis to account for a positive productivity–diversity relationship. No significant correlation was however found between alpha diversity and productivity, neither at the NE Pacific scale nor at the scale of the whole CCFZ.



To conclude, our study supports the assumptions behind the creation of nine large APEIs, namely that gradients of sea-surface primary productivity determine large-scale patterns, and that nodule densities determine local-scale patterns in community structure, species composition and functioning (Wedding et al., 2013). However, environmental conditions at the APEI#3 seem to be beyond the range of those found in exploration contract areas, which may explain why the community structure

and species composition of benthic assemblages are so different. The fact that the APEI#3 lies mostly north of the Clarion Fracture Zone may however also contribute to its dissimilarity with the areas located in the CCFZ per se.

## 4.2 Species turnover and geographic ranges

Species turnover was best illustrated by the distance-decay of NNESS similarity, which showed two different patterns. Firstly, APEI#3 showed very low similarity with all other areas, irrespective of distance. Secondly, similarity decayed linearly with

distance among the exploration areas located within the CCFZ. Beyond variation in food inputs, as discussed above, the large dissimilarity of polychaete assemblages in APEI#3 may suggest a major physiographic barrier between the north and south of the Clarion Fracture. The Clarion Fracture Zone is a long and narrow submarine mountain range characterized by a peak and through exceeding 1800 m difference in elevation (Hall and Gurnis, 2005), which may be a barrier to dispersal for abyssal fauna. In the Atlantic, the Vema-TRANSIT expedition tested the influence of the Mid-Atlantic Ridge (MAR) and the Vema

Fracture Zone (VFZ) on distribution and connectivity patterns of abyssal fauna with contrasting results (Riehl et al., 2018a). The MAR is not a barrier to dispersal for nematode species of the genus *Acantholaimus* (Lins et al., 2018), a pattern already found for 61 copepod species of the genus *Mesocletodes* (Menzel et al., 2011). However, the MAR is differently permeable to dispersal for three families of isopods, depending on their habits and swimming abilities (Bober et al., 2018). In particular, connectivity was very low for Macrostylidae species, a family of burrowing isopods with limited dispersal abilities (Riehl et

al., 2018b). The species composition of the two polychaete families Spionidae and Polynoidae also differed on both sides of the VFZ, which may be due to limited dispersal and different habitat characteristics (Guggolz et al., 2018). In the CCFZ, Bonifácio and Menot (2019) described 17 new species of polynoids, of which four species are shared between APEI#3 and the exploration areas. In the abyssal Pacific, the CCFZ and the Peru Basin share nine species of scavenging amphipods (Patel et al. (2018)), which thus potentially cross the Clipperton and Galapagos Fracture Zones. However, species identification was

based on morphology only, although cryptic species are common among scavenging amphipods, even in abyssal lineages (Melo, 2004; Havermans et al., 2013). The influence of the fracture zones on the dispersal of the abyssal fauna remains to be better understood as the Clarion and Clipperton fractures may act as a barrier for species with low dispersal abilities such as infaunal brooders. If so, the representativeness of seven out of the nine APEIs, which are partly lying beyond the fractures, may be questionable.

Moreover, the slope of the linear decay of NNESS similarity within the CCFZ suggests an average range of 14 to 25 km per species. This average range masks large variance between a small pool of widespread species, known from two or more areas, and a large pool of rare species, yet only known from one study area and in most cases only known from a single individual. This high frequency of singletons may also significantly bias the estimation of species ranges (see below for a discussion on





singletons). However, based on the best knowledge we have, our study suggests that, on average, the spatial range of polychaete species in the CCFZ is on the order of 20 km. This figure can be compared with the scale of a mining operation (Volkmann et al., 2018; Volkmann and Lehnen, 2018): rounding production rate to 1.5 Mt year$^{-1}$ and with a nodule abundance of 15 kg m$^{-2}$, an area of a 100 km² would be mined each year. In other words, nodule mining would affect each year an area that is equivalent

to the average geographic range of a polychaete species.

## 4.3 How many polychaete species live in the CCFZ? The under-sampling bias

Considering that the economic feasibility of nodule mining requires, for any single operation, mining a minimum of ca. 100 km² of abyssal seafloor per year for a couple of decades (Volkmann et al., 2018; Volkmann and Lehnen, 2018), there is no doubt that the benthic ecosystem will be subjected to adverse environmental impacts and that recovery, if any, will take

centuries (Miljutin et al., 2011; Vanreusel et al., 2016; Gollner et al., 2017; Jones et al., 2017). The main issue that has to be addressed is how significant these adverse impacts will be; will they cause "serious harm" (Levin et al., 2016) and in particular what will be the magnitude of biodiversity loss (Van Dover et al., 2017)? To assess the significance of adverse impacts due to nodule mining, one of the key unknowns is whether the deep sea, including abyssal fauna, is hyper-diverse (Hessler and Jumars, 1974; Grassle and Maciolek, 1992; Paterson et al., 1998) or not (May, 1992; Rex et al., 2005).

Locally, alpha diversity of polychaete assemblages is high in the CCFZ (Paterson et al., 1998; Glover et al., 2002; Wilson, 2017), and it is particularly so for the equitability component of diversity, as exemplified by the slopes of individual-based rarefaction curves and a ratio of individuals to species of two to three at a local scale. At none of the sampling areas does rarefaction curves level off, highlighting that species richness has been systematically under-sampled, even at DOMES A, where 41 box-cores have been sampled (Wilson, 2017). At a regional scale, Glover et al. (2002) reported a total of 177

polychaete species in 2.94 m² along a 3260 km latitudinal gradient of productivity in the NE Pacific and a total of 183 species in 21 m² along a 2800 km longitudinal transect crossing the CCFZ. Janssen et al. (2015) found 233 MOTUs of polychaetes from epibenthic sledge samples of the BGR and Ifremer areas separated by 1400 km. Along this same transect, and using an integrative taxonomy approach, we report here a total of 275 species from 30 quantitative box cores, covering an area of 7.5 m². The two latter studies, relying partly or totally on DNA barcoding, yield higher numbers of species than the two former

regional assessments based on morphology only. Our personal observations during the identification process effectively allowed the identification of cryptic species sometimes sympatrically distributed. This presence of cryptic species has been already observed by Janssen et al. (2015) and Bonifácio and Menot (2019) with the former suggesting that such extreme environmental conditions have already selected for the best morphological characters, resulting in convergent speciation in other aspects as well, such as behavior or physiology. Integrative taxonomy thus not only provides more accurate estimates of

species diversity, but also facilitates comparisons across datasets. Over 90 % of the species in the abyssal Pacific are new to science (Glover et al., 2002) and there are few attempts to try to name them (Paterson et al., 2016; Bonifácio and Menot, 2019), although DNA sequences can easily be matched. Therefore, 26 MOTUs are shared between Janssen et al. (2015) and our study. The overlap is low but it should be noted that we had only 71 COI sequences belonging to 45 MOTUs to compare with the



556 COI sequences belonging to 233 MOTUs from Janssen et al. (2015). This highlights a shortcoming of COI-based barcoding because success rates for COI sequencing are generally low and a combination of several genetic markers plus morphology is essential to accurately assess species diversity. In addition, Janssen et al. (2015) used an epibenthic sledge and we used a box corer. These two devices sample different components of benthic communities. During the SO239 cruise,

epibenthic sledge samples provided a collection of 278 specimens and 80 MOTUs of polynoids, a family of larger epifaunal polychaetes (Bonifácio et al., 2016; Bonifácio and Menot, 2019), but in our box core samples, we only found one polynoid. Overall, the combination of high local diversity, unsaturated rarefaction curves, high levels of cryptic diversity and high rates of species turnover suggest that polychaete diversity in the CCFZ is large and vastly under-sampled. Within the eastern CCFZ, the linear decay of NNESS similarity suggests a species turnover of 0.04 to 0.07 species km$^{-1}$, and decomposition of the beta

diversity shows that 90 % of dissimilarity is due to spatial turnover. This rate of species change is one order of magnitude higher than the rate found by Wilson (2017) for polychaetes (0.0056 species km$^{-1}$) and even higher than the rate for isopods (0.012 species km$^{-1}$). These discrepancies may again reflect a high level of cryptic richness. Wilson (2017) acknowledged that the rates of change he found may be underestimated, particularly for polychaetes, due to the fact that identifications were based on morphology only. The rate of species turnover that we report here for a 1440 km transect across the eastern CCFZ is

however 20 times lower than the rates of 1 species km$^{-1}$ reported by Grassle and Maciolek (1992) from a 180 km transect at 2100 m in the Northwest Atlantic. This difference is roughly consistent with Grassle and Maciolek (1992) hypothesis that in the deepest and most oligotrophic parts of the ocean, species richness may be lower by one order of magnitude. Still, an extrapolation of our rate of species turnover to the 6 million km² of the CCFZ, as Grassle and Maciolek (1992) did for the whole deep sea, yields predictions of at least 240,000 polychaete species, i.e. a number of species equivalent to the number of

accepted marine species globally (WoRMS, 2019).

This prediction is in sharp contrast with the outcome of non-parametric estimators of species richness such as Chao or Jackknife, which provides a maximum estimate of 498 species. Such estimators however implicitly assume that the number of singletons decreases with increasing sample size (Melo, 2004), but the number of singletons steadily increased with sample size in this study. In such circumstances, the non-parametric species estimators underestimates species richness (Melo, 2004;

Coddington et al., 2009). In an intensive survey of spiders in 1 ha of tropical forest, Coddington et al. (2009) found 29 % of singletons and tested the null hypothesis of under-sampling against ecologically driven hypotheses to explain this "anomalously" high frequency of singletons. They concluded that under-sampling was the most parsimonious explanation for the high frequency of singletons and that it causes a systematic negative bias of species richness estimators. In the deep-sea, an anomalously high rate of singletons of about one-third of the species is in fact the rule of macrofaunal surveys (Gage, 2004)

and the most parsimonious hypothesis that still needs to be tested thus is that the deep-sea macrofauna has been systematically under-sampled.

Although under-sampling causes an underestimation of species richness, it may also lead to an overestimation of the distance-decay of similarity, because singletons, considered as endemic to an area in the analysis, may have much wider distributions. In conclusion, our level of certainty on the number of polychaete species inhabiting the CCFZ and potentially threatened by





nodule mining ranges from 498 to 240,000 species. The former estimate assumes that we have already sampled about half of the regional diversity and further suggests that most species have a large geographical range. The latter estimate assumes that we have sampled 0.1 % of the polychaete species in the CCFZ and that these species have narrow geographical ranges about the size of a yearly mined area.

**5 Conclusions**

As part of the JPI Oceans project "Ecological aspects of mining impact", four nodule exploration contract areas and one APEI were sampled across the eastern half of the CCFZ to characterize the benthic communities found there. The main objectives of the present study were to describe and identify alpha and beta diversity patterns of polychaete assemblages with the aim of increasing basic knowledge on the ecology of these abyssal communities, testing the hypotheses that supported spatial
conservation planning in the CCFZ, assessing the representativeness of an APEI and improving the assessment of potential risk to biodiversity due to nodule mining.

In the abyssal NE Pacific, patterns of polychaete abundance follow the northward and westward gradients of decreasing sea-surface primary productivity. The increasingly oligotrophic conditions cause a shift in the trophic structure and species composition of polychaete assemblages that is consistent with the assumption that led to the creation of nine APEIs in the
CCFZ. The most significant shift in community structure and composition was however found between the APEI#3 and the nodule exploration areas. APEI#3 is found in oligotrophic conditions, north of the Clarion Fracture Zone, whereas exploration areas experience mesotrophic conditions south of the Clarion Fracture Zone. The scantiness of food supply and a barrier to dispersal may compromise the representativeness of APEI#3 and thus question its ability to meet its purpose of preserving the biodiversity of the CCFZ fauna.
Within the CCFZ per se, the diversity of polychaete assemblages is even higher than previously thought due to a high level of cryptic richness. Species turnover is high with a minimum estimated rate of species change of 0.04 species km$^{-1}$, suggesting an average geographical range of 25 km and a number of polychaete species in the CCFZ that may equal the number of all currently known marine species. If true, the risk of species extinction is very high because the footprint of nodule mining would largely exceed the range of many species. On the contrary, non-parametric estimators of species richness suggest that
total species richness across the five study areas does not exceed 498 species. Both methods of estimating species richness can however be severely biased by singletons. Singletons represent 49 % of the 275 species of polychaetes that were sampled. The most parsimonious hypothesis to explain such a high rate of singletons is under-sampling. The assessment of potential risks and scales of biodiversity loss thus requires an appropriate inventory of species richness in the CCFZ.





**Data availability**

Abundance data analyzed in the present study are available in the Pangaea (Bonifacio et al., 2019) whereas DNA sequences are available in BOLD or GenBank databases.

**Author contribution**

LM and PMA conceived the project and designed the sampling. LM and PB performed the sampling and processed the samples. PB identified (morphology and DNA) the polychaetes. LM and PB analyzed and interpreted the data. All authors prepared and contributed to the manuscript.

**Competing interests**

The authors declare that they have no conflict of interest.

**Acknowledgement**

The research leading to these results has received funding from the Ifremer program "*Ressources Minérales Marines*" (REMIMA), the JPI Oceans pilot action "Ecological Aspects of Deep-Sea Mining" and the European Union Seventh Framework Program (FP7/2007–2013) under the MIDAS project, grant agreement no. 603418. We are grateful to the crew of the RV *Sonne*, and all people involved in the field sampling and sample processing during the SO239 cruise. We would like
to thank for their expertise in washing and sieving samples to Dr. Stefanie Kaiser, Sarah Schnurr and Dr. Ana Hilário; and for live-sorting the worms to Lenka Neal. Also, thanks to Baptiste François for sample sorting in the laboratory. We are grateful, as well, to all people involved in molecular analysis: Dr. Aliou Dia, Guillaume Lannuzel, Emmanuelle Omnes, Alana Jute, Mohamed Dosoky and Gavin Campbell. Special thanks to Dr. Thomas Dahlgren for taking care and sharing data of some families of polychaetes; to Prof. Dr. Ann Vanreusel, Dr. Freija Hauquier and Dr. Felix Janssen for providing the abiotic data
of CCFZ.

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



## Tables

**Table 1. Area, locality, station, date, depth, geographical position and nodule density of all 34 box corer deployments across the CCFZ during the SO239 cruise. "*" indicates box cores considered as non-quantitative, not included in the analyses.**

| Area | Locality | Station | Date | Depth (m) | Latitude | Longitude | Nodule density (kg m$^{-2}$) |
|---|---|---|---|---|---|---|---|
| **BGR** | BGR-PA | 12 | 20/03/15 | 4118 | 11.8471667 | -117.05933 | 26.40 |
| **BGR** | BGR-PA | 15 | 21/03/15 | 4133 | 11.8443333 | -117.05217 | 26.80 |
| **BGR** | BGR-PA | 16 | 21/03/15 | 4122 | 11.8573333 | -117.052 | 24.00 |
| **BGR** | BGR-PA | 21 | 22/03/15 | 4120 | 11.8535 | -117.0595 | 22.80 |
| **BGR** | BGR-PA | 23 | 22/03/15 | 4122 | 11.85 | -117.05267 | 20.80 |
| **BGR*** | BGR-RA | 51 | 27/03/15 | 4348 | 11.8236667 | -117.52367 | 0.00 |
| **BGR** | BGR-RA | 57 | 28/03/15 | 4370 | 11.8075 | -117.52433 | 8.00 |
| **BGR** | BGR-RA | 58 | 28/03/15 | 4350 | 11.8205 | -117.54167 | 1.60 |
| **BGR** | BGR-RA | 60 | 29/03/15 | 4325 | 11.8076667 | -117.55033 | 18.00 |
| **IOM** | IOM-control | 88 | 02/04/15 | 4433 | 11.079 | -119.65883 | 0.00 |
| **IOM** | IOM-control | 89 | 02/04/15 | 4437 | 11.0758333 | -119.66083 | 1.20 |
| **IOM** | IOM-control | 90 | 03/04/15 | 4434 | 11.074 | -119.66417 | 0.00 |
| **IOM** | IOM-disturb | 94 | 03/04/15 | 4414 | 11.0736667 | -119.6555 | 0.40 |
| **IOM** | IOM-disturb | 95 | 03/04/15 | 4418 | 11.0735 | -119.65583 | 0.80 |
| **IOM** | IOM-disturb | 97 | 04/04/15 | 4421 | 11.0728333 | -119.65617 | 0.20 |
| **IOM*** | IOM-resed | 105 | 05/04/15 | 4423 | 11.0711667 | -119.65533 | 0.00 |
| **IOM** | IOM-resed | 106 | 05/04/15 | 4425 | 11.0716667 | -119.65483 | 0.20 |
| **IOM** | IOM-resed | 107 | 05/04/15 | 4425 | 11.0721667 | -119.6545 | 0.30 |
| **GSR** | GSR | 119 | 08/04/15 | 4516 | 13.8591667 | -123.25267 | 26.47 |
| **GSR** | GSR | 127 | 09/04/15 | 4514 | 13.8443333 | -123.246 | 27.10 |
| **GSR** | GSR | 128 | 09/04/15 | 4511 | 13.8516667 | -123.252 | 27.10 |
| **GSR** | GSR | 137 | 11/04/15 | 4510 | 13.856 | -123.238 | 25.20 |
| **GSR** | GSR | 138 | 11/04/15 | 4503 | 13.8481667 | -123.23467 | 26.47 |
| **Ifremer** | Ifremer | 159 | 15/04/15 | 4921 | 14.049 | -130.13433 | 19.80 |
| **Ifremer** | Ifremer | 162 | 16/04/15 | 4951 | 14.049 | -130.126 | 20.20 |
| **Ifremer** | Ifremer | 169 | 17/04/15 | 4964 | 14.0421667 | -130.12733 | 24.10 |
| **Ifremer** | Ifremer | 180 | 18/04/15 | 4936 | 14.0416667 | -130.13633 | 16.00 |
| **Ifremer** | Ifremer | 181 | 18/04/15 | 4896 | 14.0465 | -130.1415 | 16.80 |
| **Ifremer** | Ifremer | 182 | 18/04/15 | 4957 | 14.0423333 | -130.1275 | 22.40 |
| **APEI#3** | APEI#3 | 195 | 21/04/15 | 4833 | 18.7958333 | -128.36217 | 6.28 |
| **APEI#3** | APEI#3 | 196 | 21/04/15 | 4847 | 18.7971667 | -128.34617 | 1.80 |
| **APEI#3*** | APEI#3 | 203 | 23/04/15 | 4843 | 18.774 | -128.35317 | 2.88 |





| | | | | | | | |
|---|---|---|---|---|---|---|---|
| **APEI#3** | APEI#3 | 204 | 23/04/15 | 4816 | 18.7733333 | -128.33617 | 3.65 |
| **APEI#3*** | APEI#3 | 209 | 24/04/15 | 4819 | 18.7845 | -128.3725 | 3.65 |



**Table 2. Data available from previous studies and from the present study included in regional-scale analyses.**

| Area | Year | References | Depth (m) | Latitude | Longitude | Number of box cores | Mean abundance (ind. 0.25 m$^{-2}$) | Total number of species | ES163 | Bootstrap | 2002–2018 average POC at the seafloor (g C m$^{-2}$ year$^{-1}$) |
|---|---|---|---|---|---|---|---|---|---|---|---|
| DOMES A | 1977/78 | Glover et al. (2002); Wilson (2017) | 5100 | 8.45 | -150.78333 | 47 | 16 | 104 | 56 | 203 (based on 41 box cores) | 1.46 |
| Kaplan West | 2004 | Smith et al. (2008b) | 5000 | 9.55195 | -150.00845 | | 5 | | | | 1.50 |
| Kaplan Central | 2004 | Smith et al. (2008b) | 5000 | 14.0710333 | -130.109 | | 21 | | | | 1.88 |
| Ifremer | 2015 | Present study | 4937 | 14.049 | -130.13433 | 6 | 28 | 73 | | 91 | 1.90 |
| PRA | 1989 | Glover et al. (2002); Wilson (2017) | 4800 | 12.95 | -128.31667 | 16 | 65 | 100 | 47 | 310 | 2.04 |
| ECHO 1 | 1982 | Glover et al. (2002); Wilson (2017) | 4500 | 14.6666667 | -126.41667 | 15 | 42 | 113 | 60 | 274 (based on 14 box cores) | 2.05 |
| B4S03 | 2015 | De Smet et al. (2017) | 4500 | 14.1124806 | -125.87147 | 4 | 20 | 12 | | | 2.11 |
| B4N01 | 2015 | De Smet et al. (2017) | 4500 | 14.7064111 | -125.46118 | 5 | 14 | 10 | | | 2.04 |
| B6S02 | 2015 | De Smet et al. (2017) | 4500 | 13.8940389 | -123.29704 | 3 | 21 | 14 | | | 2.11 |
| GSR | 2015 | Present study | 4510 | 13.8443333 | -123.246 | 5 | 59 | 104 | 79 | 126 | 2.11 |
| IOM | 2015 | Present study | 4425 | 11.0758333 | -119.66083 | 8 | 37 | 107 | 79 | 131 | 2.07 |
| Kaplan East | 2003 | Smith et al. (2008b) | 4000 | 14.9308333 | -119.0495 | | 21 | | | | 2.27 |
| BGR | 2015 | Present study | 4206 | 11.8205 | -117.54167 | 8 | 58 | 156 | 88 | 192 | 2.23 |
| APEI#3 | 2015 | Present study | 4832 | 18.7845 | -128.3725 | 3 | 5 | 9 | | 11 | 1.66 |
| CLIMAX II | 1969 | Paterson et al. (1998) | 5010 | 28 | -155 | 10 | 16 | 46 | | | 1.91 |
| EqPac 0 | 1992 | Glover et al. (2002) | 4300 | 0 | -140 | 3 | 84 | 73 | 71 | | 3.55 |
| EqPac 2 | 1992 | Glover et al. (2002) | 4400 | 2 | -140 | 4 | 60 | 82 | 82 | | 2.68 |
| EqPac 5 | 1992 | Glover et al. (2002) | 4400 | 5 | -140 | 3 | 80 | 76 | 75 | | 2.33 |
| EqPac 9 | 1992 | Glover et al. (2002) | 4900 | 9 | -140 | 3 | 13 | 23 | | | 1.78 |





**Table 3. Observed species richness (Sobs) and estimators of species richness for each sampled area and for eastern CCFZ.**

| Area | Sobs | Individual-based | | | | Sample-based | | | | | |
|---|---|---|---|---|---|---|---|---|---|---|---|
| | | n | Chao 1 | ACE | ES12 | n | Chao 2 | Jackknife 1st order | Jackknife 2nd order | Bootstrap | S3 |
| **BGR** | 156 | 415 | 355 ± 61 | 334 ± 11 | 11 ± 1 | 8 | 311 ± 46 | 240 ± 34 | 295 | 192 ± 15 | 82 ± 9 |
| **IOM** | 107 | 274 | 191 ± 30 | 225 ± 10 | 11 ± 1 | 8 | 182 ± 26 | 162 ± 22 | 195 | 131 ± 10 | 58 ± 6 |
| **GSR** | 104 | 263 | 157 ± 19 | 196 ± 9 | 11 ± 1 | 5 | 161 ± 20 | 153 ± 26 | 178 | 126 ± 12 | 77 ± 6 |
| **Ifremer** | 73 | 154 | 163 ± 38 | 181 ± 8 | 11 ± 1 | 6 | 160 ± 36 | 115 ± 19 | 142 | 91 ± 9 | 45 ± 5 |
| **APEI#3** | 9 | 12 | 20 ± 10 | 27 ± 2 | 9 ± 0 | 3 | 30 ± 27 | 14 ± 4 | 17 | 11 ± 2 | 9 ± 0 |
| **CCFZ** | 275 | 1118 | 450 ± 41 | 484 ± 13 | 11 ± 1 | 30 | 467 ± 44 | 411 ± 29 | 498 | 334 ± 14 | 66 ± 13 |





**Figures**

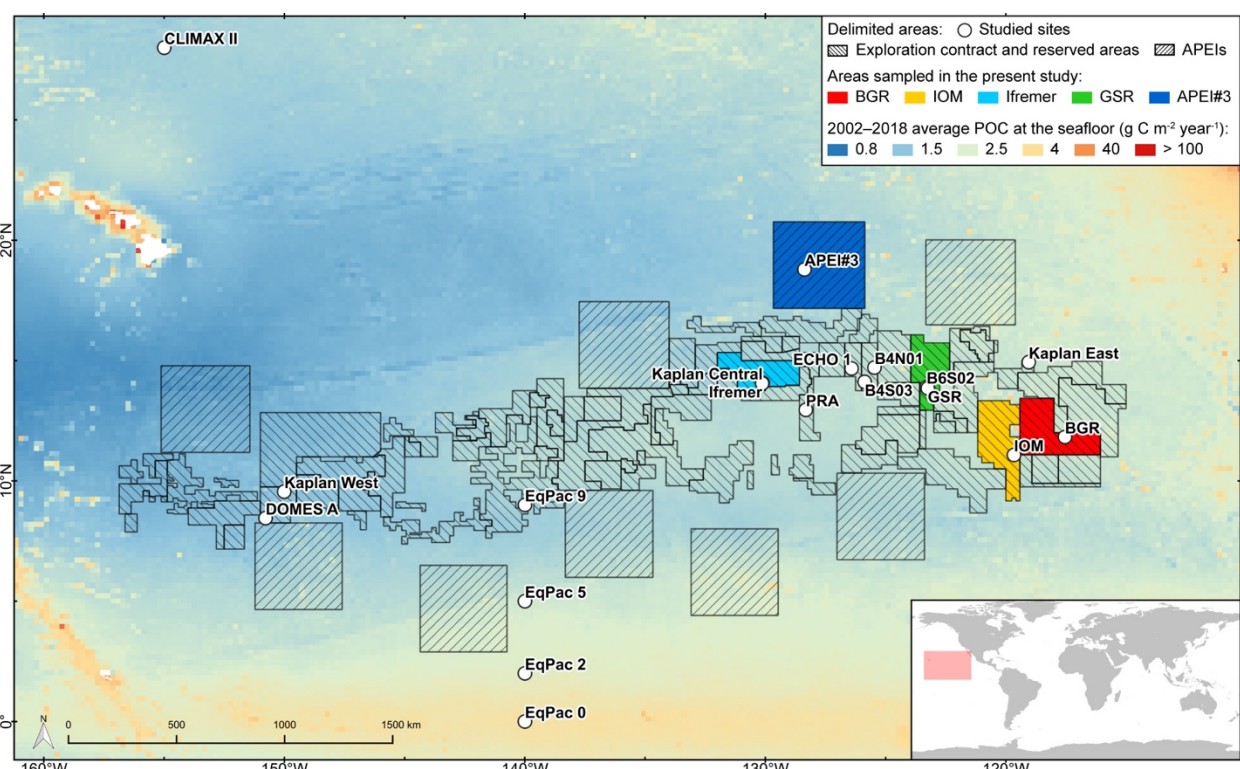

**Figure 1. Map of the nodule exploration contract, reserved areas and areas of particular interest (APEI) in the Clarion-Clipperton Fracture Zone (CCFZ) showing the sampling areas from this study and previous macrobenthic surveys. The areas sampled during the SO239 cruises are shown in color. The background map shows average particulate organic carbon (POC) flux at the seafloor during the 2002–2018 period.**





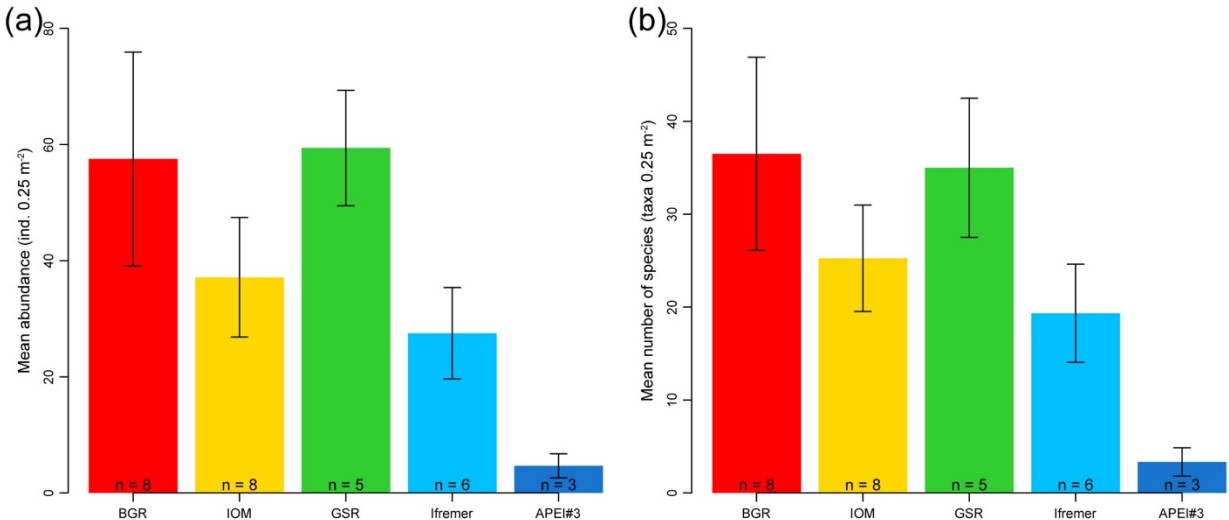

**Figure 2. Bar plots of mean abundance per box core (a) and mean species density per box core (b) of polychaete assemblages for each sampled area within the eastern CCFZ. "n" indicates the number of box cores samples; error bars show the standard deviation.**





**Figure 3.** Correlation matrix between biotic and abiotic variables from sampled areas within the eastern CCFZ. Diagonal panels show the distribution frequency of values for each variable. Below-the-diagonal panels show the correlation plot between pairs of variables. Above-the-diagonal panels show the Spearman coefficient correlations between pairs of variables. "*" indicates p < 0.05, "**" p < 0.01 and "***" p < 0.001.





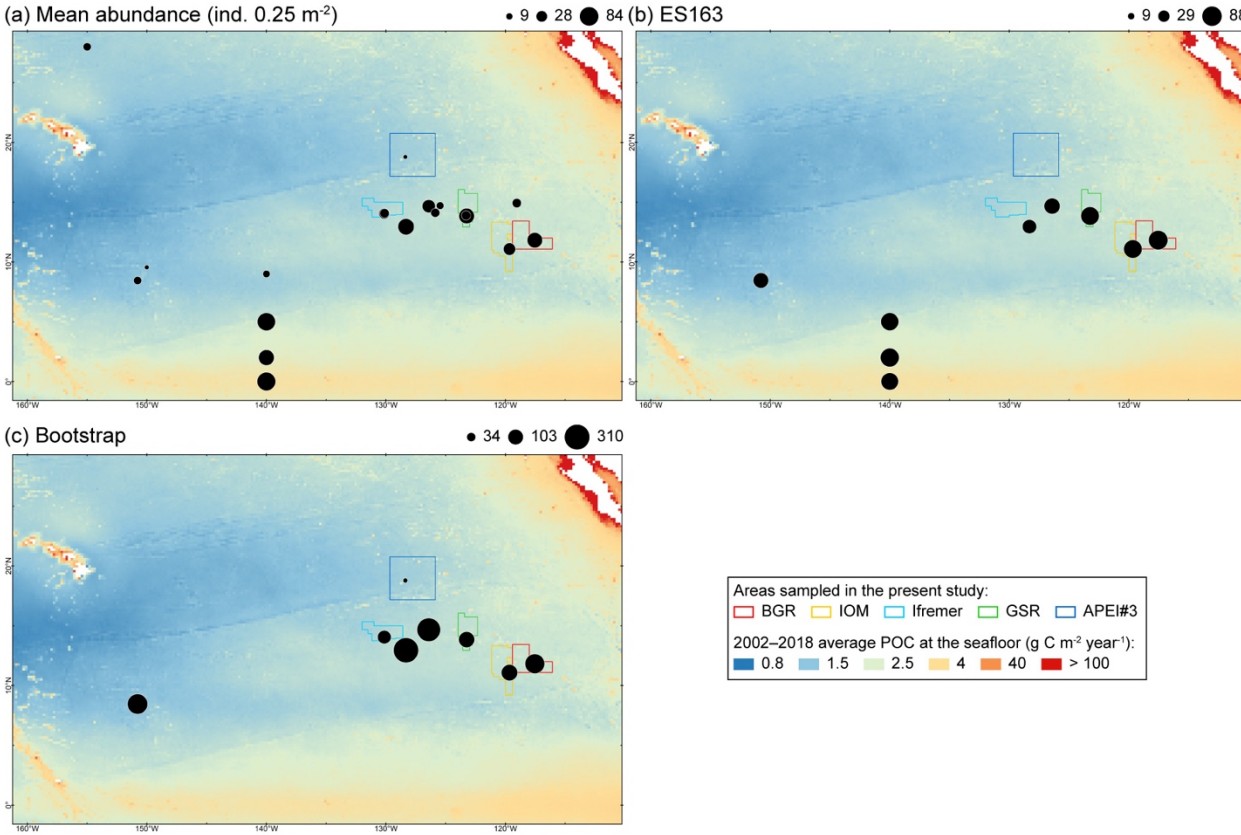

**Figure 4. Map of mean abundance (a) and diversity estimators, ES163 (b) and bootstrap (c), from the Northeast Pacific in relation to the 2002–2018 average particulate organic carbon (POC) concentration at the seafloor along the CCFZ. The background map shows average POC flux at the seafloor during the 2002–2018 period.**







**Figure 5. Bar plots of the relative abundance of families (a) and trophic guilds (b) for each sampled area within the eastern CCFZ. Gradient color in (a) corresponds to the different guilds in (b).**


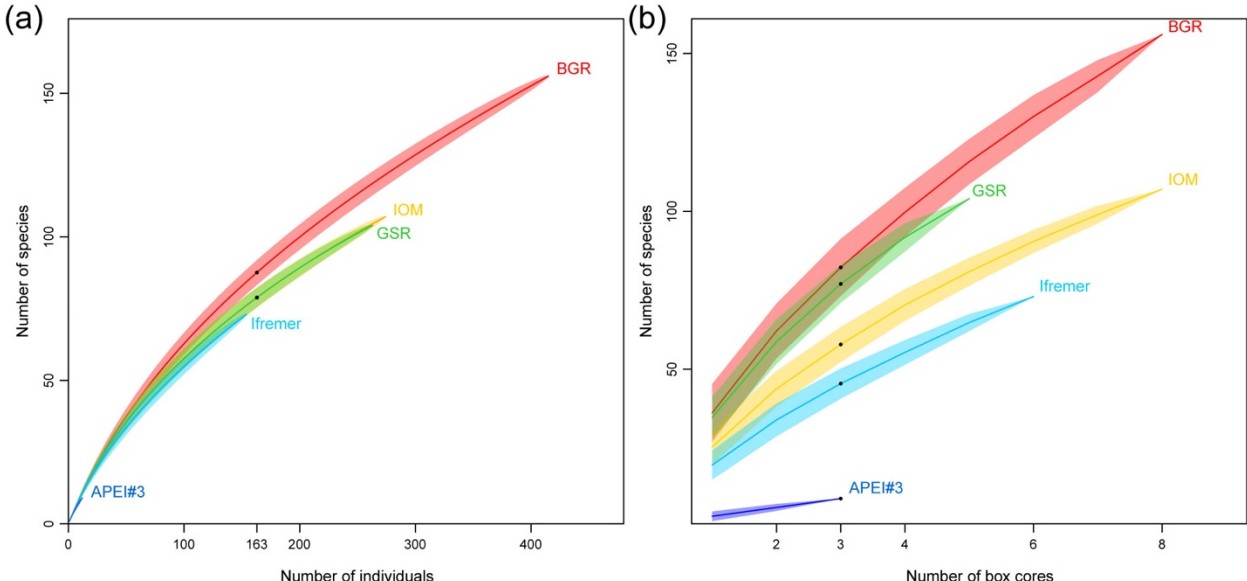

**Figure 6. Individual-based rarefaction curves (a) and sample-based accumulation curves (b) for each sampled area within the eastern CCFZ.**





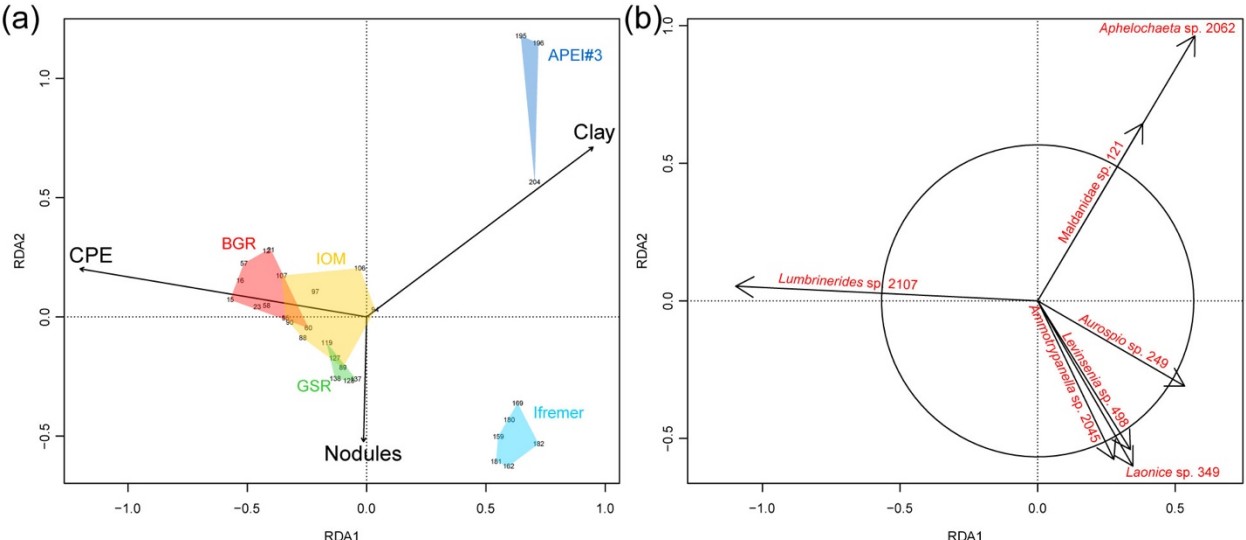

**Figure 7. Redundancy analysis (RDA) biplot based on the chord-normalized expected species shared (CNESS) distance constrained by the selected variables (a, scaling 2) and showing species significantly contributing to the ordination diagram (b, scaling 1).**





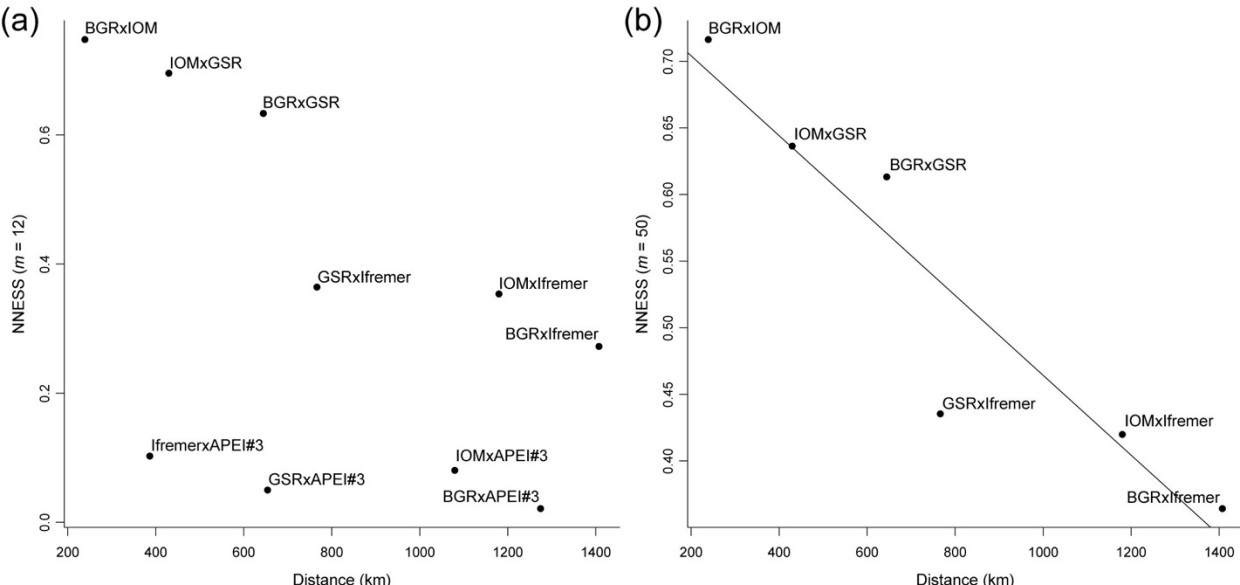

**Figure 8. Distance-decay of new normalized expected species shared (NNESS) between BGR, IOM, GSR, Ifremer and APEI#3 using**
*m* **= 12 (a); and between BGR, IOM, GSR and Ifremer using** *m* **= 50, regression with** *y* **intercept: 0.7642, slope: -0.0002999 (b).**





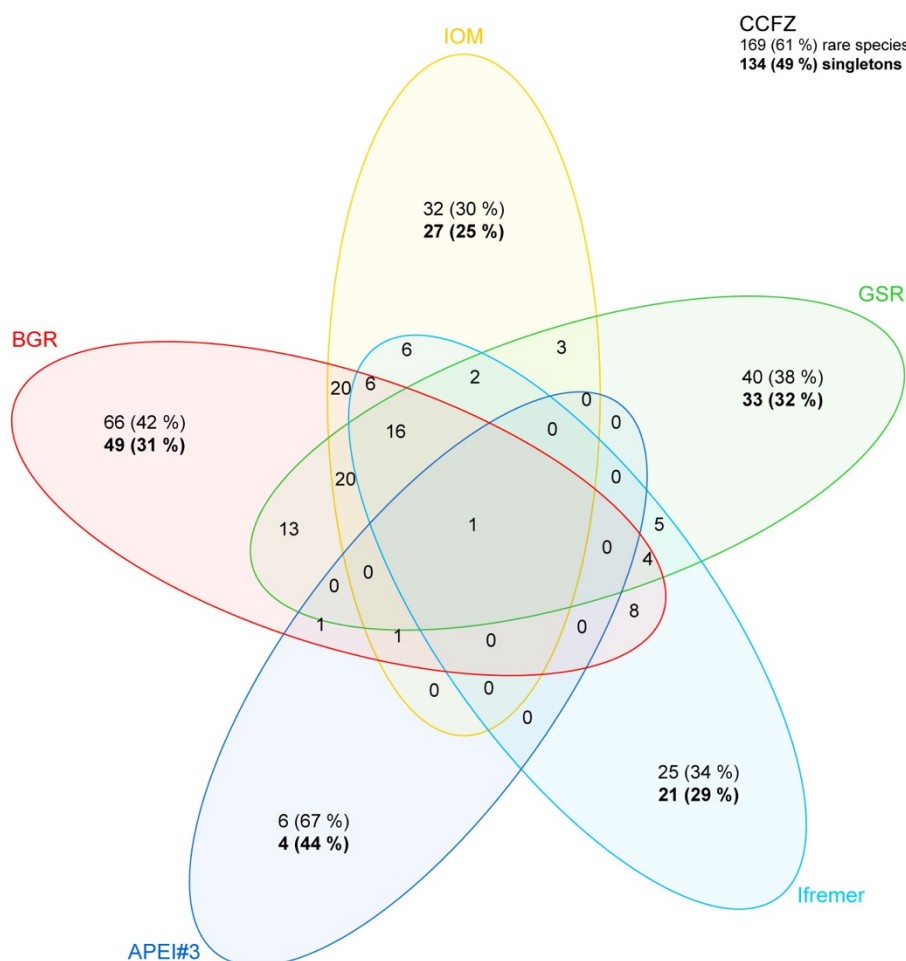

**Figure 9. Venn diagram with the records of rare (being recorded in only one area, with corresponding percentage) and common species among sampled areas and for the eastern CCFZ. Bold values indicate the number of species with a single specimen (singletons, with corresponding percentage).**





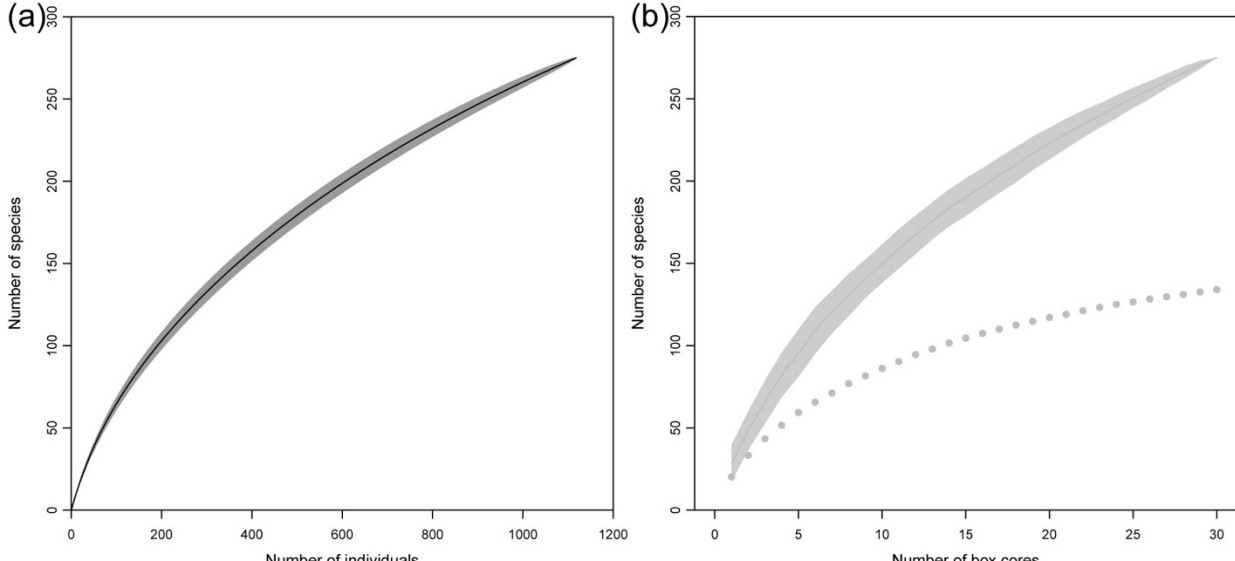

**Figure 10. Individual-based rarefaction curve (a) and sample-based accumulation curves (b) within the eastern CCFZ. The dotted curve shows the sample-based accumulation of singletons.**