# Peer review of "Alpha and beta diversity patterns of polychaete assemblages across the nodule province of the eastern Clarion-Clipperton Fracture Zone (Equatorial Pacific)"

_Biogeosciences, 2019_

## Referee Comment (RC1) · Paul Dando (Referee) · 7 Aug 2019

The paper presents the data on the diversity patterns in polychaetes collected on a cruise that was "aimed at improving species inventories, determining species ranges --" in the polymetallic nodule area of the Clarion-Clipperton Fracture Zone (CCFZ).

Given the importance of the results, for areas that may be subject to disturbance by nodule mining, I do not understand why there is no reference to the distribution of the rest of the major macro-infaunal groups, given that Wilson (2017) showed that different groups respond in different ways. It is important to know whether similar distribution patterns to those found for polychaetes occurred in the other groups in the studied

area.

It is clear that a lot of effort has gone into obtaining and working up this data. It is therefore unfortunate that a poor description of what was done and the reasons for this make it difficult to understand what fraction of the infauna was sampled and the consequences for the overall polychaete biodiversity and the comparisons between different box cores.

Given that the topography of the polymetallic nodules affects local water flow patterns and creates different microhabitats (Mullineaux, 1989). The cruise report gives information on the nodule differences between individual cores but this information does not appear to have been used in analyzing polychaete numbers and distributions, although this should have a major influence on species composition and numbers. I would like to see an analysis of polychaete species distribution and numbers with respect to the differences in nodule topography, numbers and sizes between cores. Treating all box core samples within an area as replicates does not appear to be valid. Similarly no use has been made of the data on species distribution with respect to sediment depth.

There should be an enlargement of the sampled area in Fig. 1 to allow for the individual core stations to be plotted. A section on the differences between samples taken within an individual area would be beneficial, since a lot of information on the ecology has been lost by only comparing means for each area.

Sampling strategy

The sampling strategies need to be clearly explained with reference to the following questions, either by amplification of the text or by giving a references.

1. Since there are 9 APEIs, why was the only one sampled from an oligotrophic area when the exploration blocks sampled were all in mesotrophic areas?

2. Within each block how was it decided where to sample, given the problem of determining the geographic range of species? [see Wilson (2017) for one approach to

this.]

3. Given the known high percentage of species represented by only a single individual in box core samples from red clay polymetallic nodule areas (Hessler & Jumars, 1974 and several later papers), why were such a small number of box cores taken in each area, as opposed to taking the same overall number of box core samples from fewer areas.

4. Why was it decided (apparently post-sampling) to treat every box core sample from a single area as a replicate, given the geochemical differences within some of the areas and also differences in the number, size and depth distribution of the polymetallic nodules in box cores from the same area?

5. Each box core was sliced into 0-3, 3-5 and 5-10 cm depth sections that were sieved separately. Why was this done when the data from each layer were then added for the data analysis? The slicing procedure is not described but when slicing box cores polychaetes are frequently fragmented. What precautions were taken that an individual was not counted more than once, for example by only counting head-ends.

6. The banked data shows that, although most individuals were found in the 0-3 cm layer, in some cores over 20% of the individuals were present in the 5-10 cm layer. It is therefore reasonable to assume that an unknown fraction of the biodiversity was lost in the samples because the cores were not sampled for macrofauna deeper than 10 cm. In some deep-sea sediments (at > 2000 m) polychaetes are known to penetrate over 100 cm below the sediment surface and the major infaunal biomass can often be found below 10 cm sediment depth. Given that some box cores were sampled below 10 cm depth, since nodules were recorded at 25 cm depth, why is no mention made of animals being present below 10 cm? Can the authors cite any reference to deeper sampling for infauna in the working areas? It might have been assumed that only the upper 10 cm of sediment would be disturbed by nodule harvesting, however most nodule-mining prototypes have been based on bottom crawlers that would cause

sediment compression and affect deeper-burrowing organisms.

7. Surface polymetallic nodules do penetrate the sediment to a degree. We are not told now deep this was in the different cores. Even if it was only 1 cm, given the large number and size of the nodules in some cores (see photographs in the cruise report) this would greatly decrease the volume of sediment available for sieving in the 0-3 cm layer and bias the results, considering that all cores were equated only on an area. The fauna results should also be considered with respect to sediment volume.

8. It is unclear what happened to animals collected when the nodules were washed free of sediment. Were these animals added to the 0-3 cm layer and if so was this done before picking animals off the nodules?

9. What happened to the animals picked off the nodules, were these treated as epifauna and not considered here? Some serpulids are included in the species listed in the dataset – were these all epifaunal on the nodules?

10. Polychaetes are known to occur as infauna within the polymetallic nodules (Thiel et. al. 1993). Some polychaete species were only found in crevices within the nodules and knowledge of differences in species composition in nodules from the different exploration blocks would be important information. Was this part of the infauna sampled? Will there be a separate publication dealing with the nodule-associated fauna, since the present manuscript does not cover the habitat of the species within the core samples?

11. Both the biomass and the biovolume of the infauna can affect the geochemistry, were these measured?

Specific Comments

Table 2: the data should be checked and the header clarified. Does the data refer to only polychaetes or all macroinfauna? For example, Table 7 in Wilson (2017) gives a mean polychaete density of 21 individuals 0.25 m-2 for Domes A compared with the 16 given in the present Table 2. It is also necessary to know if like is being compared with

like with respect to combined fractions of samples from the box cores. Since most of the previous studies did not look for cryptic species I think you should give the number of morphologically identified species in parenthesis for the sites recorded in the present study.

Figure 3 needs a lot more explanation and labelling. Below the diagonal some of the plots appear to use mean data from areas and others data from individual cores without a clear explanation.

In Figures 7 and 9 yellow text does not show well on a white background – use a coloured background for the text.

Conclusions: The first paragraph does not belong here – it should be in the introduction.

Supplementary data: It would be useful to have a species list available as a supplement to the paper, although I realise that the taxonomic studies are ongoing.

References

Hessler, R. R. & Jumars, P. A. (1974) Abyssal community analysis from replicate box cores in the central North Pacific, Deep SeaRes., 21: 185-209

Mullineaux, L. (1989) Vertical distributions of the epifauna on manganese nodules: implications for settlement and feeding. Limnol. Oceanog., 32: 1247-1262

Thiel, H. et. Al. (1993). Manganese nodule crevice fauna. Deep Sea Res. Part I 40: 419–423

Wilson, G. D. F.:(2017) Macrofauna abundance, species diversity and turnover at three sites in the Clipperton-Clarion Fracture Zone,. Mar. Biodivers., 47: 323-347

---

## Referee Comment (RC2) · Anonymous Referee #2 · 13 Aug 2019

The paper provides results of box corer-based study of polychaete macrofauna in the eastern part of the CCZ that significantly contributes to our knowledge on faunal communities in the CCZ. The study's title is appropriate but may raise expectations that the study covers the entire CCZ (see below). The work addresses relevant scientific questions of strong societal significance – and contributes knowledge that can help an informed societal decision regarding if and how to do nodule mining. The study contributes to our understanding of patterns benthic assemblages in the deep sea and how they relate to environmental drivers. At the same time it raises important questions in the context of a potential exploitation of nodules, e.g., regarding the appropriateness of the concept and location of the APEIs and the underlying regional management plan

[Figure]

– and about future effort necessary to answer the questions that have to be answered before mining commences.

The scientific methods are valid and up-to-date and the experiments and calculations are adequately described. The authors acknowledge related work and clearly discriminate their data from data obtained by others. In some cases, where they discuss their data (e.g., regarding winners and loosers of the 'competition' of meio- and macrofauna) in relation to that of others they should be more specific about the exact content of the cited data so the reader can better comprehend the authors' discussions and conclusions. The data presented are sufficient to reach conclusions part of which – and that is not criticizing the ambitions work behind this study – are pointing out gaps in knowledge. The paper is well structured and reads mostly fine – in some parts of the results the language may seem a bit repetitive (abundances and species numbers are presented with almost the same wording). The abstract provides a good summary. In the way it is currently presented, the first part of the discussion (about the relative success of meio an macrofauna in different deep-sea environments) should be either significantly reduced or more specific with a better and quantitative presentation of data on Meiofauna from the study area from other studies. I would vote for a reduction of that part as it seems to be a bit off the main focus anyway.

To me, the main shortcoming of the paper is, that the authors are very carefully when stating their conclusions and overcautious if it's about the consequences of their findings. As the work touches societal concerns and areas of strong debates the authors should address in more depth the implication of their work regarding procedures and management of nodule mining, e.g., can we decide on regional management based on the available knowledge and are the APEIs appropriate as they are? Are there any specific recommendations that can be provided, e.g., regarding the size or arrangement of mining patches? What should be the focus of future studies and what would be the expected effort needed to come to scientifically sound conclusions? The conclusions part is so far rather summarizing what has been stated already before and may be a

good place to discuss these things. The fact, that those discussions are rather limited in the current version of the manuscript is the reason for my general proposition that the manuscript should undergo a major revision before publication.

Below some more detailed comments

MAJOR ISSUES – IMPLICATIONS FOR MINING / MINING REGULATION

Regarding the main shortcoming of the paper mentioned above I urge the authors to significantly extend the discussion of their results towards the implications of their work with regard to nodule mining and its regulation. This could be distributed in several parts of the discussion as well as in a separate section in the discussion or in the conclusions. This, of course, has to be done with some caution to not extend beyond the scope of the study and has to take into account that this is a scientific publication and not a policy paper. However, it is clear that the motivation of the study – and certainly of societies providing the funding for mining-related investigations these days – is to provide the basis for scientifically sound procedures and decisions regarding deep-sea mineral exploration and exploitation. This should be better reflected in the text. This includes recommendations regarding the management and regulations - where the data of the study allow this - but also specific requests for future investigations where the results reveal significant gaps. Below I am providing some examples where I think the discussion needs to move beyond where it currently terminates.

Page 3, line 15/16 'The distribution of APEIs at the periphery of the CCFZ thus deviates from an optimal design.' Page 11, line 20-23 'The biogeochemical settings as well as the biological patterns of the three size groups of the benthic fauna thus converge to conclude that the structure and functioning of the benthic ecosystem in APEI#3 is not representative of any of the four exploration contract areas included in this study.' Page 12, line 26-29 'The influence of the fracture zones on the dispersal of the abyssal fauna remains to be better understood as the Clarion and Clipperton fractures may act as a barrier for species with low dispersal abilities such as infaunal brooders. If so, the

representativeness of seven out of the nine APEIs, which are partly lying beyond the fractures, may be questionable.' If these statements hold true, the concept of APEIs and the regional management plan as a whole don't seem to be appropriate. What is the advice of the authors to overcome this problem? What do we know about the other APEIs and how their environmental conditions and faunal communities compare to license areas? What would be an optimal APEI layout and how would you – from the results of your study – address the question whether an area is suited as APEI or not. Can we use some easily measured sedimentological (grainsize?) or biogeochemical measurement to assess the probability that an APEI will host similar faunal communities than a specific license area? Should the assessment of correlations of habitat characteristics and fauna in APEIs be a focus of future studies? If you consider the lack of knowledge potentially only a few years before exploitation commences: Should the ISA setup a scheme by which contractors carry out or fund baseline studies in the APEIs? One consideration that lead to the APEIs' current position outside the area covered by license areas was to allow for very large areas. In light of the fact that, according to the current planning, only part of the license areas will be used for nodule extraction and the seemingly low species' ranges: do we need APEIs to conserve biodiversity or would the areas inside the patch of license areas, that are not mined do the job? Or do we need APEIs somewhere else, e.g., smaller ones between license areas?

Page 13, line 1-5 'However, based on the best knowledge we have, our study suggests that [. . .] nodule mining would affect each year an area that is equivalent to the average geographic range of a polychaete species.' Spatial ranges – especially if they are indeed that small – are highly relevant. Can we use polychaetes as key species here or would we need to have similar data also for other size classes and other groups of macrofauna? What data are available already? What are the implications of theses results for mining operations and their regulation? Do we need more research to understand whether the estimated spatial range is really true or just mirrors the inappropriate sampling effort available scientific knowledge is based upon? Or do we

'know enough' and could provide specific suggestions as for how to spatially arrange mined patches? Taking this further: if we take the precautionary approach seriously: if we have an average species range of 20km (and, for some of the species obviously a smaller one) wouldn't we need to restrict the mining operations by contractors including secondary impacts by the plume to that size until it is proven, that the high turnover of beta diversity is an artifact of undersampling?

Page 15, line 27/28 'The assessment of potential risks and scales of biodiversity loss thus requires an appropriate inventory of species richness in the CCFZ.' While the conclusions are basically a summary up to this point, this is where the discussion in implications starts: How should this goal should be achieved? How much of this work is, according to the knowledge of the authors, already done by baseline work of the different contractors and just needs metaanalysis of the pooled contractor's data (e.g., by an independent scientific consortium)? Or are samples and data lacking and more dedicated sampling campaigns needed? What effort would this take? Or - if that is too hard to estimate, how would you control if enough data are available (based on rarefaction curves? Based on biodiversity descriptors merging?)? If you compare this to what is found in the ISA regulations and guidelines: how does that compare? What about the key-species concept? Could that become appropriated once the necessary knowledge was obtained or do you think we always have to cope with the full complexity when we want to address environmental impacts of deep-sea operations (assessment of the risks prior to operations, assessment of impacts happening during operations).

OTHER MAJOR ISSUES

Page line 3-27 The discussion of meiofauna in nodule areas comes as a bit of a surprise in the context of this paper, that does only provide data on macrofauna. If you want to leave this so prominent and detailed, you should first state what the data show. Does the Pape et al. study provides data from the same station so a quantitative comparison to other deep sea areas is possible? > In this case I would suggest to provide that quantitative information here. Otherwise consider reducing the discussion or move

it to a less prominent part of the discussion. Maybe you could also connect it more to the paper, e.g. as an argument for focusing environmental impact studies on macrofauna because they seem more relevant in terms of biomass and ecosystem function as in typical abyssal areas.

Connected to this: Page 10, line 16 'The contribution of meiofauna to benthic biomass generally increases along a bathymetric gradient [. . .] which suggests the occurrence of a dynamic equilibrium between meiofauna and macrofauna at abyssal depths.' Again - this is very detailed for a study that does not focus on size class comparisons. I assume the Sibuet paper focuses on non-nodule areas? I understand you want to put forward that macrofauna is particularly important in the CCZ / in nodule areas as they – different to what was previously reported - show but a relative increase as compared to Meiofauna (i.e., neither do they show a relative decrease at depth as compared to meiofauna nor do they scale with meiofauna). This really would need a quantitative basis, i.e., a comparison of macrofauna abundances (better biomass) to meiofauna abundances at your study sites relative to other areas. > consider adding more quantitative information

Connected to this: Page 10, line 23 'Due to its small size, meiofauna is likely more efficient at exploiting the low level of food input, but this interstitial fauna may also be more sensitive to high nodule coverage because its ambit is largely limited to superficial sediments.' Do you mean, that the meiofauna is restricted to the top layer where the available sediment volume is limited by the presence of nodules? I think you dont show the data but I assume that also for polychaetes the top sediment layer is the one where most individuals are found. In any way you could strengthen this idea by comparing your abundance vs. depth relationship with that of meiofauna in nodule areas. > please comment, explain and consider including this information in the manuscript

Page 14, line 7/8 'Overall, the combination of high local diversity, unsaturated rarefaction curves, high levels of cryptic diversity and high rates of species turnover suggest that polychaete diversity in the CCFZ is large and vastly under-sampled.' It needs a

discussion of the most appropriate technologies (sampling gear, analysis) and the expected effort it needs to raise our knowledge to a level appropriate to decide on mining (yes or no, spatial organization of operations and protected areas), and allow for scientifically sound impact assessment and management. Maybe in this context it should also be discussed, if (and why!) the authors believe, that polychaetes may serve as a model group for baseline and impact assessments. Or is this just the 'pet group' of the authors and any other group should be similarly addressed before taking decisions? > please extend discussions to include these points.

MINOR ISSUES

Page 1, line 1-3 Including the metaanalysis performed the study indeed addresses the entire CCZ. However, the stations of this study are all rather in the eastern part. > rephrase the title to not raise false expectations, e.g., by replacing 'across the nodule province of the Clarion-Clipperton Fracture Zone' with add 'across the nodule province of the eastern Clarion-Clipperton Fracture Zone'

Page 1, line 17 '...the SO239 cruise aimed at improving species inventories...'. Was this really the subject of the cruise as a whole or of this expedition? > consider rephrasing

Page 2, line 6 'Only about 1 % of abyssal plains have been explored to date'. In this context of this paper I would restrict the use of the term 'explore' / 'exploration' to deep-sea mining-related activities > consider rewording

Page 2, line 7 'In particular' seems to connect to the previous sentence but in fact does not. > consider remove

Page 2, line 9 '...mainly manganese and iron,...' > I would also mention copper, nickel and cobalt right away here - than you don't have to repeat that in line 13/14.

Page 2, line 15/16 '...the International Seabed Authority [...] is in charge of protecting fauna against any pollution or other hazards...' Pollution is not the main concern in

the context of nodule mining and expected impacts related to this study. > I would rephrase. Maybe just refer to harm (i.e., 'protecting fauna agains harm')?

Page 2, line 25 I dont understand what is meant with 'scaling issue'. Is this referring to the uncertainties connected to effects of the full scale, long-term operations with large plumes as compared to single experimental tracks? > please rephrase / be more specific

Page 2, line 27 > replace 'the high diversity' by 'a high diversity'

Page 3, line 23 'test the hypotheses that support spatial conservation planning in the CCFZ'. I don't think that these hypotheses (that the authors think would sever as guidance or that form the basis of the current regional management plan are explicitly stated somewhere in the publication. > consider being more specific here or state them elsewhere in the paper

Page 4, line 1/2 > replace '...were located between 4000 and 5000 m depth...' by 'had water depths between 4000 and 5000m'

Page 4, line 12/13 '...all nodules picked up from the sediment surface, washed and individually measured and weighed...' It should be mentioned already here that the water that was used for washing the nodules was sieved after washing. Have the nodules themselves been inspected for small polychaetes, e.g, living in tubes attached to the nodules? > rephrase and make sure to mention somewhere in the paper, if the data also include nodule-associated polychaetes

Page 4, line 18 '...The sieve residues from the overlying water and the washed nodules were combined with all layers for the community analysis...' Was the material combined (i.e., before analysis) or the data? > specify in the text

Page 4, line 20/21 '...(see Section 2.3 DNA extraction, amplification, sequencing, and alignment)' No need to refer to a section that follows directly > remove

Page 4, line 24/25 '...and 1600 bp of 18S genes...' ? Are 18S data really used in this

study (I could not find it later on)? If not: restrict M&M to 16S and COI or discuss why that approach was not successful or not included in the analyses.

Page 5, line 13 'To separate closely related species...' [...] observed between intraspecific and interspecific variations' What does 'closely related species' mean? Specimen that could not be discriminated based on morphology? > specify Page 5, line 13-17 To separate closely related species [...] observed between intraspecific and interspecific variations' This section is describing the principle not what acutally was done. This does not fully qualify for a Materials and Methods part > Move to another part of the study (introduction?) or rephrase.

Page 5, line 25 '. . .to calculate nodule density. . .' Is nodule mean size or size distribution also considered in this study? If not, why was this not included as a parameter that may shape communities? > explain, consider adding explanation to the paper

Page 5, line 25-29 'Particulate organic carbon flux (POC, mg C m$-2$ d$-1$) at the seafloor for our study areas [. . .] applying the Suess algorithm (POC at the seafloor as a function of the net primary production scaled by depth; Suess, 1980; Table 2).' How do POC fluxes estimated with different methods compare where they overlap (i.e. in the study area?) > consider adding that information to the paper.

Page 6, line 1 '2.6 Regional-scale data' Also the Ocean Productivity-based POC fluxes in the previous section refer to the regional scale > choose another headline, e.g., 'Regional scale polychaete community data'

Page 6, line 6 > add references for ES163 and bootstrap

Page 6, line 20/21 'Spearman correlations were sought between biotic and abiotic variables, using data from the SO239 cruise in the CCFZ and data compiled from the literature.' The data used for these correlations should match the data sources listed in section 2.5 > to avoid confusion I suggest to just refer to section 2.5. here. If the 'biotic and abiotic variables' include data not mentioned in section 2.5 add them there.

Page 6, line 23 to Page 7, line 14 Also in this section it should be described what has been done while a description of how the methods work does not seem appropriate for the M&M section (e.g., ' Low values of m give a high weight to dominant species, high values of m give a high weight to rare species.'). > Rephrase, possibly move parts to other sections

Page 7, line 21 '...tended to decrease from east to west with high spatial variation' 1. the main axes does not seem to go strictly longitudinal > replace 'east to west' by 'southeast to northwest' 2. 'high spatial variation' would make more sense in a study design, that follows a clear geographical transect. > consider rephrasing, e.g., 'high variability between neighboring areas'.

Page 7, line 21 'The mean abundance in each study area tended to decrease from east to west with high spatial variation...' 'with high spatial variation' would make more sense in a study design, that follows a clear geographical transect with similar lateral distance between sampling points. > consider rephrasing, e.g., 'high variability between neighboring areas'

Page 8, line 2 'The relative contributions of trophic guilds also varied among the areas...' Is there an explanation found somewhere, how trophic guilds were determined? > If not, add description and references to M&M.

Page 8, line 6 'Off the 1223 polychaetes, 1118 specimens belonging to 78 possible genera within 40 families were identified down to Morphospecies...' What are 'possible genera'? > consider rewording, e.g., '...possibly belonging to 78 genera...'?

Page 8, line 6/7 '1118 specimens [...] were identified down to morphospecies (see Section Data availability)' Not sure why you refer to that section here. > please provide explanation and consider including it in the text.

Page 8, line 14 'The mean number of species tended to decrease from east to west with high spatial variation...' see comment above (regarding Page 7, line 21, second

comment)

Page 10, line 13/14 'Both processes [i.e., increased friction and sediment deposition / biodeposition rates] may stabilize sediments and increase organic carbon supply as tube lawns do' I dont see the connection to sediment stability. > please explain better what your idea is here

Page 10, line 15/16 'The divergent response of meiofauna to the presence of nodules further suggests some sort of competition between meiofauna and macrofauna.' I can see that - if nodules increase food supply but meiofauna abundances are relatively small, meifauna may be unable to make full use of the additional food. What I don't understand is why the reason does need to involve competition with macrofauna (see also my major comment on the meiofauna discussion above). > please provide explanation and consider including it in the text.

Page 11, line 34/35 'No significant correlation was however found between alpha diversity and productivity, neither at the NE Pacific scale nor at the scale of the whole CCFZ.' Do the authors have a hypothesis why this can be the case? Could it be related to the fact that most of the tested areas lie within more or less similar mesotrophic conditions and that this 'biased' data set is not fully appropriate to address this question? > please consider discussing the reason for the missing significant correlation of diversity and productivity on larger scales.

Page 12, line 5/6 'The fact that the APEI#3 lies mostly north of the Clarion Fracture Zone may however also contribute to its dissimilarity with the areas located in the CCFZ per se.' This statement reads quite vague as the idea of geographical barriers is not mentioned and elaborated before the next section > please consider adding (see next section) after the statement.

Page 12, line 12/13 '...characterized by a peak and through ...' Typo > change 'through' to 'trough'

Page 12, line 24-26 'However, species identification was based on morphology only, although cryptic species are common among scavenging amphipods, even in abyssal lineages (Melo, 2004; Havermans et al., 2013)' Another reason is, of course, that scavenging amphipods are typically highly motile. > consider adding mobility as an argument why scavenging amphipode distribution is not limited by fracture zones.

Page 13, line 5 'In other words, nodule mining would affect each year an area that is equivalent to the average geographic range of a polychaete species.' This sounds like one mining operation would lead to the extinction of one polychaete ('only' - as some may argue). > consider removing 'a', i.e., write 'equivalent to the average geographic range of polychaete species.. . .'

Page 13, line 27/28 '. . .suggesting that such extreme environmental conditions. . .' I don't share the view that the deep sea is per se an extreme environment. > replace 'such extreme' with 'the specific' or exploain what specifically is considered extreme

Page 14, line1/2 'This highlights a shortcoming of COI-based barcoding because success rates for COI sequencing are generally low. . .' ? Are current molecular approaches appropriate if only are relatively small proportion could be identified based on 16S and COI and even less with both? Where is the problem and can it be overcome? If there new promising methods that base on other regions of the genome: how can we safeguard comparability of the full data set including new and older data?

Page 15, line 2-4 'The latter estimate assumes that we have sampled 0.1 % of the polychaete species in the CCFZ and that these species have narrow geographical ranges about the size of a yearly mined area.' If I understand right, this refers to the expected annual area exploited as part of one mining operation – not the total annually mined area > replace 'a yearly mined area' with 'the area that will presumably mined in one year by a single mining operation'.

Page 31, Fig. 3 Irrespective of the fact that the variables are provided in the diagonal panels I would prefer if to the side of the plot the variables would be indicated like in

https://images.app.goo.gl/oFQRE6xD7fvFwxJR6

Page 32, Fig. 4 '...in relation to the 2002–2018 average particulate organic carbon (POC) concentration at the seafloor along the CCFZ. The background map shows average POC flux at the seafloor during the 2002–2018 period.' How can the maps show relations to POC concentration and flux at the same time? > consider rephrasing the caption. The caption should also state that this shows / includes data from published studies and refer to section 2.6

---

## Referee Comment (RC3) · Anonymous Referee #3 · 29 Aug 2019

A well written manuscript and a huge contribution of data for an area with limited but increasing data. This will be a useful resource for other researchers and deep-sea management in the region.

A few minor comments to accompany the comments on the attached manuscript. To me the hypotheses were not clear, I think the manuscript would really benefit if they were clearly defined in the introduction and revisited in the conclusions. An overview of why benthic diversity is important to a broader audience would be useful for the bigger picture as well as the ecological role of polychaetes within benthic communities. Why people should care about them? I love polychaetes but a lot of people don't. A

brief description of the mining process, not all readers will be aware of this may be with a comment on the current likelihood of these operations happening and if so when. As stated the diversity estimates are very different, is there an additional method that can be used? With two measures that are both biased towards "singletons" is there a method that is not? Some images of the polychaetes would be nice, especially as photography was an important part of the method. These can be really useful for other research groups, will these be made publicly available? Maybe include a plate in the methods or results. I am not 100% familiar with the diversity analysis so can not critically comment on the methods/results for those sections. I leave this to the other reviewer and the editor. Many terms are not clearly defined but really important as often misinterpreted between papers. Table 1 could be supplementary.

Please also note the supplement to this comment:
https://www.biogeosciences-discuss.net/bg-2019-255/bg-2019-255-RC3-supplement.pdf

―――――――――――――――――――――――

**Supplement:**

[revised manuscript text omitted]

---

## Referee Comment (RC4) · Anonymous Referee #4 · 30 Aug 2019

The manuscript by Bonifacio and colleagues represents a relevant contribution to the study of polymetallic nodules fields. As the Authors state, it is necessary to understand ecological processes and diversity patterns occurring in these environments and to assess the impact of mining activities before starting with their exploitation, and in this frame, this manuscript is of great value. I would therefore endorse its publication in Biogeosciences. The manuscript is in my opinion clear and well-written. My only concern is represented by the use of the word "morphotype", as it is a somewhat ambiguous term. The most widespread use (at least, in my experience) regards morphotypes as divergent morphological variants within the same alleged species, pointing at either cryptic diversity or phenotypic plasticity. However, in this case it is employed to define

individuals that morphologically can be assigned to the same taxon; if I correctly understood, the use of the term "species" or "taxa" has been avoided because molecular data often challenge this interpretation. I think however that in this case the best way would be the use of "morphospecies" or "taxa identified on the basis of morphological features". The term "morphospecies" is employed twice, in both cases at page 8. At line 7 the term is used in the same way of "morphotype" in the remaining manuscript, and as I would advise, but at line 13 the use of "morphospecies" is inappropriate, as here the Authors clearly refer to taxa identified by the combined morphological and molecular data. Morphospecies clearly do not correspond to OTUs (p. 8, line 13), and here the Authors are referring to OTUs. I suggest to carefully re-read the manuscript, as there is some terminological confusion around "morphotype", "morphospecies" and "OTU".

Some minor comments follow.

P. 1, line 10: I suggest to add "environmental" to "footprint".

P. 7, line 31: Replace "paranoids" with "paraonids".

P. 11, line 18: "(Magalhães and Bailey-Brock (2017)": replace with "(Magalhães and Bailey-Brock, 2017)"

P. 11, lines 27-28: Although all Eunicida are usually considered as carnivores, Jumars et al. (2015) suggest that the diet of Lumbrineridae might be more varied, and that sediment and decaying vegetal debris might represent an important diet component for several species. In particular, the reported characterising species belongs to the genus Lumbrinerides, a genus including small, possibly pedomorphic species that at least in shallow environments occupy an ecological niche totally different from larger species of Lumbrineris and Scoletoma. I think that in this case carnivory is not obvious.

P. 12, lines 20-21: results by Guggolz et al. (2018) have been partially ridiscussed in Guggolz et al. (2019: Scientific Reports 9: 9260). I suggest to check and cite also this

work.

---

## Author Comment (AC1) · 12 Nov 2019

Referee comments in black.

*Author's responses in green.*

The paper presents the data on the diversity patterns in polychaetes collected on a cruise that was "aimed at improving species inventories, determining species ranges - -" in the polymetallic nodule area of the Clarion-Clipperton Fracture Zone (CCFZ).

Given the importance of the results, for areas that may be subject to disturbance by nodule mining, I do not understand why there is no reference to the distribution of the rest of the major macro-infaunal groups, given that Wilson (2017) showed that different groups respond in different ways. It is important to know whether similar distribution patterns to those found for polychaetes occurred in the other groups in the studied area.

*We agree that, for the time being, the diversity and distribution patterns of different taxonomic and functional groups must be assessed. To achieve this goal, considering the diversity of the abyssal fauna, as well as the large proportion of new species, each taxonomic group has been processed by its own set of specialists. This is a chance as the lack of taxonomic expertise is a major impediment to our knowledge of the abyssal fauna. As a consequence, also, the results for each group taxonomic group will be published separately – we all have to valorize our work. Results on tanaids for example, the second most abundant group of the macrofauna after polychaetes have just been published (Blazewicz et al. 2019). Tanaid assemblages show similar patterns as polychaetes, this is mention page 12 line 16 (the reference has been updated from Pabis et al. submitted to Blazewicz et al. 2019). Eventually, the results for each taxonomic group, from this project as well as other ongoing projects will be synthesized to provide a global picture of distribution and diversity patterns of the benthos in the CCFZ.*

It is clear that a lot of effort has gone into obtaining and working up this data. It is therefore unfortunate that a poor description of what was done and the reasons for this make it difficult to understand what fraction of the infauna was sampled and the consequences for the overall polychaete biodiversity and the comparisons between different box cores.

*This general comment is addressed more specifically below where are discussed the epifauna versus infauna and nodule infauna versus sediment infauna.*

Given that the topography of the polymetallic nodules affects local water flow patterns and creates different microhabitats (Mullineaux, 1989). The cruise report gives information on the nodule differences between individual cores but this information does not appear to have been used in analyzing polychaete numbers and distributions, although this should have a major influence on species composition and numbers. I would like to see an analysis of polychaete species distribution and numbers with respect to the differences in nodule topography, numbers and sizes between cores.

*The influence of nodules on the structure and composition of polychaete assemblages has been tested. The variable named "nodules" is the wet weight of nodules per box-core, extrapolated to a square meter (see page 6 lines 26, the data are given in Table 1). This measure of nodules density was related to the abundance and richness of polychaetes from each box core sample using Spearman correlations (Figure 3); and related to polychaete composition by the Redundancy Analysis (RDA, Figure 7). Unfortunately, the number and size of individual nodules were not recorded during the cruise.*

*The variable "Nodules" has been renamed to "Nodule density" throughout the manuscript and figures for clarity.*

Treating all box core samples within an area as replicates does not appear to be valid.

*Box-core samples within areas have been used as replicates in order to assess regional-scale variations in the abundance, richness and composition of polychaete assemblages. We think that the approach is valid because the distance between areas is much larger than the distance between individual samples within areas. The samples within an area can be considered as representative of the populations within that area compared to the other areas. There are two instances where the assumption is questionable:*

*- In the BGR area, two sub-areas were sampled: a prospective Area (PA); and a reference Area (RA), with a low nodule density. There was no statistical difference in the abundance and richness of polychaetes between the two sub-areas. Samples from the two sub-areas were thus considered as replicate samples for the BGR area.*

*- In the IOM area, three sub-areas were also sampled: one that had been disturbed by a BIE-experiment, one that had been impacted by the plume and one control, undisturbed area. The three sub-areas had otherwise similar environmental settings and there were no statistical differences in the abundance or richness of polychaetes.*

*This has been clarified in the sampling strategy with the addition of the following paragraph (page 4 line 26 – page 5 line 5) and the reference in reference list:*

*"The sampling strategy resulted from a combination of objectives that were unique to each area, together with the overarching aim of describing alpha and beta diversity patterns across a productivity gradient that included both contract areas for nodule exploration and an APEI (Martínez Arbizu and Haeckel, 2015). In the BGR area, two sub-areas were sampled: a Prospective Area (PA) that could be mined in the future and a Reference Area (RA) that could serve as a preservation area. In the IOM area, three sub-areas were sampled: one that had been directly disturbed by a BIE-experiment (Radziejewska, 2002), one that had been impacted by the plume and one control, undisturbed area. These levels of sampling stratification are however out of the scope of the present study, which focuses on variations between contract areas. After checking that there was no statistically significant difference on the abundance and richness of polychaetes between sub-areas, all samples within an area were deemed representative of that area and considered as replicate samples. The level of replication within areas accordingly varied as a function of sampling stratification. The aim was to collect a minimum of five replicate samples per strata but due to sampling failures and time constraints, it couldn't be systematically achieved (Table 1)."*

Similarly no use has been made of the data on species distribution with respect to sediment depth.

*We considered that the vertical distribution of the species was not relevant to the questions we asked. Moreover, patterns in vertical distribution are difficult to test because sub-samples from each sediment layer can hardly be considered as independent samples, which violates the most basic assumption of all statistical tests.*

There should be an enlargement of the sampled area in Fig. 1 to allow for the individual core stations to be plotted. A section on the differences between samples taken within an

individual area would be beneficial, since a lot of information on the ecology has been lost by only comparing means for each area.

*Figure 1 (below) has been modified to show the distribution of individual core stations within area or sub-area. The caption has been modified accordingly:*

*From "Figure 1. Map of the nodule exploration contract, reserved areas and areas of particular interest (APEI) in the Clarion-Clipperton Fracture Zone (CCFZ) showing the sampling areas from this study and previous macrobenthic surveys. The areas sampled during the SO239 cruises are shown in color. The background map shows average particulate organic carbon (POC) flux at the seafloor during the 2002–2018 period."*

*To "Figure 1. (a) Map of the nodule exploration contracts, reserved areas and Areas of Particular Environmental Interest (APEI) in the Clarion-Clipperton Fracture Zone (CCFZ) showing the sampling areas from this study (in color) and previous macrobenthic surveys ; the background map shows the average particulate organic carbon (POC) flux at seafloor during the 2002-2018 period. The areas sampled during the SO239 cruises are enlarged in following figures: BGR (b and c), IOM (d), GSR (e), Ifremer (f) and APEI#3 (g); with detailed local hydroacoustic maps based on multibeam system EM122 (Martínez Arbizu and Haeckel, 2015; Greinert, 2016) in background."*

[Figure]

Sampling strategy

The sampling strategies need to be clearly explained with reference to the following questions, either by amplification of the text or by giving a references.

1. Since there are 9 APEIs, why was the only one sampled from an oligotrophic area when the exploration blocks sampled were all in mesotrophic areas?

*The question was about the representativeness of APEIs given the fact that these APEIs were pushed at the periphery and even beyond the Fracture Zones that delimit the nodule province and thus into a different productivity regime. Two APEIs could potentially be sampled APEI#3 and APEI#6. APEI#6 was sampled by other teams in the framework of other projects the same year as the SO239 cruise– in order to avoid duplication of efforts and increase knowledge on APEIs, the APEI-3 was sampled during SO239.*

2. Within each block how was it decided where to sample, given the problem of determining the geographic range of species? [see Wilson (2017) for one approach to this.]

*The sampling strategy resulted from a combination of objectives that have been explicated in the text (Page 4 line 27 – page 5 line 6):*

*The sampling strategy resulted from a combination of objectives that were unique to each area, together with the overarching aim of describing alpha and beta diversity patterns across a productivity gradient that included both contract areas for nodule exploration and an APEI (Martínez Arbizu and Haeckel, 2015). In the BGR area, two sub-areas were sampled: a Prospective Area (PA) that could be mined in the future and a Reference Area (RA) that could serve as a preservation area. In the IOM area, three sub-areas were sampled: one that had been directly disturbed by a BIE-experiment (Radziejewska, 2002), one that had been impacted by the plume and one control, undisturbed area. These levels of sampling stratification are however out of the scope of the present study, which focuses on variations between contract areas. After checking that there was no statistically significant difference on the abundance and richness of polychaetes between sub-areas, all samples within an area were deemed representative of that area and considered as replicate samples. The level of replication within areas accordingly varied as a function of sampling stratification. The aim was to collect a minimum of five replicate samples per strata but due to sampling failures and time constraints, it couldn't be systematically achieved (Table 1).*

3. Given the known high percentage of species represented by only a single individual in box core samples from red clay polymetallic nodule areas (Hessler & Jumars, 1974 and several later papers), why were such a small number of box cores taken in each area, as opposed to taking the same overall number of box core samples from fewer areas.

*The SO239 cruise was a multidisciplinary cruise in the framework of a collaborative European project. The sampling strategy is thus a tradeoff between multiple objectives, optimization of transit time and adaptive management. As explained above, the sampling design within area was guided by specific objectives for each of the area. The sampling effort was constrained by ship time even though the cruise lasted for 50 days, which is close to the endurance of the RV Sonne.*

4. Why was it decided (apparently post-sampling) to treat every box core sample from a single area as a replicate, given the geochemical differences within some of the areas and

also differences in the number, size and depth distribution of the polymetallic nodules in box cores from the same area?

*The reason for considering box core sample from a single area as replicates has been explained above. It should be noted here that variations in organic content and sediment grain size were low within each area (see Table below from Hauquier et al. 2019). Variations in nodule density were high in some instances but the influence of nodule density on the abundance, richness and composition of polychaete assemblages was assessed (Figure 3 and Figure 7).*

|  | CPE (µg ml$^{-1}$) | TN (weight %) | TOC (weight %) | clay (%) | silt (%) |
|---|---|---|---|---|---|
| **APEI-3** | 0.06 ± 0.00 | 0.10 ± 0.00 | 0.29 ± 0.02 | 35.48 ± 5.40 | 61.63 ± 4.91 |
| **IFREMER** | 0.08 ± 0.02 | 0.12 ± 0.03 | 0.40 ± 0.07 | 15.41 ± 1.77 | 71.89 ± 3.80 |
| **GSR** | 0.11 ± 0.05 | 0.16 ± 0.07 | 0.47 ± 0.11 | 15.64 ± 1.66 | 70.89 ± 2.42 |
| **IOM** | 0.17 ± 0.03 | 0.12 ± 0.01 | 0.53 ± 0.12 | 10.74 ± 0.47 | 73.39 ± 0.89 |
| **BGR_RA** | 0.20 ± 0.09 | 0.10 ± 0.02 | 0.43 ± 0.12 | 11.21 ± 0.89 | 72.90 ± 1.27 |
| **BGR_PA** | 0.28 ± 0.11 | 0.12 ± 0.01 | 0.58 ± 0.08 | 12.21 ± 0.65 | 70.31 ± 3.01 |

5. Each box core was sliced into 0-3, 3-5 and 5-10 cm depth sections that were sieved separately. Why was this done when the data from each layer were then added for the data analysis? The slicing procedure is not described but when slicing box cores polychaetes are frequently fragmented. What precautions were taken that an individual was not counted more than once, for example by only counting head-ends.

*The layering was mostly used in order to facilitate sieving and sorting. At the ECHO I site in the CCFZ, Spiess et al. (1987) reported that about 70% of the macrofauna was concentrated in the top water and 0-1 cm depth, while less than 10% was found in the 5-10 cm depth. To prevent double counting, we have only counted head-ends.*

*This has been clarified (page 5 lines 10-11):*

*From "The upper 10 cm of each core was sliced into three layers (0–3, 3–5 and 5–10 cm), each layer transferred into cold seawater and sieved using the same mesh size."*

*To "The upper 10 cm of each core was sliced into three layers (0–3, 3–5 and 5–10 cm) **to facilitate sieving and sorting;** each layer was transferred into cold seawater (4 °C) and sieved using the same mesh size..."*

*A precision about the counting head-ends was added page 6 lines 9-10:*

*From "Preserved specimens were examined under a Leica M125 stereomicroscope and a Nikon Eclipse E400 microscope and morphologically identified…"*

*To "Preserved specimens were examined under a Leica M125 stereomicroscope and a Nikon Eclipse E400 microscope, **counted (anterior-ends only)** and morphologically identified …"*

6. The banked data shows that, although most individuals were found in the 0-3 cm layer, in some cores over 20% of the individuals were present in the 5-10 cm layer. It is therefore reasonable to assume that an unknown fraction of the biodiversity was lost in the samples because the cores were not sampled for macrofauna deeper than 10 cm. In some deep-sea sediments (at > 2000 m) polychaetes are known to penetrate over 100 cm below the sediment surface and the major infaunal biomass can often be found below 10 cm sediment depth. Given that some box cores were sampled below 10 cm depth, since nodules were

recorded at 25 cm depth, why is no mention made of animals being present below 10 cm? Can the authors cite any reference to deeper sampling for infauna in the working areas? It might have been assumed that only the upper 10 cm of sediment would be disturbed by nodule harvesting, however most nodule-mining prototypes have been based on bottom crawlers that would cause sediment compression and affect deeper-burrowing organisms.

*Hessler and Jumars (1974) sampled the CCFZ macrofauna down to 20 cm but did not report on the vertical distribution of the fauna. Since then, we are not aware of any study that sampled the macrofauna below 10 cm in the CCFZ and in its recommendations for contractors, the International Seabed Authority also suggest to sample down to 10 cm. In fact, the 5-10 cm layer of sediment is already very sticky with little evidence of bioturbation and difficult to sieve. After sample processing, the box-corers were emptied by hand. Nodules were occasionally found buried in sediments, but this was rare. These buried nodules won't be a target for the mining industry. Yet, on one occasion a large maldanid polychaete was found at about 50 cm depth. We thus agree that large burrowers can live below 10 cm but their densities are so low that a box-core sample is too small to provide a precise and accurate estimate of these populations. For all these reasons, we followed the widely use standard of sampling down to 10 cm only.*

7. Surface polymetallic nodules do penetrate the sediment to a degree. We are not told now deep this was in the different cores. Even if it was only 1 cm, given the large number and size of the nodules in some cores (see photographs in the cruise report) this would greatly decrease the volume of sediment available for sieving in the 0-3 cm layer and bias the results, considering that all cores were equated only on an area. The fauna results should also be considered with respect to sediment volume.

*We agree that quantifying the volume occupied by nodules would have been an important factor to consider. The depth penetration, the size and area covered by the nodules were not assessed during the SO239 cruise, but we acknowledge that this should be done (and in fact has been done during a subsequent cruise).*

*The following phrase was added in the Discussion to point this out page 11 lines 14-15:*

*"In our study, the volume and surface occupied by nodules were not quantified but the positive relationship between nodule density and polychaete abundance shows that space is not a limiting factor for polychaetes."*

8. It is unclear what happened to animals collected when the nodules were washed free of sediment. Were these animals added to the 0-3 cm layer and if so was this done before picking animals off the nodules?

*The larger epifauna attached to the nodules was immediately picked up before nodules were removed from box core sample. The nodules were then gently washed with cold sea water to remove most of the sediment sticking to the nodules and possible associated fauna. The smaller sessile fauna remained attached to the nodules and was processed on its own. The sediments washed from the nodules were added to the 0-3 cm layer.*

*The text has been changed to precise sample processing (page 5 lines 7-16):*

*From "The overlying water column was siphoned and filtered using a sieve of 300 μm of mesh size. The box core sample surface was photographed, and all nodules picked up from the sediment surface, washed and individually measured and weighed. The upper 10 cm of*

*each core was sliced into three layers (0–3, 3–5 and 5–10 cm), each layer transferred into cold seawater and sieved using the same mesh size. The overlying water residue and the 0–3 cm layer were immediately sieved in the cold room with cold seawater (4 °C) and then live-sorted. All polychaete specimens were photographed, individualized and preserved in cold (-20 °C) 80 % ethanol and then kept at -20 °C (DNA-friendly). The 0–3 cm residue, 3–5 and 5–10 cm layers were fixed in formalin for 48 to 96 h and preserved in 96 % ethanol and later sorted in the laboratory (not DNA-friendly). The sieve residues from the overlying water and the washed nodules were combined with all layers for the community analysis."*

*To "The overlying water was siphoned and sieved using a sieve of 300 μm of mesh size. The box core sample surface was photographed, and all nodules picked up from the sediment surface, washed with cold seawater over a 300 μm-mesh sieve and individually weighed.* ***Sessile polychaetes, if present, remained attached to the nodules and were not considered in this study.*** *The upper 10 cm of each core was sliced into three layers (0–3, 3–5 and 5–10 cm) to facilitate sieving and sorting; each layer was transferred into cold seawater (4 °C) and sieved using the same mesh size. The 0–3 cm layer was immediately sieved in the cold room with cold seawater (4 °C).* ***The sieve residues from the overlying water and nodule washing were added to the 0-3 cm layer and live-sorted.*** *All polychaete specimens were photographed, individualized and preserved in cold (-20 °C) 80 % ethanol and then kept at -20 °C (DNA-friendly). The 0–3 cm residue, 3–5 and 5–10 cm layers were fixed in formalin for 48 to 96 h and preserved in 96 % ethanol and later sorted in the laboratory (not DNA-friendly). All layers were combined for the community analysis."*

9. What happened to the animals picked off the nodules, were these treated as epifauna and not considered here? Some serpulids are included in the species listed in the dataset – were these all epifaunal on the nodules?

*Yes, the sessile epifauna, including sessile polychaetes such as serpulids, has been treated separately and are not considered in this study. The serpulids present in the samples can be from nodule washing residue.*

*The following sentence was added page 5 lines 9-10:*
*"Sessile polychaetes, if present, remained attached to the nodules and were not considered in this study."*

10. Polychaetes are known to occur as infauna within the polymetallic nodules (Thiel et. al. 1993). Some polychaete species were only found in crevices within the nodules and knowledge of differences in species composition in nodules from the different exploration blocks would be important information. Was this part of the infauna sampled? Will there be a separate publication dealing with the nodule-associated fauna, since the present manuscript does not cover the habitat of the species within the core samples?

*No, the nodule crevice fauna was not sampled during this cruise. We recognize that this is a neglected component of the benthos. Sampling the crevice infauna requires to break up the nodules, this was not done during the cruise.*

11. Both the biomass and the biovolume of the infauna can affect the geochemistry, were these measured?

*Each polychaete specimen was sized and those measures could be used to calculate a biovolume. Conversion to biomass however is a rough approximation due to the large number of damaged specimens and not included in this paper.*

Specific Comments

Table 2: the data should be checked and the header clarified. Does the data refer to only polychaetes or all macroinfauna? For example, Table 7 in Wilson (2017) gives a mean polychaete density of 21 individuals 0.25 m-2 for Domes A compared with the 16 given in the present Table 2. It is also necessary to know if like is being compared with like with respect to combined fractions of samples from the box cores. Since most of the previous studies did not look for cryptic species I think you should give the number of morphologically identified species in parenthesis for the sites recorded in the present study.

*Data in Table 2 refer to polychaetes only. Glover et al. (2002) and Wilson (2017) published results from the same dataset, except that for DOMES A, Glover et al. have considered 47 box cores while Wilson has considered 41 box cores, which may explain the differences in mean polychaete density between the two studies. In Table 2 of our manuscript, data for DOMES A and ECHO 1 were taken from Glover et al. (2002), except for Bootstrap values, which were taken from Wilson (2017).*

*Rows were added in Table 2, for each of the three sites DOMES A, PRA and ECHO 1 in order to make the difference between data taken from Glover et al., 2002 and data taken from Wilson (2017). The correspondent rows have been changed, from:*

| Area | Year | References | Depth (m) | Latitude | Longitude | Number of box cores | Mean abundance (ind. 0.25 m$^{-2}$) | Total number of species | ES163 | Bootstrap | 2002–2018 average POC at the seafloor (g C m$^{-2}$ year$^{-1}$) |
|------|------|------------|-----------|----------|-----------|---------------------|-------------------------------------|-------------------------|-------|-----------|---------------------------------------------------------------|
| DOMES A | 1977/78 | Glover et al. (2002); Wilson (2017) | 5100 | 8.45 | -150.78333 | 47 | 16 | 104 | 56 | 203 (based on 41 box cores) | 1.46 |
| PRA | 1989 | Glover et al. (2002); Wilson (2017) | 4800 | 12.95 | -128.31667 | 16 | 65 | 100 | 47 | 310 | 2.04 |
| ECHO 1 | 1982 | Glover et al. (2002); Wilson (2017) | 4500 | 14.6666667 | -126.41667 | 15 | 42 | 113 | 60 | 274 (based on 14 box cores) | 2.05 |

*To:*

| Area | Year | References | Depth (m) | Latitude | Longitude | Number of box cores | Mean abundance (ind. 0.25 m$^{-2}$) | Total number of species | ES163 | Bootstrap | 2002–2018 average POC at the seafloor (g C m$^{-2}$ year$^{-1}$) |
|------|------|------------|-----------|----------|-----------|---------------------|-------------------------------------|-------------------------|-------|-----------|---------------------------------------------------------------|
| DOMES A | 1977/78 | Glover et al. (2002) | 5100 | 8.45 | -150.78333 | 47 | 16 | 104 | 56 | | 1.46 |
| | | Wilson (2017) | | | | 41 | | | | 203 | |
| PRA | 1989 | Glover et al. (2002) | 4800 | 12.95 | -128.31667 | 16 | 65 | 100 | 47 | | 2.04 |
| | | Wilson (2017) | | | | | | | | 310 | |
| ECHO 1 | 1982 | Glover et al. (2002) | 4500 | 14.6666667 | -126.41667 | 15 | 42 | 113 | 60 | | 2.05 |
| | | Wilson (2017) | | | | 14 | | | | 274 | |

Figure 3 needs a lot more explanation and labelling. Below the diagonal some of the plots appear to use mean data from areas and others data from individual cores without a clear explanation.

*Indeed, to make it clearer the Figure 3 has been separated in two: (a) by box-core samples and (b) by area. Thus, the caption of Figure 3 has been changed:*

*From "Figure 3. Correlation matrix between biotic and abiotic variables from sampled areas within the eastern CCFZ. Diagonal panels show the distribution frequency of values for each variable. Below-the-diagonal panels show the correlation plot between pairs of variables. Above-the-diagonal panels show the Spearman coefficient correlations between pairs of variables. "\*" indicates p < 0.05, "\*\*" p < 0.01 and "\*\*\*" p < 0.001."*

*To "Figure 3. Correlation matrix between biotic and abiotic variables from sampled areas within the eastern CCFZ. Diagonal panels show the distribution frequency of values for each variable. Below-the-diagonal panels show the correlation plot between pairs of variables. Above-the-diagonal panels show the Spearman coefficient correlations between pairs of variables.* **Abundance, richness and nodule density per box-core (a) and average biotic and abiotic variables per area (b).** *POC Eastern values provided by Volz et al. (2018); POC NE Pacific values were estimated in the present study. "\*" indicates p < 0.05, "\*\*" p < 0.01 and "\*\*\*" p < 0.001."*

In Figures 7 and 9 yellow text does not show well on a white background – use a coloured background for the text.

*The yellow has been replaced by a dark orange in all figures to better highlight the graph and texts. See for example Figure 9 below as an example:*

[Figure]

Conclusions: The first paragraph does not belong here – it should be in the introduction.

*This has deleted from Conclusions but partially integrated into Introduction (page 4 lines 3-10) as suggested.*

Supplementary data: It would be useful to have a species list available as a supplement to the paper, although I realise that the taxonomic studies are ongoing.

*We agree that a species list would be quite helpful, but about 90% of the polychaete species in the CCFZ are new to Science (Glover et al., 2002). The task of identifying and then naming most of the 275 morphotypes found in this study is thus huge and out of the scope of this paper. The best we can do is to provide the list of morphospecies together with their DNA barcodes when available. This information are available as supplementary material on PANGEA in the abundance file.*

*The following sentence was added in the beginning of Results to clarify the availability of the dataset page 8 lines 23-24: "The dataset has been archived in the information system PANGAEA and is available in open access (Bonifácio et al., 2019)."*

References

Hessler, R. R. & Jumars, P. A. (1974) Abyssal community analysis from replicate box cores in the central North Pacific, Deep SeaRes., 21: 185-209

Mullineaux, L. (1989) Vertical distributions of the epifauna on manganese nodules: implications for settlement and feeding. Limnol. Oceanog., 32: 1247-1262

Thiel, H. et. Al. (1993). Manganese nodule crevice fauna. Deep Sea Res. Part I 40: 419–423

Wilson, G. D. F.:(2017) Macrofauna abundance, species diversity and turnover at three sites in the Clipperton-Clarion Fracture Zone,. Mar. Biodivers., 47: 323-347

*References cited in the authors answers:*

*Błażewicz, M., Menot, L., Jóźwiak, P., Pabis, K.: High species richness and unique composition of the tanaidacean communities associated with five areas in the Pacific Polymetallic Nodule Fields, Prog. Oceanogr., 2019.*

*Glover, A. G., Smith, C. R., Paterson, G. L. J., Wilson, G. D. F., Hawkins, L., and Sheader, M.: Polychaete species diversity in the central Pacific abyss: local and regional patterns, and relationships with productivity, Mar. Ecol. Prog. Ser., 240, 157-170, 10.3354/meps240157, 2002.*

*Hauquier, F., Macheriotou, L., Bezerra, T. N., Egho, G., Martínez Arbizu, P., and Vanreusel, A.: Distribution of free-living marine nematodes in the Clarion–Clipperton Zone: implications for future deep-sea mining scenarios, Biogeosciences, 16, 3475–3489,10.5194/bg-16-3475-2019, 2019.*

*Hessler, R. R., and Jumars, P. A.: Abyssal community analysis from replicate box cores in the central North Pacific, Deep Sea Res., 21, 185-209, 1974.*

Martínez Arbizu, P., and Haeckel, M.: RV SONNE Fahrtbericht / Cruise Report SO239: EcoResponse Assessing the Ecology, Connectivity and Resilience of Polymetallic Nodule Field Systems, Balboa (Panama) – Manzanillo (Mexico,) 11.03.-30.04.2015. GEOMAR Report, N. Ser. 025., GEOMAR Helmholtz-Zentrum für Ozeanforschung, Kiel, Germany, 2015.

Radziejewska, T.: Responses of deep-sea meiobenthic communities to sediment disturbance simulating effects of polymetallic nodule mining, Int. Rev. Hydrobiol., 87, 457-477, 2002.

Spiess, F.N., Hessler, R., Wilson, G., Weydert, M.:Environ- mental effects of deep-sea dredging. SIO Reference Report 87-5, Scripps Inst Oceanogr, San Diego, 1987.

Wilson, G. D. F.: Macrofauna abundance, species diversity and turnover at three sites in the Clipperton-Clarion Fracture Zone, Mar. Biodivers., 47, 323-347, 10.1007/s12526-016-0609-8, 2017.

---

## Author Comment (AC2) · 12 Nov 2019

Referee comments in black.

*Author's responses in green.*

The scientific methods are valid and up-to-date and the experiments and calculations are adequately described. The authors acknowledge related work and clearly discriminate their data from data obtained by others. In some cases, where they discuss their data (e.g., regarding winners and loosers of the 'competition' of meio- and macro-fauna) in relation to that of others they should be more specific about the exact content of the cited data so the reader can better comprehend the authors' discussions and conclusions. The data presented are sufficient to reach conclusions part of which – and that is not criticizing the ambitions work behind this study – are pointing out gaps in knowledge. The paper is well structured and reads mostly fine – in some parts of the results the language may seem a bit repetitive (abundances and species numbers are presented with almost the same wording). The abstract provides a good summary. In the way it is currently presented, the first part of the discussion (about the relative success of meio an macrofauna in different deep-sea environments) should be either significantly reduced or more specific with a better and quantitative presentation of data on Meiofauna from the study area from other studies. I would vote for a reduction of that part as it seems to be a bit off the main focus anyway.

To me, the main shortcoming of the paper is, that the authors are very carefully when stating their conclusions and overcautious if it's about the consequences of their findings. As the work touches societal concerns and areas of strong debates the authors should address in more depth the implication of their work regarding procedures and management of nodule mining.

*In our opinion, the main message of this study is that we don't know enough about species diversity and species ranges to be conclusive about the potential impact of nodule mining. We fully understand the need and eagerness to get answers to acute societal concerns, but we must be cautious of not over-interpreting the results of this study. Yet we agree that there are some questions that we can address and recommendations that can be made. In the following lines we will try to address the concerns where we can or explain why we can't.*

e.g., can we decide on regional management based on the available knowledge and are the APEIs appropriate as they are?

*The answer to these questions is 'No' but not for scientific reasons. The location of the APEIs has been initially defined based on available knowledge (see http://www.soest.hawaii.edu/oceanography/faculty/csmith/MPA_webpage/MPAindex.html ) and then severely constrained by contractor areas and reserved areas. A comparison of Figure 3 in Wedding et al. 2013 with Figure 1 in Lodge et al. 2014 clearly shows the difference between recommendations and implementation.*

*This has been clarified in the Introduction (page 3 lines 17-28):*

*From 'Due to the paucity of biological data in the CCFZ, the spatial management plan was designed mainly based on nitrogen flux at 100 m depth (a proxy for trophic inputs to the seafloor), modeled nodule densities, the distribution of large seamounts and the dispersal distances of shallow water taxa (Wedding et al., 2013). The nine proposed 400 x 400 km managed (non-mining) areas were included in the regional management plan for the CCFZ and designated as APEIs (Lodge et al., 2014). Most of the CCFZ however has already been*

*To "Due to the paucity of biological data in the CCFZ, the recommendations issued by Wedding et al. (2013) for the design of a network of protected areas were mainly based on nitrogen flux at 100 m depth (a proxy for trophic inputs to the seafloor), modeled nodule densities, the distribution of large seamounts and the dispersal distances of shallow water taxa.* **One of the main assumptions underlying the management plan is that longitudinal and latitudinal productivity-driven gradients shape the community structure and species distribution of abyssal communities. As a result, Wedding et al. (2013) divided the spatial domain of the CCFZ into 3 x 3 subregions and suggested to create one large no-mining area in each subregion. The size of the no-mining areas was defined with the aim of maintaining viable population sizes for species potentially restricted to a subregion, taking into account the inferred dispersal distances of species and of the plumes created by nodule mining (Wedding et al., 2013). Those principles were implemented in the regional management plan for the CCFZ, which resulted in the designation of 9 APEIs (Lodge et al., 2014).** *Most of the CCFZ however had already been preempted to current exploration contracts and areas reserved for future exploration. Consequently, the APEIs were located at the periphery of the CCFZ thus deviating from an optimal design.*

Are there any specific recommendations that can be provided, e.g., regarding the size or arrangement of mining patches?

*It would be premature to provide specific recommendations, first because the level of confidence on our estimates of species ranges is too low, second because polychaetes may not have the smallest species ranges (see also below).*

What should be the focus of future studies and what would be the expected effort needed to come to scientifically sound conclusions?

*In an ideal world, a stratified random sampling at nested scales, from region down to seascapes, would provide the scales of species turn-over while intensive sampling of selected habitats up to the point where the number of singletons decreases with sample size would provide accurate estimates of species diversity.*

The conclusions part is so far rather summarizing what has been stated already before and may be a good place to discuss these things. The fact, that those discussions are rather limited in the current version of the manuscript is the reason for my general proposition that the manuscript should undergo a major revision before publication.

Below some more detailed comments

MAJOR ISSUES – IMPLICATIONS FOR MINING / MINING REGULATION

Regarding the main shortcoming of the paper mentioned above I urge the authors to significantly extend the discussion of their results towards the implications of their work with regard to nodule mining and its regulation. This could be distributed in several parts of the discussion as well as in a separate section in the discussion or in the conclusions. This, of course, has to be done with some caution to not extend beyond the scope of the study and has to take into account that this is a scientific publication and not a policy paper. However, it is clear that the motivation of the study – and certainly of societies providing the funding for mining-related investigations these days – is to provide the basis for scientifically sound

procedures and decisions regarding deep-sea mineral exploration and exploitation. This should be better reflected in the text. This includes recommendations regarding the management and regulations - where the data of the study allow this - but also specific requests for future investigations where the results reveal significant gaps. Below I am providing some examples where I think the discussion needs to move beyond where it currently terminates.

Page 3, line 15/16 'The distribution of APEIs at the periphery of the CCFZ thus deviates from an optimal design.' Page 11, line 20-23 'The biogeochemical settings as well as the biological patterns of the three size groups of the benthic fauna thus converge to conclude that the structure and functioning of the benthic ecosystem in APEI#3 is not representative of any of the four exploration contract areas included in this study.' Page 12, line 26-29 'The influence of the fracture zones on the dispersal of the abyssal fauna remains to be better understood as the Clarion and Clipperton fractures may act as a barrier for species with low dispersal abilities such as infaunal brooders. If so, the representativeness of seven out of the nine APEIs, which are partly lying beyond the fractures, may be questionable.' If these statements hold true, the concept of APEIs and the regional management plan as a whole don't seem to be appropriate.

What is the advice of the authors to overcome this problem?

*As for now, this is a hypothesis that needs to be further tested. Our main advice is to foster research in the APEIs and to support this research we propose the future Environmental Compensation Fund to be created by the regulations on exploitation of mineral resources in the Area. This recommendation has been added to the Conclusions:*

*Pages 16 lines 21-24:*

*In order to ascertain that the APEIs collectively meet their goal of preserving the biodiversity of the CCFZ an ambitious research agenda is needed, the funding of which could rely on the willingness of contractors and Sponsoring States but could also become a priority of the future Environmental Compensation Fund to be created by the regulations on exploitation of mineral resources in the Area (ISBA/25/C/WP.1).*

What do we know about the other APEIs and how their environmental conditions and faunal communities compare to license areas?

*Studies on other APEIs are ongoing but few results have been published yet. Simon-Lledó et al. (2019a, 2019b) recently described megafaunal community patterns as a function of seafloor heterogeneity and nodule density from imagery surveys. Comparisons with similar megafaunal surveys undertaken in contract areas is however difficult due to a current lack of standardization of methods, both for the surveys and the image-based taxonomy.*

What would be an optimal APEI layout and how would you – from the results of your study – address the question whether an area is suited as APEI or not.

*Our results suggest that the boundaries of the management area and sub-regions used by Wedding et al. (2013) could be improved but do not contradict the conceptual bases of the current management plan. Again, the implementation rather than the design of APEIs is problematic. APEIs had several conservation objectives, including the maintenance of sustainable and healthy populations of minimum viable sizes and a full range of habitat types. The topic of the representativeness of the APEIs is too broad to be addressed here.*

Can we use some easily measured sedimentological (grainsize?) or biogeochemical measurement to assess the probability that an APEI will host similar faunal communities than a specific license area? Should the assessment of correlations of habitat characteristics and fauna in APEIs be a focus of future studies?

*In our constrained multivariate analysis (RDA), the environmental factors that were available explained 13% of the local and regional variations in polychaete community composition. The explanatory power of the model is low and could certainly be improved to some extent by a better understanding of the physico-chemical niche of species. However, the main unknown is most likely about the biology and biotic interactions of species: how long do they live, how do they reproduce and disperse, do they interact and how are they interacting between others. These would be key questions to answer, although much more challenging than looking at correlations of abiotic factors and biological variables.*

If you consider the lack of knowledge potentially only a few years before exploitation commences: Should the ISA setup a scheme by which contractors carry out or fund baseline studies in the APEIs?

*There is not such funding mechanism in the mining code for exploration nor recommendations towards contractors to carry out baseline surveys beyond their contracted area and it's beyond our expertise to assess whether such a mechanism could be implemented in the framework of current exploration contracts. The draft regulations on exploitation of mineral resources in the Area provides for the establishment of an Environmental Compensation Fund (ISBA/25/C/WP.1). The purpose of the Fund does not include the promotion of baseline studies in the APEIs but this is a recommendation we can make.*

*The following lines were added in conclusion (page 16 lines 20-24):*

*'The sampling effort in both the contract areas and the APEI however remains quite limited. In order to ascertain that the APEIs collectively meet their goal of preserving the biodiversity of the CCFZ an ambitious research agenda is needed, the funding of which could rely on the willingness of contractors and Sponsoring States but could also become a priority of the future Environmental Compensation Fund to be created by the regulations on exploitation of mineral resources in the Area (ISBA/25/C/WP.1).'*

One consideration that lead to the APEIs' current position outside the area covered by license areas was to allow for very large areas. In light of the fact that, according to the current planning, only part of the license areas will be used for nodule extraction and the seemingly low species' ranges: do we need APEIs to conserve biodiversity or would the areas inside the patch of license areas, that are not mined do the job? Or do we need APEIs somewhere else, e.g., smaller ones between license areas?

*In the current draft regulations on exploitation of mineral resources in the Area, a plan of work in the case nodule mining shall not exceed 75000 km² (ISBA/25/C/WP.1), which is the size of the exploration contracts areas. Thus, in the current planning, the only area that won't be mined are Preservation Reference Zones (PRZ) and non-mineable areas (slopes, no-nodule, etc…). Since the PRZ haven't been clearly defined yet and since non-minable area do not represent the full range of habitats in the CCFZ, and especially not the most threatened habitats, we believe that APEIs are required.*

*In addition, APEIs are very large for two reasons: 1) To allow for the self-sustainability of populations within the APEI. An alternative would be to create a higher number of smaller inter-connected APEI but we lack data on dispersal range of species, which is different from the geographic range of species, to discuss this alternate design. 2) Avoid the impact of the sediment plume, again we don't have the data to discuss the relevance of this point.*

Page 13, line 1-5 'However, based on the best knowledge we have, our study suggests that [. . .] nodule mining would affect each year an area that is equivalent to the average geographic range of a polychaete species.' Spatial ranges – especially if they are indeed that small – are highly relevant.

Can we use polychaetes as key species here or would we need to have similar data also for other size classes and other groups of macrofauna? What data are available already?

*Polychaetes are the most abundant and most diverse among the macrofauna. Polychaete might however be less threatened than peracarids, which are brooders and show narrower species ranges on average. Polychaetes might also be less functionally important than nematodes, which dominate the metazoan biomass or foraminifera, off which we know very little. If the aim is to monitor and preserve all levels of biological diversity, from gene, to species, to functions then polychaetes are likely not enough. The good news is, numerous studies have recently been undertaken in the CCFZ and are still going on. Some have been published recently but there is still a lot come. Our knowledge of benthic biodiversity in the CCFZ is going to significantly increase in the years to come.*

What are the implications of theses results for mining operations and their regulation? Do we need more research to understand whether the estimated spatial range is really true or just mirrors the inappropriate sampling effort available scientific knowledge is based upon? Or do we 'know enough' and could provide specific suggestions as for how to spatially arrange mined patches? Taking this further: if we take the precautionary approach seriously: if we have an average species range of 20km (and, for some of the species obviously a smaller one) wouldn't we need to restrict the mining operations by contractors including secondary impacts by the plume to that size until it is proven, that the high turnover of beta diversity is an artifact of undersampling?

*Indeed, if the species ranges are narrow AND if the environmental objectives are to avoid species extinction then the spatial footprint of mining impact would need to be severely restricted. This is the meaning of the sentence "If true, the risk of species extinction is very high because the environmental footprint of nodule mining would largely exceed the range of many species". "If true" in this sentence refers to the fact that we cannot yet exclude that the average species range that we have estimated is biased by singletons. Thus, with the data we have we can't provide specific suggestions regarding the spatial arrangement of mining. That's what we meant by "The assessment of potential risks and scales of biodiversity loss thus requires an appropriate inventory of species richness in the CCFZ."*

*We further underlined the uncertainty regarding species range in the conclusion, page 16 lines 31-32:*

*"non-parametric estimators of species richness suggest that total species richness across the five study areas does not exceed 498 species which likely implies a species range much larger than 25 km."*

Page 15, line 27/28 'The assessment of potential risks and scales of biodiversity loss thus requires an appropriate inventory of species richness in the CCFZ.' While the conclusions are basically a summary up to this point, this is where the discussion in implications starts: How should this goal should be achieved?

*As outlined above we can suggest 1) a stratified random sampling at nested scales, from region down to seascapes and 2) an intensive sampling of selected habitats up to the point where the number of singletons decreases with sample size. We agree that these general recommendations would need to be more specific. There is a need to carefully think the sampling design and sampling effort together with statisticians. This would be a topic for another paper.*

*We can already say is that it won't be cheap. And we can also paraphrase Coddington et al. (2009) here and share their hopes "we suggest that inventory analyses continue to assess undersampling bias in order to justify the budgets required to obtain adequate data. Funding sources and consumers of these essential data can scarcely argue that inadequate results are acceptable. If results continue to demonstrate that much greater sampling intensities are required, such will eventually become the norm, rather than the exception."*

*The following lines were added in the Conclusions (page 17 lines 2-5):*

*In the framework of an ambitious and collective effort to inventory species richness in the CCFZ, a stratified random sampling at nested scales, from region down to seascapes, would provide the scales of species turn-over while intensive sampling of selected habitats up to the point where the number of singletons decreases with sample size would provide accurate estimates of species diversity.*

How much of this work is, according to the knowledge of the authors, already done by baseline work of the different contractors and just needs metaanalysis of the pooled contractor's data (e.g., by an independent scientific consortium)? Or are samples and data lacking and more dedicated sampling campaigns needed? What effort would this take? Or - if that is too hard to estimate, how would you control if enough data are available (based on rarefaction curves? Based on biodiversity descriptors merging?)? If you compare this to what is found in the ISA regulations and guidelines: how does that compare?

*This is a good point. Significant progress has been made during the last 5 years. Stratified random sampling has been carried out in the framework of the European project MIDAS (e.g. Simon-Lledo et al., 2019a, 2019b for megafaunal communities) and contractors are producing a large amount of data on macrofaunal communities in the Eastern CCFZ (e.g BGR: Janssen et al., 2015; GSR: De Smet et al., 2017; UKSR: Glover et al, 2016). A meta-analysis is going to be conducted that should provide insight onto species richness and species ranges (https://www.isa.org.jm/news/deep-ccz-biodiversity-synthesis-workshop). By the end of this Deep CCZ Biodiversity Synthesis we should be able to tell where we collectively stand in terms of what we know and what we don't know. In order to provide an accurate estimate of species richness, we would look for a decreasing trend in the accumulation curve of singletons.*

What about the key-species concept? Could that become appropriated once the necessary knowledge was obtained or do you think we always have to cope with the full complexity when we want to address environmental impacts of deep-sea operations (assessment of the risks prior to operations, assessment of impacts happening during operations).

*We must stress here that there is a difference between studies aiming at assessing the potential risks of mining and studies aiming at monitoring the impact of mining (i.e. Environmental Impact Assessment, EIA). The key species concept would apply to the EIA, which is beyond the scope of our study.*

OTHER MAJOR ISSUES

Page line 3-27 The discussion of meiofauna in nodule areas comes as a bit of a surprise in the context of this paper, that does only provide data on macrofauna. If you want to leave this so prominent and detailed, you should first state what the data show. Does the Pape et al. study provides data from the same station so a quantitative comparison to other deep sea areas is possible? > In this case I would suggest to provide that quantitative information here. Otherwise consider reducing the discussion or move it to a less prominent part of the discussion. Maybe you could also connect it more to the paper, e.g. as an argument for focusing environmental impact studies on macro-fauna because they seem more relevant in terms of biomass and ecosystem function as in typical abyssal areas.

*Page 11 lines 8-22. All meiofauna discussion has been reduced as suggested:*

*From 'Food supply, sediment grain size and the density of nodules are the three main environmental factors that seem to drive the structure and composition of polychaete assemblages in the CCFZ.*

*Nodules have antagonistic influences on different size groups of benthic communities. Meiofaunal assemblages are less abundant in nodule-rich than in nodule-free sediments (Miljutina et al., 2010; Pape et al., 2018). Nodules however increase habitat heterogeneity, providing hard substrate for sessile organisms and generally enhancing the standing stocks of both sessile and vagile megafauna (Amon et al., 2016; Vanreusel et al., 2016; Simon-Lledó et al., 2019). Similarly, nodules seem to enhance macrofaunal density (De Smet et al., 2017) and diversity (Yu et al., 2018). Our results support the reported positive and significant relationship between polychaete abundance and nodule density (De Smet et al. (2017). The macrofauna in nodule fields may benefit from increased food supply and the release from competition with meiofauna. Nodules increase seafloor roughness, thereby increasing friction (Sternberg, 1970; Boudreau and Scott, 1978) and potentially sediment deposition rates. The large sessile suspension feeders may similarly enhance biodeposition (Graf and Rosenberg, 1997). Both processes may stabilize sediments and increase organic carbon supply as tube lawns do, for example (Michael et al., 2000). An increase in food supply may explain the higher densities of polychaetes in nodule-rich areas. The divergent response of meiofauna to the presence of nodules further suggests some sort of competition between meiofauna and macrofauna. The contribution of meiofauna to benthic biomass generally increases along a bathymetric gradient to outweigh that of macrofauna at abyssal depths (Thiel, 1975; Rex et al., 2006; Wei et al., 2010). This pattern is assumed to reflect a selective advantage for small size at very low levels of food input (Thiel, 1975, 1979; Sebens, 1982, 1987; Rex and Etter, 1998). Sibuet et al. (1989) reported however a linear relationship between meiofaunal and macrofaunal biomass at abyssal sites. Both size classes indeed co-varied with organic carbon burial flux, which suggests the occurrence of a dynamic equilibrium between meiofauna and macrofauna at abyssal depths. Due to its small size, meiofauna is likely more efficient at exploiting the low level of food input, but this interstitial fauna may also be more sensitive to high nodule coverage because its ambit is largely limited to superficial sediments. The opposite effects of nodule coverage on meiofaunal and*

*macrofaunal densities may thus lie in a release from the advantage of being smaller in the abyss, inducing a shift in size-group equilibrium toward increased macrofaunal densities. These results suggest that nodule coverage have an influence on the functioning of the ecosystem, because it modifies biotic interactions and resource allocation among functional groups.'*

*To 'Food supply, sediment grain size and the density of nodules are the three main environmental factors that seem to drive the structure and composition of polychaete assemblages in the CCFZ.*

*The abundance and richness of polychaetes were positively correlated with nodule density, which is consistent with previous studies showing that nodules enhance macrofaunal densities and polychaete diversity (De Smet et al., 2017; Yu et al., 2018). Nodules may have antagonistic influences on different size groups of benthic communities. Meiofaunal assemblages are less abundant in nodule-rich than in nodule-free sediments, which may be due to the lower volume of sediment available in nodule areas (Miljutina et al., 2010; Hauquier et al. 2019). In our study, the volume and surface occupied by nodules were not quantified but the positive relationship between nodule density and polychaete abundance shows that space is not a limiting factor for polychaetes. Nodules also increase habitat heterogeneity, providing hard substrate for sessile organisms and generally enhancing the standing stocks of both sessile and vagile megafauna (Amon et al., 2016; Vanreusel et al., 2016; Simon-Lledó et al., 2019).  Nodules increase seafloor roughness, thereby increasing friction (Sternberg, 1970; Boudreau and Scott, 1978) and potentially sediment deposition rates. The large sessile suspension feeders may similarly enhance biodeposition (Graf and Rosenberg, 1997). Both processes may decelerate water current, stabilizing sediments and, thus, increase organic carbon supply as polychaete tube lawns do, for example (Friedrichs et al., 2000). An increase in food supply may explain the higher densities of polychaetes in nodule-rich areas.'*

Connected to this: Page 10, line 16 'The contribution of meiofauna to benthic biomass generally increases along a bathymetric gradient [. . .] which suggests the occurrence of a dynamic equilibrium between meiofauna and macrofauna at abyssal depths.' Again - this is very detailed for a study that does not focus on size class comparisons. I assume the Sibuet paper focuses on non-nodule areas? I understand you want to put forward that macrofauna is particularly important in the CCZ / in nodule areas as they – different to what was previously reported - show but a relative increase as compared to Meiofauna (i.e., neither do they show a relative decrease at depth as compared to meiofauna nor do they scale with meiofauna). This really would need a quantitative basis, i.e., a comparison of macrofauna abundances (better biomass) to meiofauna abundances at your study sites relative to other areas. > consider adding more quantitative information

*This has been deleted following previous comment.*

Connected to this: Page 10, line 23 'Due to its small size, meiofauna is likely more efficient at exploiting the low level of food input, but this interstitial fauna may also be more sensitive to high nodule coverage because its ambit is largely limited to superficial sediments.' Do you mean, that the meiofauna is restricted to the top layer where the available sediment volume is limited by the presence of nodules? I think you dont show the data but I assume that also for polychaetes the top sediment layer is the one where most individuals are found. In any way you could strengthen this idea by comparing your abundance vs. depth

relationship with that of meiofauna in nodule areas. > please comment, explain and consider including this information in the manuscript

*This has been deleted as previous comment.*

Page 14, line 7/8 'Overall, the combination of high local diversity, unsaturated rarefaction curves, high levels of cryptic diversity and high rates of species turnover suggest that polychaete diversity in the CCFZ is large and vastly under-sampled.' It needs a discussion of the most appropriate technologies (sampling gear, analysis) and the expected effort it needs to raise our knowledge to a level appropriate to decide on mining (yes or no, spatial organization of operations and protected areas), and allow for scientifically sound impact assessment and management

*Differences in sampling gear, type of preservation and how the diversity estimators consider the large presence of singletons were discussed along the Discussion. The expected efforts needed to reach an appropriate biodiversity assessment was suggested in the Conclusions with the following sentences:*

*Page 16 lines 20-24:*

*'The sampling effort in both the contract areas and the APEI however remains quite limited. In order to ascertain that the APEIs collectively meet their goal of preserving the biodiversity of the CCFZ an ambitious research agenda is needed, the funding of which could rely on the willingness of contractors and Sponsoring States but could also become a priority of the future Environmental Compensation Fund to be created by the regulations on exploitation of mineral resources in the Area (ISBA/25/C/WP.1).'*

*And page 17 line 2-5*

*'In the framework of an ambitious and collective effort to inventory species richness in the CCFZ, a stratified random sampling at nested scales, from region down to seascapes, would provide the scales of species turn-over while intensive sampling of selected habitats up to the point where the number of singletons decreases with sample size would provide accurate estimates of species diversity. Both strategies are needed to assess the potential risks and scales of biodiversity loss due to nodule mining in the CCFZ'.*

Maybe in this context it should also be discussed, if (and why!) the authors believe, that polychaetes may serve as a model group for baseline and impact assessments. Or is this just the 'pet group' of the authors and any other group should be similarly addressed before taking decisions? > please extend discussions to include these points.

*Polychaetes are the most abundant and most diverse among the macrofauna. Polychaete might however be less threatened than peracarids, which are brooders and show narrower species ranges on average. Polychaetes might also be less functionally important than nematodes, which dominate the metazoan biomass or foraminifera, off which we know very little. If the aim is to monitor and preserve all levels of biological diversity, from gene, to species, to functions then polychaetes are likely not enough.*

*The following text was added in aim explaining the relevance of studying polychaetes:*

*Page 4  line 6-10*

*'To tackle these issues [study aims], we focused on polychaete assemblages. Polychaetes are the dominant and most diverse group of the macrofauna; they can be quantitatively*

*sampled, and identify down to species level using a combination of morphological and molecular methods (Hessler and Jumars, 1974; Janssen et al., 2015; Wilson, 2017). Polychaetes also show a wide range of biological traits, from trophic behaviors to life history strategies, and play a major role in the functioning of benthic communities (Hutchings, 1998; Jumars et al., 2015).'*

MINOR ISSUES

Page 1, line 1-3 Including the metaanalysis performed the study indeed addresses the entire CCZ. However, the stations of this study are all rather in the eastern part. > rephrase the title to not raise false expectations, e.g., by replacing 'across the nodule province of the Clarion-Clipperton Fracture Zone' with add 'across the nodule province of the eastern Clarion-Clipperton Fracture Zone'

*Page 1 lines 1-3. Done as suggested.*

Page 1, line 17 '. . .the SO239 cruise aimed at improving species inventories. . .'. Was this really the subject of the cruise as a whole or of this expedition? > consider rephrasing

*Page 1 line 17. This has been changed to '. . . the SO239 cruise **provided data to improve** species inventories.'*

Page 2, line 6 'Only about 1 % of abyssal plains have been explored to date'. In this context of this paper I would restrict the use of the term 'explore' / 'exploration' to deep- sea mining-related activities > consider rewording

*Page 2 line 8. This has been changed to 'Only about 1 % of abyssal plains have been **investigated** to date…'*

Page 2, line 7 'In particular' seems to connect to the previous sentence but in fact does not. > consider remove

*Page 2 line 9. Done as suggested (removed).*

Page 2, line 9 '. . .mainly manganese and iron,. . .' > I would also mention copper, nickel and cobalt right away here - than you don't have to repeat that in line 13/14.

*Page 2 lines 10-11. Done as suggested. It has been changed to '…, mainly manganese and iron **but also copper, nickel and cobalt …'***

*The repeated words were removed.*

Page 2, line 15/16 '. . .the International Seabed Authority [. . .] is in charge of protecting fauna against any pollution or other hazards...' Pollution is not the main concern in the context of nodule mining and expected impacts related to this study. > I would rephrase. Maybe just refer to harm (i.e., 'protecting fauna agains harm')?

*Page 2 line 19. Done as suggested.*
*It has been changed to '. . .the International Seabed Authority [. . .] and is in charge of protecting fauna against any **harm**...'*

Page 2, line 25 I dont understand what is meant with 'scaling issue'. Is this referring to the uncertainties connected to effects of the full scale, long-term operations with large plumes as compared to single experimental tracks? > please rephrase / be more specific

*Page 2, line 31. This has been changed from 'Beyond the scaling issue…' to 'Beyond the **unpredictable effects of the full-scale mining**…'*

Page 2, line 27 > replace 'the high diversity' by 'a high diversity'

*Page 3 line 1. Done.*

Page 3, line 23 'test the hypotheses that support spatial conservation planning in the CCFZ'. I don't think that these hypotheses (that the authors think would sever as guidance or that form the basis of the current regional management plan are explicitly stated somewhere in the publication. > consider being more specific here or state them elsewhere in the paper

*Page 3 lines 20-22. The following sentence was added in Introduction: "One of the main assumptions underlying the management plan is that longitudinal and latitudinal productivity-driven gradients shape the community structure and species distribution of abyssal communities."*

*Furthermore, we have reworded the aims (page 4 lines 3-6):*

*From "The structure and composition of polychaete assemblages were analyzed to describe and identify alpha and beta diversity patterns, test the hypotheses that support spatial conservation planning in the CCFZ, assess the representativeness of an APEI and potentially improve the assessment of potential risks to biodiversity due to nodule mining."*

*To "The aims of our study were (a) **to test the hypotheses that support spatial conservation planning in the CCFZ, particularly the environmental drivers of alpha and beta diversity such as organic carbon fluxes to the seafloor and nodule density;** (b) to assess the representativeness of an APEI (i.e. APEI#3) and (c) to improve the assessment of potential risks of biodiversity loss due to nodule mining."*

Page 4, line 1/2 > replace '. . .were located between 4000 and 5000 m depth. . .' by 'had water depths between 4000 and 5000m'

*Page 4 lines 19-20. Done as suggested.*

Page 4, line 12/13 '. . .all nodules picked up from the sediment surface, washed and individually measured and weighed. . .' It should be mentioned already here that the water that was used for washing the nodules was sieved after washing. Have the nodules themselves been inspected for small polychaetes, e.g, living in tubes attached to the nodules? > rephrase and make sure to mention somewhere in the paper, if the data also include nodule-associated polychaetes

*Page 5 lines 8-10. This has been changed and the nodule-associated polychaetes clarified:*

*From 'The box core sample surface was photographed, and all nodules picked up from the sediment surface, washed and individually measured and weighed.'*

*To 'The box core sample surface was photographed, and all nodules picked up from the sediment surface, **washed with cold seawater over a 300 μm-mesh sieve** and individually weighed. **Sessile polychaetes, if present, remained attached to the nodules and were not considered in this study.**'*

Page 4, line 18 '. . .The sieve residues from the overlying water and the washed nodules were combined with all layers for the community analysis...' Was the material combined (i.e., before analysis) or the data? > specify in the text

*These layers (overlying water, the nodule washing water and the 0–3 cm layer) were combined after sieving, before sorting. It has been clarified in previous sentences (page 5 lines 13-14):*

*From 'The overlying water residue and the 0–3 cm layer were immediately sieved in the cold room with cold seawater (4 °C) and then live-sorted.'*

*To 'The sieve residues from **the overlying water and nodule washing were added to the 0-3 cm layer and live-sorted**."*

*The correspondent lines (page 5 line 16) cited by Referee #2 were changed:*

*From 'The sieve residues from the overlying water and the washed nodules were combined with all layers for the community analysis'*

*To 'All layers were combined for the community analysis.' making reference to all layers (overlaying water and washed nodules added to 0-3 cm; 3-5 and 5-10 cm).*

Page 4, line 20/21 '. . .(see Section 2.3 DNA extraction, amplification, sequencing, and alignment)' No need to refer to a section that follows directly > remove

*Done (removed).*

Page 4, line 24/25 '. . .and 1600 bp of 18S genes. . .' ? Are 18S data really used in this study (I could not find it later on)? If not: restrict M&M to 16S and COI or discuss why that approach was not successful or not included in the analyses.

*This is correct, the 18S data is not considered in the present study. However, we will available them concomitantly with the other genes. Because this, the methods used to amplify them must figure in the manuscript.*

*We have added (page 9 lines 17-19) the following sentence in the Results section to clarify: 'The 18S gene was sequenced for phylogenetic purposes on a restricted number of specimens. The 21 sequences of the 18S gene that have been obtained are mentioned here because they were archived concomitantly with COI and 16S sequences in GenBank and BOLD public datasets but they are not further considered in this study.'*

Page 5, line 13 'To separate closely related species...' [...] observed between intraspecific and interspecific variations' What does 'closely related species' mean? Specimen that could not be discriminated based on morphology? > specify Page 5, line 13-17 To separate closely related species [...] observed between intraspecific and interspecific variations' This section is describing the principle not what acutally was done. This does not fully qualify for a Materials and Methods part > Move to another part of the study (introduction?) or rephrase.

*Page 6 lines 11-12. This has been clarified with the addition of the suggested information and rephrasing:*

*From 'To separate closely related species, the principle of phylogenetic species was used, [...] observed between intraspecific and interspecific variations'*

*To 'We separated closely related **species (specimens that could not be discriminated morphologically**) using the principle of phylogenetic species, whereby the genetic divergence among specimens belonging to the same species (intraspecific) is smaller than the divergence among specimens from different species (interspecific) (Hebert et al., 2003b)'*

Page 5, line 25 '. . .to calculate nodule density. . .' Is nodule mean size or size distribution also considered in this study? If not, why was this not included as a parameter that may shape communities? > explain, consider adding explanation to the paper

*The nodules were weighted but not sized.*

*Page 5 lines 8-9. It was corrected:*

*From 'The box core sample surface was photographed, and all nodules picked up from the sediment surface, washed and individually measured and weighed.'*

*To 'The box core sample surface was photographed, and all nodules picked up from the sediment surface, washed with cold seawater over a 300 µm-mesh sieve and individually weighed.'*

Page 5, line 25-29 'Particulate organic carbon flux (POC, mg C m−2 d−1) at the seafloor for our study areas [. . .] applying the Suess algorithm (POC at the seafloor as a function of the net primary production scaled by depth; Suess, 1980; Table 2).' How do POC fluxes estimated with different methods compare where they overlap (i.e. in the study area?) > consider adding that information to the paper.

*Spearman correlation shows a significant correlation (rho 0.90, p < 0.05). This was included in the Figure 3.*

Page 6, line 1 '2.6 Regional-scale data' Also the Ocean Productivity-based POC fluxes in the previous section refer to the regional scale > choose another headline, e.g., 'Regional scale polychaete community data'

*Page 7 line 1. Done as suggested but also modified in order to clarify the range of the meta-analysis: '2.6 NE Pacific-scale polychaete community data'*

Page 6, line 6 > add references for ES163 and bootstrap

*References of the papers providing ES163 and bootstrap are given in the text (page 7 line 5) as well as in Table 2. Reference of the paper describing bootstrap are given in the text (page 7 line 19).*

*Furthermore, ES163 has been explained in the text with the addition of the following sentence:*

*Page 7 lines 14-15 'Based on these data the expected number of species was calculated for 12 individuals (ES12) and 163 individuals (ES163); as well as for three samples (S3).'*

Page 6, line 20/21 'Spearman correlations were sought between biotic and abiotic variables, using data from the SO239 cruise in the CCFZ and data compiled from the literature.' The data used for these correlations should match the data sources listed in section 2.5 > to avoid confusion I suggest to just refer to section 2.5. here. If the 'biotic and abiotic variables' include data not mentioned in section 2.5 add them there.

*Figure 3 (below). The variables POC flux at seafloor from Volz et al. (2018) and the POC flux at seafloor estimated in the present study were included in the revised Figure 3 as POC Eastern and POC NE Pacific, respectively. The following sentence was added in the caption 'POC Eastern values provided by Volz et al. (2018); POC NE Pacific values were estimated in the present study.'*

*Moreover, in order to be clear, the figure was separated in (a) when having data from each box-core sample such abundance, richness and nodule density (n=30); and (b) when having data averaged by area (n=5) such clay, silt, TN, TOC, CPE, POC Eastern and POC NE Pacific, and averaged abundance, richness and nodule density.*

*The corresponding caption was changed from 'Figure 3. Correlation matrix between biotic and abiotic variables from sampled areas within the eastern CCFZ. Diagonal panels show the distribution frequency of values for each variable. Below-the-diagonal panels show the correlation plot between pairs of variables. Above-the-diagonal panels show the Spearman coefficient correlations between pairs of variables. "*" indicates p < 0.05, "**" p < 0.01 and "***" p < 0.001.'*

*To 'Figure 3. Correlation matrix between biotic and abiotic variables from sampled areas within the eastern CCFZ. Diagonal panels show the distribution frequency of values for each variable. Below-the-diagonal panels show the correlation plot between pairs of variables. Above-the-diagonal panels show the Spearman coefficient correlations between pairs of variables. **Abundance, richness and nodule density per box-core (a) and average biotic and abiotic variables per area (b).  POC Eastern values provided by Volz et al. (2018); POC NE Pacific values were estimated in the present study**. "*" indicates p < 0.05, "**" p < 0.01 and "***" p < 0.001.'*

[Figure]

Page 6, line 23 to Page 7, line 14 Also in this section it should be described what has been done while a description of how the methods work does not seem appropriate for the M&M section (e.g., ' Low values of m give a high weight to dominant species, high values of m give a high weight to rare species.'). > Rephrase, possibly move parts to other sections

*The NNESS and CNESS distances are not commonly used in community ecology although they have interesting properties, in particular to rationalize the weight given to abundant and rare species rather than the arbitrary choice of a data transformation (e.g square root, double square root…). For this reason we think that it is worth providing some information on the use of these metrics in the M&M section.*

Page 7, line 21 '…tended to decrease from east to west with high spatial variation' 1. the main axes does not seem to go strictly longitudinal > replace 'east to west' by 'southeast to

northwest' 2. 'high spatial variation' would make more sense in a study design, that follows a clear geographical transect. > consider rephrasing, e.g., 'high variability between neighboring areas'.

*Page 8 line 25-26. Done as suggested.*

Page 8, line 2 'The relative contributions of trophic guilds also varied among the areas...' Is there an explanation found somewhere, how trophic guilds were determined? > If not, add description and references to M&M.

*The trophic guilds were defined based on literature (Jumars et al., 2015).*

*The following phrase was included in the section 2.4 Operational taxonomic units (OTUs) (page 6 lines 19-20): 'Trophic guilds were determined following Jumars et al. (2015) at family level.'*

*Consequently, the section name (page 6, line 8) has been changed to '2.4 Taxonomic identification and **feeding guilds classification'***

*The changing from "Operational taxonomic units (OTUs)…" to "Taxonomic identification…" followed a comment of Referee #4.*

Page 8, line 6 'Off the 1223 polychaetes, 1118 specimens belonging to 78 possible genera within 40 families were identified down to Morphospecies. . .' What are 'possible genera'? > consider rewording, e.g., '. . .possibly belonging to 78 genera. . .'?

*This makes reference to possible new genera. It was corrected to valid genera only. Also, we have corrected the number of sampled polychaetes which is 1233 instead of 1223.*

*This was changed (page 9 line 11) to 'Off the **1233** polychaetes, 1118 specimens belonging to **62 genera** within 40 families were identified down to morphospecies.'*

Page 8, line 6/7 '1118 specimens [...] were identified down to morphospecies (see Section Data availability)' Not sure why you refer to that section here. > please provide explanation and consider including it in the text.

*This has been removed.*

Page 8, line 14 'The mean number of species tended to decrease from east to west with high spatial variation. . .' see comment above (regarding Page 7, line 21, second comment)

*Page 9 lines 20-21. Done as suggested.*

Page 10, line 13/14 'Both processes [i.e., increased friction and sediment deposition / biodeposition rates] may stabilize sediments and increase organic carbon supply as tube lawns do' I dont see the connection to sediment stability. > please explain better what your idea is here

*It has been showed (Graf and Rosenberg, 1997; Friedrichs et al., 2000) that biological structures in the sediment-water interface favored the biodeposition and avoid erosion (e.g. polychaete tube lawns) by deceleration of water flow leading to a possible increase in food supply.*

*Page 11 lines 20-21. It has been rephrased to 'Both processes **may decelerate water current stabilizing sediments and, thus,** increase organic carbon supply as **polychaete** tube lawns do'.*

*Also, the correspondent reference was incorrectly written (names in the place of surnames), we have changed to 'Friedrichs et al., 2000'*

Page 10, line 15/16 'The divergent response of meiofauna to the presence of nodules further suggests some sort of competition between meiofauna and macrofauna.' I can see that - if nodules increase food supply but meiofauna abundances are relatively small, meifauna may be unable to make full use of the additional food. What I don't understand is why the reason does need to involve competition with macrofauna (see also my major comment on the meiofauna discussion above). > please provide explanation and consider including it in the text.

*This has been deleted as suggested.*

Page 11, line 34/35 'No significant correlation was however found between alpha diversity and productivity, neither at the NE Pacific scale nor at the scale of the whole CCFZ.' Do the authors have a hypothesis why this can be the case? Could it be related to the fact that most of the tested areas lie within more or less similar mesotrophic conditions and that this 'biased' data set is not fully appropriate to address this question? > please consider discussing the reason for the missing significant correlation of diversity and productivity on larger scales.

*Sorry but the sentence was not completely right. We have changed (page 12 lines 30-31) to:*

*"Species richness and productivity were significantly correlated at Eastern CCFZ scale, but no significant correlation was found between alpha diversity and productivity in the meta-analysis at the scale of the NE Pacific".*

*Furthermore, we believe that the missing significant correlation between diversity estimators and productivity at whole CCFZ scale is mostly due the differences in methods, in particular integrative vs. morphological taxonomy.*

*The following phrase was added (page 12 lines 31-32):*

*"The reason diversity and productivity were not correlated in the meta-analysis that included data from the literature could be mainly methodological. In particular, the use of integrative taxonomy in this study versus morphological taxonomy in previous works might hinder comparisons of diversity metrics."*

Page 12, line 5/6 'The fact that the APEI#3 lies mostly north of the Clarion Fracture Zone may however also contribute to its dissimilarity with the areas located in the CCFZ per se.' This statement reads quite vague as the idea of geographical barriers is not mentioned and elaborated before the next section > please consider adding (see next section) after the statement.

*This has been deleted because was vague and the next section will better discuss it.*

Page 12, line 12/13 '...characterized by a peak and through ...' Typo > change 'through' to 'trough'

*Page 13 line 14. Done as suggested.*

Page 12, line 24-26 'However, species identification was based on morphology only, although cryptic species are common among scavenging amphipods, even in abyssal lineages (Melo, 2004; Havermans et al., 2013)' Another reason is, of course, that scavenging

amphipods are typically highly motile. > consider adding mobility as an argument why scavenging amphipode distribution is not limited by fracture zones.

*Page 12 lines 27-28. It has been changed:*

*From 'In the abyssal Pacific, the CCFZ and the Peru Basin share nine species of scavenging amphipods (Patel et al. (2018), which thus potentially cross the Clipperton and Galapagos Fracture Zones''*

*To 'In the abyssal Pacific, the CCFZ and the Peru Basin share nine species of scavenging amphipods (Patel et al. (2018), which are **highly motile** and thus potentially cross the Clipperton and Galapagos Fracture Zones'*

Page 13, line 5 'In other words, nodule mining would affect each year an area that is equivalent to the average geographic range of a polychaete species.' This sounds like one mining operation would lead to the extinction of one polychaete ('only' - as some may argue). > consider removing 'a', i.e., write 'equivalent to the average geographic range of polychaete species.. . .'

*Page 14 line 8-10. Done as suggested.*

Page 13, line 27/28 '...suggesting that such extreme environmental conditions...' I don't share the view that the deep sea is per se an extreme environment. > replace 'such extreme' with 'the specific' or explain what specifically is considered extreme

*Page 14 lines 31-32. Done as suggested.*

Page 14, line1/2 'This highlights a shortcoming of COI-based barcoding because success rates for COI sequencing are generally low...' ? Are current molecular approaches appropriate if only are relatively small proportion could be identified based on 16S and COI and even less with both? Where is the problem and can it be overcome? If there new promising methods that base on other regions of the genome: how can we safeguard comparability of the full data set including new and older data?

*The current molecular approach using COI and 16S genes has proved to be appropriate in delineating species (e.g., Carr et al., 2011), but the sequencing success, especially for COI, is low. The reasons for failure can be numerous from bad DNA preservation to inappropriate DNA primers, annealing temperatures, etc…To some point, the only way to overcome the problem is to invest more time and efforts to get DNA sequences out of reluctant samples. For now, the most parsimonious method in our opinion is to associate morphology and DNA.*

*Page 15 lines 5-7. It has been changed:*

*From 'This highlights a shortcoming of COI-based barcoding because success rates for COI sequencing are generally low and a combination of several genetic markers plus morphology is essential to accurately assess species diversity.'*

*To 'This highlights a shortcoming of COI-based barcoding because success rates for COI sequencing are generally low. A combination of **several genetic markers associated to formal morphological descriptions are thus essential** to accurately assess species diversity.'*

Page 15, line 2-4 'The latter estimate assumes that we have sampled 0.1 % of the polychaete species in the CCFZ and that these species have narrow geographical ranges about the size of a yearly mined area.' If I understand right, this refers to the expected annual area exploited as part of one mining operation – not the total annually mined area >

replace 'a yearly mined area' with 'the area that will presumably mined in one year by a single mining operation'.

*Page 16 line 9. This has been changed.*

Page 31, Fig. 3 Irrespective of the fact that the variables are provided in the diagonal panels I would prefer if to the side of the plot the variables would be indicated like in https://images.app.goo.gl/oFQRE6xD7fvFwxJR6

*Done as suggested, please see the revised figure 3 below:*

[Figure]

Page 32, Fig. 4 '...in relation to the 2002–2018 average particulate organic carbon (POC) concentration at the seafloor along the CCFZ. The background map shows average POC flux

at the seafloor during the 2002–2018 period.' How can the maps show relations to POC concentration and flux at the same time? > consider rephrasing the caption. The caption should also state that this shows / includes data from published studies and refer to section 2.6

*Page 38 Figure 4. This has been rephrased:*

[revised manuscript text omitted]

---

## Author Comment (AC3) · 12 Nov 2019

Referee comments in black.

*Author's responses in green.*

A well written manuscript and a huge contribution of data for an area with limited but increasing data. This will be a useful resource for other researchers and deep-sea management in the region.

A few minor comments to accompany the comments on the attached manuscript. To me the hypotheses were not clear, I think the manuscript would really benefit if they were clearly defined in the introduction and revisited in the conclusions. An overview of why benthic diversity is important to a broader audience would be useful for the bigger picture as well as the ecological role of polychaetes within benthic communities. Why people should care about them? I love polychaetes but a lot of people don't. A brief description of the mining process, not all readers will be aware of this may be with a comment on the current likelihood of these operations happening and if so when. As stated the diversity estimates are very different, is there an additional method that can be used? With two measures that are both biased towards "singletons" is there a method that is not? Some images of the polychaetes would be nice, especially as photography was an important part of the method. These can be really useful for other research groups, will these be made publicly available? Maybe include a plate in the methods or results. I am not 100% familiar with the diversity analysis so can not critically comment on the methods/results for those sections. I leave this to the other reviewer and the editor. Many terms are not clearly defined but really important as often misinterpreted between papers. Table 1 could be supplementary.

*In general lines we have clarified the hypothesis in the introduction and aims; added why benthic diversity and polychaetes are important also in Introduction and Conclusions; and added some words in Introduction about mining operation. Unfortunately, the under-sampling and the singletons bias affects all current diversity estimators. A plate with photographs of polychaetes will not help with any aim of the manuscript. Pictures of polychaetes will eventually be available in BOLD database. The table 1 contains important data and we leave it integrated in the manuscript.*

*Below you can find the answers to each comment.*

Page 1, line 9. I felt that the abstract was very long... down to the editor but I think it could be shortened to really highlight the most important findings.

*We generally appreciate abstracts that are informative about the context, the methods, and the main results, in addition to the main findings. We would thus prefer to not shorten it.*

Page 1, line 14, 'unknown'. This counteracts the previous sentence that they were designated on best knowledge.

*It is indeed the paradox that we wished to emphasize by specifying that the design of the APEIs was based on "the best –albeit very limited – scientific knowledge"*

*We rephrased the sentence to make it clear (page 1, lines 13-14):*

*From "The APEIs were created based on the best – albeit very limited – scientific knowledge for the area."*

*To "The scientific principles for the design of the APEIs were based on the best – albeit very limited – knowledge for the area."*

Page 1, line 23, 'singletons'. This term is used a lot and I don't think it is defined. It is not always clear if it is a single specimen from an MOTU or morpho species.

*Singletons are species known from only one specimen, or MOTUs known from a single sequence.*

*The text has been modified to define "singleton" where it first appears in the Introduction (page 3, line 12):*

*From "(b) high frequencies of singletons ranging… "*

*To "(b) high frequencies of singletons **(MOTUs known from a single unique DNA sequence)** ranging …"*

*And later on, in the Results section (page 10, line 32-33):*

*From "Of these, 134 species were singletons."*

*To "Of these, 134 species were **singletons (i.e. morphospecies known from a single specimen)**."*

Page 1, line 23, 'food fluxes'. This was measured indirectly, I'd suggest changing it to represent what was actually measured or explain it is a proxy.

*Page 1, lines 23-24. We changed "food fluxes" by "organic carbon fluxes" in order to precise the variable used. We didn't get into the details of the methods here in order to not weigh down the abstract, which is already long.*

*From "The patterns in community structure and composition were mainly attributed to variations in food fluxes at the regional scale and nodule density at the local scale."*

*To "The patterns in community structure and composition were mainly attributed to variations in **organic carbon fluxes** to the seafloor at the regional scale and nodule density at the local scale"*

Page 1, line 24, 'regional scale and nodule density at local scale'. Is there a quantitative measure of scale or difference between the two?

*Page 1, line 24. The regional scale refers to the scale of the CCFZ and beyond, over 1000-km while, the local scale refers to within-area variations; NE Pacific-scale refers to all meta-analysis data covering the NE Pacific Basin. In order to keep the abstract as short as possible we didn't provided details here but we went through the text and provides explicit definition where needed.*

Page 2, line 1, 'reflect our level of uncertainty'. Can you comment using expert opinion about the levels you may expect?

*Page2, line 1. In the absence of a clear theoretical background to explain the maintenance of abyssal diversity and in the absence of empirical data to accurately quantify levels of diversity, there is unfortunately no information on which to form an expert opinion.*

Page 2, lines 6-7, 'Only about 1 % of abyssal plains have been explored to date: much remains to be discovered'. This is true but there is a lot of research effort trying to change

this. It would be nice to read that there is a positive trend in the number and resources for deep-sea studies.

*Page 2, line 8. "A lot of research" is somehow subjective. As far as the CCFZ is concerned, there has been 3 academic research cruises funded over the last 5 years. Whether this is a lot is debatable.*

Page 2, lines 9-10, 'possibly containing 34 billion metric tons of manganese (Michael et al., 2000; Morgan, 2000)'. Is there a monetary value for this?

*Page 2, lines 13-16. A word was missing in the that phrase. It has been changed 'possibly containing 34 billion metric tons of manganese **nodules** (Michael et al., 2000; Morgan, 2000)'. We also add a value of ore given by Volkmann et al. (2018, Table 1) to provide a monetary value (in bold):*

*"In the Equatorial Pacific Ocean, the Clarion-Clipperton Fracture Zone (CCFZ) harbors the largest polymetallic nodule field with nodule densities as high as 75 kg m$^{-2}$ (average 15 kg m$^{-2}$) and possibly containing 34 billion metric tons of manganese nodules (Hein and Petersen, 2013; Morgan, 2000), **which may represent a minimum sale value of 16 000 billion US Dollars (Volkmann et al, 2018)."***

Page 2, lines 21-22, 'Such small experiments however hardly mimic the cumulative impacts of any single nodule mining operation that could last for 20 years.'. Reference?

*Page 2, line 31. We have added the missing references:*

*Such small experiments however hardly mimic the cumulative impacts of any single nodule mining operation that could last for 20 years (**Glover and Smith, 2003; Jones et al, 2017**).*

Page 3, line 4, 'species turnover'. Please define.

*"Species turnover" has been defined in the first appearance in the manuscript (page 3, line 10-11):*

*From "(a) high rates of species turnover with only 12 % of polychaete…"*

*To "(a) high rates of species turnover **(i.e. species replacement)** with only 12 % of polychaete…"*

*And further defined in the Material and methods (page 8, line 15-16):*

*From "…turnover – which is dissimilarity due to species turnover"*

*To "turnover – which is dissimilarity due to species **replacement**"*

Page 3, line 6, 'richness'. Is this the same as cryptic species? sorry to get confused, these terms are widely used but often undefined. Would be good to be clear.

*Page 3, line 13. Indeed, "cryptic richness" was not correct, the appropriate term is "cryptic diversity" which indicates the presence of "cryptic species" (species morphologically indistinguishable).*

*It has been corrected all along the manuscript.*

Page 3, line 19, 'grounds'. Basis?

*Page 3 line 31. Done as suggested.*

Page 3, line 22, 'methods?'

*Page 4, line 8. Done as suggested.*

Page 3, lines 23-24, 'test the hypotheses that support spatial conservation planning in the CCFZ'. Can these be defined? They are mentioned again in the conclusions but I was still unsure where they are laid out

*We have deleted the correspondent sentence from the beginning of Conclusions as suggested by Referee #1.*

*We have clarified it in Introduction:*

*Page 3, lines 20-21. Addition of the sentence: "One of the main assumptions underlying the management plan is that longitudinal and latitudinal productivity-driven gradients shape the community structure and species distribution of abyssal communities.".*

*Page 4, lines 3-6. Changing the following sentence:*

*From "The structure and composition of polychaete assemblages were analyzed to describe and identify alpha and beta diversity patterns, test the hypotheses that support spatial conservation planning in the CCFZ, assess the representativeness of an APEI and potentially improve the assessment of potential risks to biodiversity due to nodule mining."*

*To "The aims of our study were (a)* **to test the hypotheses that support spatial conservation planning in the CCFZ, particularly the environmental drivers of alpha and beta diversity such as organic carbon fluxes to the seafloor and nodule density***; (b) to assess the representativeness of an APEI (i.e. APEI#3) and (c) to improve the assessment of potential risks of biodiversity loss due to nodule mining."*

Page 4, line 19, 'from each DNA-friendly polychaete'. Define DNA friendly? Ethanol fixed, whole?

*Yes. 'DNA friendly' and 'not DNA friendly' were already defined page 5, lines 14-16. The entire worms were fixed with ethanol and a piece of each one was dissected for molecular studies.*

Page 5, lines 7-9, 'All sequences obtained in this study have been deposited in BOLD (http://www.boldsystems.org; (Ratnasingham and Hebert, 2007)) or GenBank (http://www.ncbi.nlm.nih.gov/genbank/). Do you have the accession numbers for these? would be good to be in a table in the supplementary.

*Page 5, lines 18-20. The BOLD IDs and GenBank accession number are in the dataset available in Pangaea databases.*

*The following sentence has been modified in the section Data Availability (page 17, line 8-10):*

*From "Abundance data analyzed in the present study are available in the Pangaea (Bonifacio et al., 2019) whereas DNA sequences are available in BOLD or GenBank databases."*

*To "DNA sequences are available in BOLD (http://dx.doi.org/10.5883/DS-GKG001) or GenBank databases. The abundance data analyzed in the present study together with BOLD IDs (Sample ID and Process ID) and GenBank accession numbers are available in the PANGAEA database (Bonifácio et al., 2019)."*

Page 6, line 9, 'species'. Species or MOTUs?

*As pointed out by Referee #4 there was confusion in the use of terms such as morphospecies, MOTUs and morphotypes. Those were corrected in the revised version. Moreover, in general we consider morphospecies as species.*

Page 6, line 10, 'damaged species'. Was DNA not taken from fragmented specimens? Can this be mentioned in the earlier methods sections.

*It should be 'damaged specimens'. This has been corrected as suggested (page 7, line 11). DNA was taken from very few fragments but only head-ends were counted.*

*Page 5, lines 17-18. The following sentence was changed:*

*From "In the laboratory, from each DNA-friendly polychaete specimen, a small piece of tissue was dissected …"*

*To "In the laboratory, from each DNA-friendly polychaete specimen **and from very few fragments**, a small piece of tissue was dissected …"*

*Page 5, lines 18-19. The following sentence was added: "DNA sequences from fragments without head were archived in BOLD and GenBank (Bonifácio et al., 2019) but were not further used for the purpose of this paper."*

Page 7, line 2, 'forward, backward and stepwise selection procedures'. Can this be explained?

*Page 8, line 5. Roughly, the forward procedure adds each predictable variable one by one to the model until be statistically significant; the backward includes all variables in the model and removes one by one until be statistically significant); and stepwise is a combination of both forward and backward. Actually, the selection procedure used in the case was the "forward".*

*As requested by Referee #2 in Methods section it should be described what has been done while a description of how the methods work does not seem appropriate for the M&M section. Thus, a citation to the reference of Borcard et al., 2011 (and in References) who explain in detail the methods was added.*

*Borcard, D., Gillet, F., and Legendre, P. (Eds.): Numerical ecology with R, Springer-Verlag, New York, USA, 2011.*

Page 7, line 30, 'polychaetes', Of the 1233 polychaetes?

*Page 9, line 4. Yes. But it doesn't need to be repeated here.*

Page 8, lines 1-2, 'The relative contributions of trophic guilds also varied among the areas'. How were trophic guilds defined? Family level traits from the literatures or another study?

*The trophic guilds were defined based on literature (Jumars et al., 2015).*

*The following sentence was included in the section 2.4 Operational taxonomic units (OTUs) (page 6, lines 19-20): "Trophic guilds were determined following Jumars et al. (2015) at family level."*

*Consequently, the section (page 6, line 8) has been changed to "2.4 Taxonomic identification **and feeding guilds classification"***

*The changing from "Operational taxonomic units (OTUs)…" to "Taxonomic identification…" followed a comment of Referee #4.*

Page 8, lines 13-15, 'total of 275 morphospecies (i.e. OTUs) were recognized. The mean number of species tended to decrease from east to west with high spatial variation (Fig. 2b). Mean richness varied from 37 $\pm$ 10 taxa 0.25 m$^{-2}$ in BGR to 3 $\pm$ 2 taxa 0.25 m$^{-2}$ in APEI#3.' How do these numbers compare to other published CCZ studies? in terms of MOTUS by area?

*Page 9, lines 20-24. The overall species richness is discussed in Discussion section '4.3 How many polychaete species live in the CCFZ? The under-sampling bias' and shown in Table 2. However, this can not be a simple comparison because of differences in sampling effort sampling design and methodologies for species identifications.*

Page 9, line 2, 'forward selection procedure kept'. ??

*Page 10, line 6. Yes, the "forward selection procedure" selected a best set of variables explaining the variability of polychaete assemblages.*

Page 9, line 9, 'indeed'. not needed

*Removed as suggested.*

Page 9. Line 29, 'Aurospio sp. 249'. Is this a common trait for Aurospio... other examples/oceans where this genus dominates and is well distributed. e.g. https://doi.org/10.1111/j.1439-0485.2008.00265.x

*Page 10, line 18. Completely agree. This aspect of Aurospio is discussed in the submitted paper of Guggolz et al. entitled: High diversity and pan-oceanic distribution of deep-sea polychaetes: Prionospio and Aurospio (Annelida: Spionidae) in the Atlantic and Pacific Ocean*

*We have added the following sentence in order to show differences in distribution of some polychaete species (page 13, lines 23-25) : "… or species of Aurospio and Prionospio which could show pan-oceanic distribution (i.e., Pacific and Atlantic oceans; Guggolz et al., submitted)."*

Page 10, line 1, 'Discussion'. Some switching between abundance and density... may be clarify if different or if the same keep consistent

*We have changed "macrofaunal density" into "macrofaunal abundance". Further, we kept "abundance" when referring to numbers of animals and "density" when referring to numbers of nodules.*

Page 10, line 3, 'Food supply'. This was measured indirectly, may be mention as a proxy of or indicated by

*The following sentence was added in M&M (page 6, line 31): "POC flux at seafloor was considered as a proxy for food supply to benthic communities."*

Page 12, lines 5-6, 'The fact that the APEI#3 lies mostly north of the Clarion Fracture Zone may however also contribute to its dissimilarity with the areas located in the CCFZ per se.'. Can you suggest a more suitable APEI area? Would you expect other APEIs given their position to be better?

*The main issue with the placement of APEIs is that most of the area to be managed in between the two fracture zones has already been preempted to contract or reserved areas. Moreover, the knowledge on the environment and benthic communities of the APEIs is still*

*quite limited. Our recommendation is to first gain knowledge on these APEIs. This has been highlighted in the Conclusions (page 16, lines 18-24):*

*The scantiness of food supply and a barrier to dispersal may thus compromise the representativeness of APEI#3 and question its ability to meet its purpose of preserving the biodiversity from any of the contract areas considered in this study. The sampling effort in both the contract areas and the APEI however remains quite limited. In order to ascertain that the APEIs collectively meet their goal of preserving the biodiversity of the CCFZ an ambitious research agenda is needed, the funding of which could rely on the willingness of contractors and Sponsoring States but could also become a priority of the future Environmental Compensation Fund to be created by the regulations on exploitation of mineral resources in the Area (ISBA/25/C/WP.1).*

Page 12, line 28, 'infaunal brooders'. Can you comment on the likely reproductive traits of the polychaetes based on family level data? What proportion of the polychaetes in this study are likely to be brooders?

*Page 13, line 32. Reproductive strategies data are limited and sometimes with variation within a family. Also, in the deep-sea we don't know much more about these traits, so we prefer to do no suppositions about.*

Page 13, lines 4-5, 'In other words, nodule mining would affect each year an area that is equivalent to the average geographic range of a polychaete species.'. I feel this is really important but not clearly worded. Maybe restructure.

*This is potentially important because if true, species extinction due to nodule mining is very likely. We made it clearer in the Conclusions (page 16, lines 27-29):*

*"Species turnover is high with a minimum estimated rate of species change of 0.04 species $km^{-1}$, suggesting an average geographical range of 25 km and a number of polychaete species in the CCFZ that may equal the number of all currently known marine species. If true, the risk of species extinction is very high because the environmental footprint of nodule mining would largely exceed the range of many species."*

Page 13, lines 31-31, 'Over 90 % of the species in the abyssal Pacific are new to science (Glover et al., 2002) and there are few attempts to try to name them (Paterson et al., 2016; Bonifácio and Menot, 2019)'. Realistically there are not enough experts, time or resource to name species new to scienc :( but there are other approaches such as turbo taxonomy that could be mentioned as an alternative

Summers, M.M., Al-Hakim, I.I. and Rouse, G.W., 2014. Turbo-taxonomy: 21 new species of Myzostomida (Annelida). Zootaxa, 3873(4), pp.301-344.

*There are not enough experts in the world, but resources need to be invested in formation and hiring taxonomists. Turbo-taxonomy is not the ideal world neither. It is completely based on molecular approach that is not completely efficient currently; and it did not provide always a detailed morphological descriptions which should be used by ecologist. Morphological identification remains less expensive than barcoding. According to Stein et al. (2014), current barcoding costs using Sanger sequencing are between 1.7 and 3.4 times as expensive as traditional taxonomic approaches, excluding the cost of field sampling (which is common to both approaches). Taxonomy needs to be considered as a science and not a tool.*

Page 14, lines 28-29, 'In the deep-sea, an anomalously high rate of singletons of about one-third of the species is in fact the rule of macrofaunal surveys'. Can you clarify or reword?

*Page 15, lines 33-34. This was rephrased to "In the deep-sea, an anomalously high rate of singletons **(about one-third of the sampled species)** is **usually** the rule of macrofaunal surveys …"*

Page 15, lines 9-11, 'testing the hypotheses that supported spatial conservation planning in the CCFZ, assessing the representativeness of an APEI and improving the assessment of potential risk to biodiversity due to nodule mining.'. Can these be clearly laid out in the introduction and then revisited?

*As suggested the objectives have been clarified in the Introduction.*

*Page 4, lines 3-6:*

*"The aims of our study were (a) to test the hypotheses that support spatial conservation planning in the CCFZ, particularly the environmental drivers of alpha and beta diversity such as organic carbon fluxes to the seafloor and nodule density; (b) to assess the representativeness of an APEI (i.e. APEI#3) and (c) to improve the assessment of potential risks of biodiversity loss due to nodule mining."*

*And then revisited e.g.*

*Page 16, lines 11-13*

*"Food inputs and nodule density influence the structure and composition of polychaete assemblages in the CCFZ. This is a confirmation of hypotheses underpinning the design of the APEIs"*

Page 15, line 20, 'even higher'. % range?

*The range is between 498 and 240,000 species as stated page 16 lines 5-6:*

*"In conclusion, our level of certainty on the number of polychaete species inhabiting the CCFZ and potentially threatened by nodule mining ranges from 498 to 240,000 species."*

Page 15, line 21, 'cryptic richness'. diversity/species?

*This has been changed to "cryptic diversity" across the manuscript.*

Page 15, line 23, 'footprint of nodule mining'. Estimate of area?

*This was exemplified in the Discussion section '4.2 Species turnover and geographic ranges' Page 14, line 8  "… an area of a 100 km² would be mined each year."*

Page 15, lines 27-28, 'The assessment of potential risks and scales of biodiversity loss thus requires an appropriate inventory of species richness in the CCFZ.'. How close are we to achieving this? What would be required - international database or the use of pre-existing databases?

*The issue with pre-existing data and databases is that the species lists can't be merged as there is no formal description of species. The use of DNA barcoding and integrative taxonomy in recent years will progressively allow to facilitate data integration. Still, there is a need for a large and coordinated research program in order to fully assess the scales of biodiversity in the CCFZ. The following recommendation was added in the Conclusions (page 17, lines 2-5):*

*"In the framework of an ambitious and collective effort to inventory species richness in the CCFZ, a stratified random sampling at nested scales, from region down to seascapes, would provide the scales of species turn-over while intensive sampling of selected habitats up to the point where the number of singletons decreases with sample size would provide accurate estimates of species diversity."*

*References cited in the authors answers:*

*Bonifácio, P., Neal, L., Omnes, E., Baptiste, F., Dahlgren, T. G., and Menot, L.: The polychaete fauna of the Clarion-Clipperton Fracture Zone from boxcore samples during SONNE cruise SO239, PANGAEA, https://doi.pangaea.de/10.1594/PANGAEA.902860, 2019.*

*Borcard, D., Gillet, F., and Legendre, P. (Eds.): Numerical ecology with R, Springer-Verlag, New York, USA, 2011.*

*Guggolz, T., Meißner, K., Schwentner, M., Dahlgren, T.G., Wiklund, H., Bonifácio, P., and Brandt, A.: High diversity and pan-oceanic distribution of deep-sea polychaetes: Prionospio and Aurospio (Annelida: Spionidae) in the Atlantic and Pacific Ocean, Org. Divers. Evol., submitted.*

*Hein, J. R., and Petersen, S.: The geology of Manganese nodules, in: Deep Sea Minerals: Manganese Nodules, a physical, biological, environmental, and technical review, 1B, edited by: Baker, E., and Beaudoin, Y., Secretariat of the Pacific Community (SPC), Noumea, New Caledonia, 7-18, 2013.*

*Jumars, P. A., Dorgan, K. M., and Lindsay, S. M.: Diet of Worms Emended: An Update of Polychaete Feeding Guilds, Ann. Rev. Mar. Sci., 7, 497-520, doi:10.1146/annurev-marine-010814-020007, 2015.*

*Morgan, C. L.: Resource estimates of the Clarion-Clipperton manganese nodule deposits, in: Handbook of marine mineral deposits, edited by: Cronan, D. S., CRC Press, Boca Raton, 145-170, 2000.*

*Stein, E. D., Martinez, M. C., Stiles, S., Miller, P. E. and Zakharov, E. V.: Is DNA Barcoding Actually Cheaper and Faster than Traditional Morphological Methods: Results from a Survey of Freshwater Bioassessment Efforts in the United States?, edited by M. Casiraghi, PLoS ONE, 9(4), e95525, doi:10.1371/journal.pone.0095525, 2014.*

*Volkmann, S. E., and Lehnen, F.: Production key figures for planning the mining of manganese nodules, Mar. Georesour. Geotec., 36, 360-375, 10.1080/1064119X.2017.1319448, 2018.*

*Volkmann, S. E., Kuhn, T., and Lehnen, F.: A comprehensive approach for a techno-economic assessment of nodule mining in the deep sea, Mineral Economics, 31, 319-336, 10.1007/s13563-018-0143-1, 2018.*

---

## Author Comment (AC4) · 12 Nov 2019

Referee comments in black.

*Author's responses in green.*

The manuscript by Bonifacio and colleagues represents a relevant contribution to the study of polymetallic nodules fields. As the Authors state, it is necessary to understand ecological processes and diversity patterns occurring in these environments and to assess the impact of mining activities before starting with their exploitation, and in this frame, this manuscript is of great value. I would therefore endorse its publication in Biogeosciences. The manuscript is in my opinion clear and well-written. My only concern is represented by the use of the word "morphotype", as it is a somewhat ambiguous term. The most widespread use (at least, in my experience) regards morphotypes as divergent morphological variants within the same alleged species, pointing at either cryptic diversity or phenotypic plasticity. However, in this case it is employed to define individuals that morphologically can be assigned to the same taxon; if I correctly understood, the use of the term "species" or "taxa" has been avoided because molecular data often challenge this interpretation. I think however that in this case the best way would be the use of "morphospecies" or "taxa identified on the basis of morphological features". The term "morphospecies" is employed twice, in both cases at page 8. At line 7 the term is used in the same way of "morphotype" in the remaining manuscript, and as I would advise, but at line 13 the use of "morphospecies" is inappropriate, as here the Authors clearly refer to taxa identified by the combined morphological and molecular data. Morphospecies clearly do not correspond to OTUs (p. 8, line 13), and here the Authors are referring to OTUs. I suggest to carefully re-read the manuscript, as there is some terminological confusion around "morphotype", "morphospecies" and "OTU".

*Sorry about this confusion around "morphotype", "morphospecies" and "OTU". We have changed morphotype and MOTU terms to morphospecies with the addition of the following sentence (page 6, line 18-19):*

*'As genetic data was only used to separate closely related species, the delimited taxa entities in the present study are referenced as morphospecies.'*

*Morphotypes has been changed to morphospecies throughout the manuscript.*

*To be more clear the section "2.4 Operational taxonomic units (OTUs)" has been reworded to "2.4 **Taxonomic identification** and feeding guilds classification"*

Some minor comments follow.

P. 1, line 10: I suggest to add "environmental" to "footprint".

*Page 1, line 10. Done as suggested.*

P. 7, line 31: Replace "paranoids" with "paraonids".

*Page 9, line 4 Done as suggested.*

P. 11, line 18: "(Magalhães and Bailey-Brock (2017)": replace with "(Magalhães and Bailey-Brock, 2017)"

*Page 12, line 14. Done as suggested.*

P. 11, lines 27-28: Although all Eunicida are usually considered as carnivores, Jumars et al. (2015) suggest that the diet of Lumbrineridae might be more varied, and that sediment and decaying vegetal debris might represent an important diet component for several species. In particular, the reported characterising species belongs to the genus Lumbrinerides, a genus including small, possibly pedomorphic species that at least in shallow environments occupy an ecological niche totally different from larger species of Lumbrineris and Scoletoma. I think that in this case carnivory is not obvious.

*Indeed Jumars et al. (2015) considered lumbrinerids are carnivores with very few exceptions. In our case, we believe that the sampled lumbrinerids are carnivorous because no clear sediment was observed in their body. Furthermore, lumbrinerids follow the pattern observed for other unquestionably carnivorous such as sigalionids and paralacydoniids. In order to reinforce this idea, we add the following sentence page 12, lines 23-24:*

*"Furthermore, other carnivorous families were relatively more abundant in the eastern areas as well, such as paralacydoniids and sigalionids."*

P. 12, lines 20-21: results by Guggolz et al. (2018) have been partially ridiscussed in Guggolz et al. (2019: Scientific Reports 9: 9260). I suggest to check and cite also this work.

*The following sentence was added in order to cite Guggolz et al. (2019) and Guggolz et al. (submitted) in section 4.2 Species turnover and geographic ranges, page 13, lines 23-25:*

*"This was however not the case for species of* Laonice*, which tended to show large ranges of up to 4000 km across the Eastern and Western Atlantic (Guggolz et al., 2019); or species of* Aurospio *and* Prionospio *which could show pan-oceanic distribution (i.e., Pacific and Atlantic oceans; Guggolz et al., submitted)."*

*References cited in the authors answers:*

*Guggolz, T., Meißner, K., Schwentner, M., and Brandt, A.: Diversity and distribution of Laonice species (Annelida: Spionidae) in the tropical North Atlantic and Puerto Rico trench, Sci. Rep., 9:9260, 10.1038/s41598-019-45807-7, 2019.*

*Guggolz, T., Meißner, K., Schwentner, M., Dahlgren, T.G., Wiklund, H., Bonifácio, P., and Brandt, A.: High diversity and pan-oceanic distribution of deep-sea polychaetes: Prionospio and Aurospio (Annelida: Spionidae) in the Atlantic and Pacific Ocean, Org. Divers. Evol., submitted.*

*Jumars, P. A., Dorgan, K. M., and Lindsay, S. M.: Diet of Worms Emended: An Update of Polychaete Feeding Guilds, Ann. Rev. Mar. Sci., 7, 497-520, doi:10.1146/annurev-marine-010814-020007, 2015.*

---

## Author Comment (AC6) · 12 Nov 2019

Please find attached to this comment the revised manuscript.

Please also note the supplement to this comment:
https://www.biogeosciences-discuss.net/bg-2019-255/bg-2019-255-AC6-supplement.pdf

―――――――――――――――――

---

## Author Comment (AC8) · 12 Nov 2019

Please find attached to this comment the revised manuscript.

Please also note the supplement to this comment:
https://www.biogeosciences-discuss.net/bg-2019-255/bg-2019-255-AC8-supplement.pdf

---

## Referee Report (RR1)

Review of" Alpha and beta diversity patterns of polychaete assemblages across the nodule province of the eastern Clarion-Clipperton Fracture Zone (Equatorial Pacific)"
BG bg-2019-255

The manuscript is greatly improved on revision. The main findings, that almost half the polychaete species sampled in the nodule zone, of the Clarion-Clipperton Fracture Zone, were only represented by a single individual and only a single species was common to all five areas need to be widely reported.

It is still not entirely clear what happened to nodule-associated polychaetes. Epifauna visible on the surface were picked off before washing and sessile polychaetes still on the nodules after washing were later removed. It appears that these these were not included in the dataset but what was their percentage contribution to the total polychaete numbers. In addition it would be useful to know if the nodules were preserved for later dissection to extract their polychaete infauna. I think that it is important to give an indication of what fraction of the polychaete fauna is included in this paper.

Infauna are important in nutrient recycling and deeper-burrowing infauna are particularly important. While appreciating the difficulties in sampling these, the rare observations on them, such as the maldanid found at 50 cm depth, should be mentioned to show how little we know about the deeper abyssal infauna. In addition any observations on deep burrows would be of interest.

Table 1 lists the data for all the box core stations. The only actual result listed in the table is the "nodules density", actually the nodule wet weight, extrapolated to the weight per m$^{-2}$. Firstly this is not the density and secondly all the biological data is recorded per box core area, i.e. in 0.25 m$^{-2}$. The table should include the basic polychaete data per box core, i.e. total numbers and number of "species" so that readers can follow the author's analysis rather than having to extract the individual core data from the database in PANGEA.

The authors recognise that the estimated number of polychaete species and the average species range in the nodule province of the CCFZ are not well constrained as a result of the limited sampling possible on this cruise. It would therefore be helpful if the authors could discuss what sampling effort might be required to obtain reasonable estimates of these.

---

## Author Response (AR2)

Object:
Submission of a **revised** manuscript to Biogeosciences special issue *Assessing environmental impacts of deep-sea mining – revisiting decade-old benthic disturbances in Pacific nodule areas*

From Paulo,
To Editor and Associate Editor of the Biogeosciences

Brest, January 16th 2020.

Dear Dr. Treude,

We are pleased to submit our **revised** manuscript entitled "*Alpha and beta diversity patterns of polychaete assemblages across the nodule province of the eastern Clarion-Clipperton Fracture Zone (Equatorial Pacific)*" by P. Bonifácio, P. Martinez-Arbizu & L. Menot for consideration to be published in Biogeosciences special issue "Assessing environmental impacts of deep-sea mining – revisiting decade-old benthic disturbances in Pacific nodule areas".

We are glad that our corrections mostly pleased both the referees. We are thankful for the positive comments and minors' corrections suggested which were completely accepted in this version. Following all suggested modifications, our revised manuscript has been improved mainly with: the addition of abundance and number of species per box core at table 1; and developing important thoughts in the conclusions; but also detailing and correcting few typos, excluding exceeding references and calling the new columns added in table 1.

As solicited by you, below you can find the answers to each comment for the two referees who suggested minor revisions. Ours answers are in green and make references to the pages/lines of the revised manuscript (marked-up and submitted separately).

We thank you and we are looking forward to hearing from you.

Best regards,
Paulo Bonifácio, Pedro Martinez-Arbizu & Lénaick Menot

**Institut français de Recherche pour l'Exploitation de la Mer**

Etablissement public à caractère industriel et commercial

**Centre de Brest**
Zone Industrielle de la Pointe du Diable
CS10070
29280 Plouzané
France

téléphone   33 (0)2 98 22 40 40
télécopie   33 (0)2 98 22 45 45
              http://www.ifremer.fr

**Siège social**
155, rue Jean-Jacques Rousseau
92138 Issy-les-Moulineaux Cedex
France
R.C.S. Nanterre B 330 715 368
APE 7219Z
SIRET 330 715 368 00297
TVA FR 46 330 715 368

téléphone   33 (0)1 46 48 21 00
télécopie   33 (0)1 46 48 21 21
              http://www.ifremer.fr

*Author's response to Referee's comments*

Dear Authors,

I appreciate your careful consideration of all my questions and suggestions. Thanks a lot for doing a very thorough job here! I have no objections if the paper is published as is.

*We're glad to read that and we thank the referee for the substantial and important suggestions given.*

It seems to me, however, that some valuable thoughts are lost in the rather condensed edits you did to the manuscript. I have the impression that the paper would still benefit if some more content of your very detailed responses would also enter the text of the manuscript.

As I said - I accept if you leave as is but suggest that you to read through your answers and check whether some content could be transferred to the manuscript. Below I am quoting the answers where I had most strongly felt that important thoughts were not fully considered in your edits to the manuscript text.

"the main unknown is most likely about the biology and biotic interactions of species: how long do they live, how do they reproduce and disperse, do they interact and how are they interacting between others. These would be key questions to answer, although much more challenging than looking at correlations of abiotic factors and biological variables."
> This seems important information showing that the environmental-variable-approach underlying the regional management plan but maybe even studies on species diversity and turnover are in the end not sufficient to fully assess the risk and provide guidance if and how mining projects should be carried out.

*The following sentence was added in the Conclusions (page 17 lines 1 to 2):*

*"Furthermore, there are vast gaps in knowledge regarding the life cycle and population dynamics that would need to be better constrained to fully assess the risks and provide guidance in mining management.*

"If the aim is to monitor and preserve all levels of biological diversity, from gene, to species, to functions then polychaetes are likely not enough."
> You explained in the MS why you've chosen Polychaetes but the information that investigations (not only to preserve everything but also to fully understand the risks associated with mining) should consider all groups and size classes is not fully conveyed.

*Answered with next comment*

"We agree that these general recommendations would need to be more specific. There is a need to carefully think the sampling design and sampling effort together with statisticians. This would be a topic for another paper."
> To me this is an important point: recommendations on sampling design and effort for a specific region or site have to evolve from a scientific / statistical (and potentially iterative)

process and cannot be prescribed. Good to mention that future studies need to address this.

*We have changed and added both suggestions in the Conclusions (page 17 lines 20 to 26):*

*From "In the framework of an ambitious and collective effort to inventory species richness in the CCFZ, a stratified random sampling at nested scales, from region down to seascapes, would provide the scales of species turn-over while intensive sampling of selected habitats up to the point where the number of singletons decreases with sample size would provide accurate estimates of species diversity. Both strategies are needed to assess the potential risks and scales of biodiversity loss due to nodule mining in the CCFZ."*

*To "In the framework of a similarly ambitious and collective effort to inventory species richness in the CCFZ, a stratified random sampling at nested scales, from region down to seascapes, would provide the scales of species turn-over while intensive sampling of selected habitats up to the point where the number of singletons decreases with sample size would provide accurate estimates of species diversity. Both **strategies should consider different taxonomic and functional groups of the abyssal fauna, which are likely to show different responses to nodule mining. Such an approach, based on standardized sampling methods and statistical-wise sampling strategies** is needed to assess the potential risks and scales of biodiversity loss due to nodule mining in the CCFZ."*

"A meta-analysis is going to be conducted that should provide insight onto species richness and species ranges (https://www.isa.org.jm/news/deep-ccz-biodiversity-synthesis-workshop). By the end of this Deep CCZ Biodiversity Synthesis we should be able to tell where we collectively stand in terms of what we know and what we don't know. In order to provide an accurate estimate of species richness, we would look for a decreasing trend in the accumulation curve of singletons."
> the take home message to include in the MS may be that joint efforts combining data from independent science and contractors are needed to get to more accurate data on species richness and turnover - and that there are promising initiatives underway to get this started.

*The following sentence was added in the conclusions (page 17 lines 16 to 20):*

*"Under the auspice of the ISA, the synthesis of ongoing studies from independent science and contractors in the CCFZ will certainly contribute in filling some knowledge gaps on species richness and turn over but differences in objectives, strategies and methodologies among studies are also likely to put some limits on the usefulness of the exercise. The JPI Oceans pilot action "Ecological aspects of deep-sea mining" demonstrated how powerful such a joined and coordinated initiative can be."*

Review of" Alpha and beta diversity patterns of polychaete assemblages across the nodule province of the eastern Clarion-Clipperton Fracture Zone (Equatorial Pacific)" BG bg-2019-255

The manuscript is greatly improved on revision. The main findings, that almost half the polychaete species sampled in the nodule zone, of the Clarion-Clipperton Fracture Zone, were only represented by a single individual and only a single species was common to all five areas need to be widely reported.

*Thank you for the suggestions and we're glad that the improvements made are approved by you. The following sentence was included in the discussion section 4.2 Species turnover and geographic ranges (page 14 lines 2 to 4):*

*"Our observations about* Aurospio *sp. 249 which was the only species sampled in all five areas confirm the potential to disperse across large geographic distances of some spionids (Guggolz et al., in press)."*

It is still not entirely clear what happened to nodule-associated polychaetes. Epifauna visible on the surface were picked off before washing and sessile polychaetes still on the nodules after washing were later removed. It appears that these these were not included in the dataset but what was their percentage contribution to the total polychaete numbers. In addition it would be useful to know if the nodules were preserved for later dissection to extract their polychaete infauna. I think that it is important to give an indication of what fraction of the polychaete fauna is included in this paper.

*To our knowledge, the nodule epifauna, including polychaetes, has not been processed and the nodules were not preserved to study the nodule infauna. We thus can't provide an indication of the nodule-associated fauna. However, according to Thiel et al. (1993), the fraction of polychaetes found in nodule crevices has low significance and representativity.*

*We have changed (page 5 lines 9 to 13):*

*From "Sessile polychaetes, if present, remained attached to the nodules and were not considered in this study."*

*To "Sessile **and crevice-inhabitant** polychaetes, if present, remained **with the** nodules and were not considered in this study. **According to Thiel et al. (1993) who washed and broke 26 nodules, the fraction of crevice inhabitant polychaetes has low significance and representativity (i.e. only 29 specimens belonging to six species) when compared with those living in sediments surrounding the nodules (i.e. 864 polychaetes)."***

Infauna are important in nutrient recycling and deeper-burrowing infauna are particularly important. While appreciating the difficulties in sampling these, the rare observations on them, such as the maldanid found at 50 cm depth, should be mentioned to show how little we know about the deeper abyssal infauna. In addition any observations on deep burrows would be of interest.

*The following sentence was added in results section 3.1 Abundance and alpha diversity (page 8 lines 29 to 30):*

*"Interestingly, only a large specimen identified as* Bathyasychis *sp. 150 was found deeper than 50 cm (bottom of box core) and so not included in the analyses."*

Table 1 lists the data for all the box core stations. The only actual result listed in the table is the "nodules density", actually the nodule wet weight, extrapolated to the weight per m-2. Firstly this is not the density and secondly all the biological data is recorded per box core area, i.e. in 0.25 m-2. The table should include the basic polychaete data per box core, i.e. total numbers and number of "species" so that readers can follow the author's analysis rather than having to extract the individual core data from the database in PANGEA.

*The columns "Total abundance (ind. 0.25 m$^{-2}$)" and "Number of species (taxa 0.25 m$^{-2}$)" were added in Table 1 (below or page 28) and the caption was changed accordingly.*

*From "Table 1. Area, locality, station, date, depth, geographical position and nodule density of all 34 box corer deployments across the CCFZ during the SO239 cruise. "*" indicates box cores considered as non-quantitative, not included in the analyses."*

*To "Table 1. **Details of sampling,** nodule density and **descriptors of alpha diversity of** all 34 box corer deployments across the CCFZ during the SO239 cruise. "*" indicates box cores considered as non-quantitative, not included in the analyses."*

*Additionally, the asterisk indicating non-quantitative boxes were transferred from the column "Area" to the column "Station" where they are clearer.*

| Area | Locality | Station | Date | Depth (m) | Latitude | Longitude | Nodule density (kg m$^{-2}$) | Total abundance (ind. 0.25 m$^{-2}$) | Number of species (taxa 0.25 m$^{-2}$) |
|---|---|---|---|---|---|---|---|---|---|
| **BGR** | BGR-PA | 12 | 20/03/15 | 4118 | 11.8471667 | -117.05933 | 26.40 | 32 | 24 |
| **BGR** | BGR-PA | 15 | 21/03/15 | 4133 | 11.8443333 | -117.05217 | 26.80 | 67 | 40 |
| **BGR** | BGR-PA | 16 | 21/03/15 | 4122 | 11.8573333 | -117.052 | 24.00 | 52 | 34 |
| **BGR** | BGR-PA | 21 | 22/03/15 | 4120 | 11.8535 | -117.0595 | 22.80 | 43 | 28 |
| **BGR** | BGR-PA | 23 | 22/03/15 | 4122 | 11.85 | -117.05267 | 20.80 | 69 | 47 |
| **BGR** | BGR-RA | 51* | 27/03/15 | 4348 | 11.8236667 | -117.52367 | 0.00 | 22 | 12 |
| **BGR** | BGR-RA | 57 | 28/03/15 | 4370 | 11.8075 | -117.52433 | 8.00 | 43 | 24 |
| **BGR** | BGR-RA | 58 | 28/03/15 | 4350 | 11.8205 | -117.54167 | 1.60 | 89 | 47 |
| **BGR** | BGR-RA | 60 | 29/03/15 | 4325 | 11.8076667 | -117.55033 | 18.00 | 65 | 48 |
| **IOM** | IOM-control | 88 | 02/04/15 | 4433 | 11.079 | -119.65883 | 0.00 | 53 | 33 |
| **IOM** | IOM-control | 89 | 02/04/15 | 4437 | 11.0758333 | -119.66083 | 1.20 | 38 | 29 |
| **IOM** | IOM-control | 90 | 03/04/15 | 4434 | 11.074 | -119.66417 | 0.00 | 42 | 24 |
| **IOM** | IOM-disturb | 94 | 03/04/15 | 4414 | 11.0736667 | -119.6555 | 0.40 | 38 | 28 |
| **IOM** | IOM-disturb | 95 | 03/04/15 | 4418 | 11.0735 | -119.65583 | 0.80 | 43 | 28 |
| **IOM** | IOM-disturb | 97 | 04/04/15 | 4421 | 11.0728333 | -119.65617 | 0.20 | 22 | 16 |
| **IOM** | IOM-resed | 105* | 05/04/15 | 4423 | 11.0711667 | -119.65533 | 0.00 | 13 | 9 |
| **IOM** | IOM-resed | 106 | 05/04/15 | 4425 | 11.0716667 | -119.65483 | 0.20 | 23 | 18 |

| | | | | | | | | | |
|---|---|---|---|---|---|---|---|---|---|
| **IOM** | IOM-resed | 107 | 05/04/15 | 4425 | 11.0721667 | -119.6545 | 0.30 | 38 | 26 |
| **GSR** | GSR | 119 | 08/04/15 | 4516 | 13.8591667 | -123.25267 | 26.47 | 46 | 29 |
| **GSR** | GSR | 127 | 09/04/15 | 4514 | 13.8443333 | -123.246 | 27.10 | 59 | 32 |
| **GSR** | GSR | 128 | 09/04/15 | 4511 | 13.8516667 | -123.252 | 27.10 | 58 | 32 |
| **GSR** | GSR | 137 | 11/04/15 | 4510 | 13.856 | -123.238 | 25.20 | 60 | 34 |
| **GSR** | GSR | 138 | 11/04/15 | 4503 | 13.8481667 | -123.23467 | 26.47 | 74 | 48 |
| **Ifremer** | Ifremer | 159 | 15/04/15 | 4921 | 14.049 | -130.13433 | 19.80 | 30 | 23 |
| **Ifremer** | Ifremer | 162 | 16/04/15 | 4951 | 14.049 | -130.126 | 20.20 | 34 | 21 |
| **Ifremer** | Ifremer | 169 | 17/04/15 | 4964 | 14.0421667 | -130.12733 | 24.10 | 25 | 15 |
| **Ifremer** | Ifremer | 180 | 18/04/15 | 4936 | 14.0416667 | -130.13633 | 16.00 | 19 | 17 |
| **Ifremer** | Ifremer | 181 | 18/04/15 | 4896 | 14.0465 | -130.1415 | 16.80 | 38 | 27 |
| **Ifremer** | Ifremer | 182 | 18/04/15 | 4957 | 14.0423333 | -130.1275 | 22.40 | 19 | 13 |
| **APEI#3** | APEI#3 | 195 | 21/04/15 | 4833 | 18.7958333 | -128.36217 | 6.28 | 4 | 3 |
| **APEI#3** | APEI#3 | 196 | 21/04/15 | 4847 | 18.7971667 | -128.34617 | 1.80 | 7 | 5 |
| **APEI#3** | APEI#3 | 203* | 23/04/15 | 4843 | 18.774 | -128.35317 | 2.88 | 3 | 2 |
| **APEI#3** | APEI#3 | 204 | 23/04/15 | 4816 | 18.7733333 | -128.33617 | 3.65 | 3 | 2 |
| **APEI#3** | APEI#3 | 209* | 24/04/15 | 4819 | 18.7845 | -128.3725 | 3.65 | 3 | 3 |

The authors recognise that the estimated number of polychaete species and the average species range in the nodule province of the CCFZ are not well constrained as a result of the limited sampling possible on this cruise. It would therefore be helpful if the authors could discuss what sampling effort might be required to obtain reasonable estimates of these.

*At the moment, there is no example of a sampling effort allowing for rarefaction curves to level off, not even at DOMES A where 41 box-cores have been sampled. To our knowledge, there is no robust statistics allowing to extrapolate the number of samples required to obtain a reasonable estimate of the number of species at local or regional scales. We have suggested the following sentences in the Conclusions (page 17 lines 20 to 26):*

*"In the framework of a similarly ambitious and collective effort to inventory species richness in the CCFZ, a stratified random sampling at nested scales, from region down to seascapes, would provide the scales of species turn-over while intensive sampling of selected habitats up to the point where the number of singletons decreases with sample size would provide accurate estimates of species diversity. Both strategies should consider different taxonomic and functional groups of the abyssal fauna, which are likely to show different responses to nodule mining. Such an approach, based on standardized sampling methods and statistical-wise sampling strategies is needed to assess the potential risks and scales of biodiversity loss due to nodule mining in the CCFZ."*

*References added in the answers:*

*Thiel, H., Schriever, G., Bussau, C. and Borowski, C.: Manganese nodule crevice fauna, Deep Sea Res. Part 1 Oceanogr. Res. Pap., 40(2), 419–423, doi:10.1016/0967-0637(93)90012-R, 1993.*

---

## Author Response (AR3)

Object:
Correction of the accepted manuscript to Biogeosciences special issue *Assessing environmental impacts of deep-sea mining – revisiting decade-old benthic disturbances in Pacific nodule areas*

From Paulo,
To Editor and Associate Editor of the Biogeosciences

Brest, January 22th 2020.

Dear Dr. Treude,

Thank you, we are happy that our manuscript is accepted to be published in Biogeosciences special issue "Assessing environmental impacts of deep-sea mining – revisiting decade-old benthic disturbances in Pacific nodule areas".

However, as explained in our manuscript (page 7 lines 14 to 15): "A few cryptic or damaged specimens that could not be classified to a lower taxonomical level were included in total abundance calculations but excluded from subsequent diversity analyses."

Consequently, the suggested technical corrections are not completely accurate because in the section 3.1 "Abundance and alpha diversity" the first parts (page 8 line 29 to page 9 line 18) are about the total abundance (counting all polychaetes sampled from box cores) whereas the second part (page 9 line 19 to page 10 line 9) being about the number of species (counting only polychaetes identified as morphoespecies). Thus, the table 1 makes references to "Total abundance" and "Number of species". We have changed the caption of Table 1 (page 27) to make it clearer:

From:
Table 1. Details of sampling, nodule density and descriptors of alpha diversity of all 34 box corer deployments across the CCFZ during the SO239 cruise. "*" indicates box cores considered as non-quantitative, not included in the analyses.

To:
Table 1. Details of sampling, nodule density, **total number of polychaete specimens and number of polychaete species** of all 34 box corer deployments across the CCFZ during the SO239 cruise. "*" indicates box cores considered as non-quantitative, not included in the analyses.

We hope you agree with this.
We thank you and we are looking forward to hearing from you.

Best wishes,
Paulo, Lénaïck and Pedro

**Institut français de Recherche pour l'Exploitation de la Mer**

Etablissement public à caractère industriel et commercial

**Centre de Brest**
Zone Industrielle de la Pointe du Diable
CS10070
29280 Plouzané
France

téléphone  33 (0)2 98 22 40 40
télécopie   33 (0)2 98 22 45 45
http://www.ifremer.fr

**Siège social**
155, rue Jean-Jacques Rousseau
92138 Issy-les-Moulineaux Cedex
France

R.C.S. Nanterre B 330 715 368
APE 7219Z
SIRET 330 715 368 00297
TVA FR 46 330 715 368

téléphone  33 (0)1 46 48 21 00
télécopie   33 (0)1 46 48 21 21
http://www.ifremer.fr

[revised manuscript text omitted]